# Controllable Sequence Editing for Biological and Clinical Trajectories

**Michelle M. Li**[1,*]**, Kevin Li**[2]**, Yasha Ektefaie**[1]**, Ying Jin**[1,3]**, Yepeng Huang**[1]**, Shvat Messica**[1]**, Tianxi Cai**[1]**, Marinka Zitnik**[1,*]

[1]Harvard University, [2]MIT, [3]University of Pennsylvania, [*]Co-corresponding authors
`michelleli,yasha_ektefaie,yepeng@g.harvard.edu, likevin@mit.edu,`
`yjinstat@wharton.upenn.edu, shvat_messica@fas.harvard.edu,`
`tcai@hsph.harvard.edu, marinka@hms.harvard.edu`

## Abstract

Conditional generation models for longitudinal sequences can produce new or modified trajectories given a conditioning input. However, they often lack control over when the condition should take effect (*timing*) and which variables it should influence (*scope*). Most methods either operate only on univariate sequences or assume that the condition alters all variables and time steps. In scientific and clinical settings, interventions instead begin at a specific moment, such as the time of drug administration or surgery, and influence only a subset of measurements while the rest of the trajectory remains unchanged. CLEF learns temporal concepts that encode how and when a condition alters future sequence evolution. These concepts allow CLEF to apply targeted edits to the affected time steps and variables while preserving the rest of the sequence. We evaluate CLEF on 8 datasets spanning cellular reprogramming, patient health, and sales, comparing against 9 state-of-the-art baselines. CLEF improves immediate sequence editing accuracy by 16.28% (MAE) on average against their non-CLEF counterparts. Unlike prior models, CLEF enables one-step conditional generation at arbitrary future times, outperforming their non-CLEF counterparts in delayed sequence editing by 26.73% (MAE) on average. We test CLEF under counterfactual inference assumptions and show up to 62.84% (MAE) improvement on zero-shot conditional generation of counterfactual trajectories. In a case study of patients with type 1 diabetes mellitus, CLEF identifies clinical interventions that generate realistic counterfactual trajectories shifted toward healthier outcomes.

## 1 Introduction

Conditional generation of longitudinal sequences is a growing challenge in machine learning, where the goal is to produce new or modified trajectories based on a conditioning input, such as an intervention applied to a system. A central task is controlling when in the trajectory the condition should take effect (*timing*) and which subset of variables it should influence (*scope*). In many domains, interventions begin at a specific moment and alter only part of the

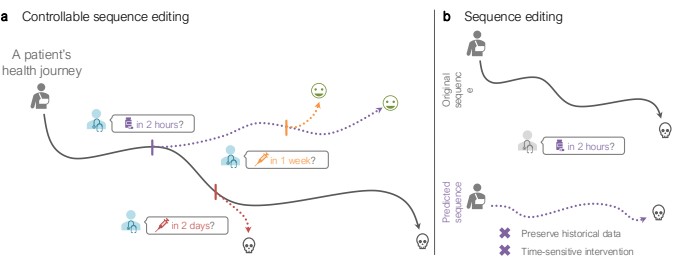

**Figure 1:** Illustrative comparison of **(a)** controllable sequence editing and **(b)** existing sequence editing. Unlike existing methods, controllable sequence editing generates sequences (dotted lines) guided by a condition while preserving historical data to model the effects of immediate (e.g., in 2 hours) or delayed (e.g., in 1 week) edits.

system while the rest of the trajectory remains unchanged. A motivating example comes from virtual cell models, which simulate how molecular or cellular states evolve under perturbations and enable large-scale *in silico* experimentation (Bunne et al., 2024; Li et al., 2025). To successfully build such virtual cells and patients, the models must consider both the type of intervention (e.g., drug, surgery) and its timing (e.g., when and how frequent). For instance, prescribing a medication should alter

a patient's trajectory only after the intervention time while preserving the medical history prior to treatment, and only those clinical variables relevant to the intervention should change while unaffected measurements remain stable (Fig. 1).

Generative language and vision models enable precise editing guided by a description, such as textual prompts or condition tokens (Zhang et al., 2023b; Gao et al., 2023; Ravi et al., 2024; Gong et al., 2024; Niu et al., 2024; Gu et al., 2025; Zhou et al., 2024). They are designed to gain more *global* context-preserving and *local* precise control over the generation of text (Chatzi et al., 2025; Niu et al., 2024; Gu et al., 2025; Zhou et al., 2024), images (Zhang et al., 2023b; Gao et al., 2023; Ravi et al., 2024), and even molecular structures (Gong et al., 2024; Dauparas et al., 2022; Zhang et al., 2024b). Their outputs preserve the input's global integrity yet contain precise local edits to satisfy the desired condition. Analogous to these models' consideration of spatial context to edit images (Zhang et al., 2023b; Gao et al., 2023) and protein pockets (Dauparas et al., 2022; Zhang et al., 2024b) via in-painting, **our work leverages temporal context to edit sequences based on a given condition**.

Controllable text generation (CTG), designed specifically to edit natural language sequences, has been extensively studied (Zhang et al., 2023a). **They excel in *immediate sequence editing***: predicting the next token or readout in the sequence under a given condition (Niu et al., 2024; Gu et al., 2025; Zhou et al., 2024; Chatzi et al., 2025; Zhang et al., 2023a; Bhattacharjee et al., 2024). However, **CTG models are unable to perform *delayed sequence editing***: modifying a trajectory in the far-future. The distinction is important: the focus is on when the edit occurs, not necessarily when its effects manifest. Whereas immediate sequence editing applies a condition *now* (e.g., administering insulin *today*), delayed sequence editing schedules a condition for the *future* (e.g., starting a chemotherapy regimen *in six weeks*). Existing CTG models cannot effectively utilize the given context to skip ahead to the future; instead, they would need to be run repeatedly to fill in the temporal gap without any guarantee of satisfying the desired condition. As a result, **CTG models are insufficient for other sequence types (i.e., not natural language) for which both immediate and delayed sequence editing are necessary**, such as cell development and patient health trajectories.

Controllable time series generation (CTsG) (Jing et al., 2024; Bao et al., 2024) utilizes diffusion modeling to generate time series under a given condition. However, these models are limited to univariate sequences and assume that the entire input sequence is affected (Jing et al., 2024; Bao et al., 2024). These methods are thus insufficient in settings where edits are only allowed after time $t$ (i.e., cannot change historical data) and affect only certain sequences (i.e., preserve unaffected co-occurring sequences). In other words, **CTsG methods are unable to make precise local edits while preserving global integrity**. Orthogonal to CTsG is the estimation of counterfactual outcomes over time (ECT) (Melnychuk et al., 2022; Bica et al., 2020; Huang et al., 2024; Wang et al., 2025). Although not generative, ECT autoregressively predicts the potential outcomes (i.e., next readout in the sequence) as a result of different future treatments (i.e., fixed set of conditions) under counterfactual inference assumptions. **While ECT preserves historical and unaffected co-occurring sequences, counterfactual inference assumptions may not always hold in real-world applications**.

**Present work.** We develop CLEF, a novel ControLlable sequence Editing Framework for instance-wise conditional generation. Specifically, CLEF learns *temporal concepts* that represent the trajectories of the sequences to enable accurate generation guided by a given condition (Def. 3.2). We show that the learned temporal concepts help preserve temporal constraints in the generated outputs. CLEF is flexible and can be used with any sequential data encoder and condition tokenizer. We evaluate CLEF on 8 datasets spanning cellular reprogramming, patient health trajectories, and sales, outperforming 9 state-of-the-art (SOTA) baselines on immediate and delayed sequence editing (Def. 3.1). We also show that any pretrained sequence encoder can gain controllable sequence editing capabilities when finetuned with CLEF. Additionally, we extend CLEF for multi-step ahead counterfactual prediction under counterfactual inference assumptions (Assumption 3.4, Eq. 4), and demonstrate (on 3 benchmark datasets) performance gains against 5 SOTA methods in settings with high time-varying confounding. Moreover, CLEF enables conditional generation models to outperform baselines in zero-shot generation of counterfactual cellular trajectories on immediate and delayed sequence editing. Further, precise edits via user interaction can be performed directly on CLEF's learned concepts. We show through real-world case studies that CLEF, given precise edits on specific temporal concepts, can generate realistic "healthy" trajectories for patients originally with type 1 diabetes mellitus.

**Our contributions are fourfold.** (1) We develop CLEF: a flexible controllable sequence editing model for conditional generation of longitudinal sequences. (2) CLEF can be integrated into the

(balanced) representation learning architectures of counterfactual prediction models to estimate counterfactual outcomes over time. (3) Beyond achieving SOTA performance in conditional sequence generation and counterfactual outcomes prediction, CLEF excels in zero-shot conditional generation of counterfactual sequences. (4) We release four new benchmark datasets on cell reprogramming and patient immune dynamics for immediate and delayed sequence editing, and evaluate on four established benchmark datasets for conditional generation and counterfactual prediction.

## 2 RELATED WORK (EXTENDED VERSION IN APPENDIX A)

**Sequence editing.** The sequence editing task has been defined in language and time series modeling via different terms, but share a core idea: Given a sequence and a condition (e.g., sentiment, attribute), generate a sequence with the desired properties. Conditional sequence generation is an autoregressive process in language (Chatzi et al., 2025) but a diffusion process in time series (Jing et al., 2024; Bao et al., 2024). Prompting is often used to guide the generation of a sequence, both textual and temporal, with a desired condition (Zhang et al., 2023a; Bhattacharjee et al., 2024; Jing et al., 2024; Bao et al., 2024). However, existing approaches are unable to generate multivariate sequences, preserve relevant historical data, and ensure time-sensitive edits. They assume that sequences are univariate and conditions affect the entire sequence (Jing et al., 2024; Bao et al., 2024). Structural causal models can be incorporated to enable counterfactual text generation while preserving certain attributes (Chatzi et al., 2025; Ravfogel et al., 2025). Estimating counterfactual outcomes over time is often formulated under the potential outcomes framework (Neyman, 1923; Rubin, 1978).

**Estimating counterfactual outcomes over time.** Predicting time-varying counterfactual outcomes entails estimating counterfactual outcomes over possible sequences of interventions (e.g., timing and ordering of sequential treatments (Melnychuk et al., 2022; Bica et al., 2020; Huang et al., 2024; Wang et al., 2025)). There are decades of research on temporal counterfactual outcomes estimation (Robins et al., 2000; Lim, 2018; Bica et al., 2020; Li et al., 2021; Melnychuk et al., 2022). Recently, machine learning models that predict time-varying counterfactual outcomes learn representations that are predictive of outcomes while mitigating treatment bias via balancing techniques (Melnychuk et al., 2022; Bica et al., 2020; Huang et al., 2024; Wang et al., 2025). On images, conditional generation models (i.e, guided diffusion, conditional variational autoencoder) have been shown to predict counterfactual outcomes without an explicit density estimation (Wu et al., 2024). However, there may be a trade-off between prediction accuracy and balanced representations (Huang et al., 2024).

**Concept-based learning.** Concepts are abstract atomic ideas or concrete tokens of text and images (LCM et al., 2024; Lai et al., 2024). Concept-based learning can explain (e.g., predict concepts in the sample) or transform black-box models into more explainable models (e.g., allow user intervention) (Koh et al., 2020; Shin et al., 2023; Ismail et al., 2024; Lai et al., 2024; Laguna et al., 2024; van Sprang et al., 2024), and mitigate distribution shifts (Zarlenga et al., 2025). While concepts are used in sequence generation (LCM et al., 2024), they have not been used for conditional generation. Concept-based learning for counterfactual prediction is limited to improving the interpretability of image classification (Dominici et al., 2025a; De Felice et al., 2025; Dominici et al., 2025b).

## 3 CLEF

CLEF generates sequences based on user-specified conditions and temporal coordinates. Given a longitudinal sequence, forecast time step, and condition token, CLEF modifies only the relevant portions of the sequence while preserving unaffected elements, ensuring global integrity. Architecturally, CLEF has two **novel** components: **concept encoder** $E$ that learns temporal concepts, representing trajectory patterns over time, and **concept decoder** $G$ that applies these concepts to generate sequences (Sec. A.3). Following SOTA conditional sequence generation and time-varying counterfactual prediction models, CLEF has a sequence encoder $F$ that extracts temporal features from historical sequence data, and a condition adapter $H$ that maps condition tokens to latent representations (Sec. A.3).

### 3.1 PROBLEM DEFINITION

Consider an observational dataset $\mathcal{D} = \{\mathbf{x}_t^{(i)}, \mathbf{s}_t^{(i)}\}_{i=1}^N$ for $N$ independent entities (e.g., cells, patients) at discrete time step $t$. For each entity $i$ at time $t$, we observe continuous time-varying covariates

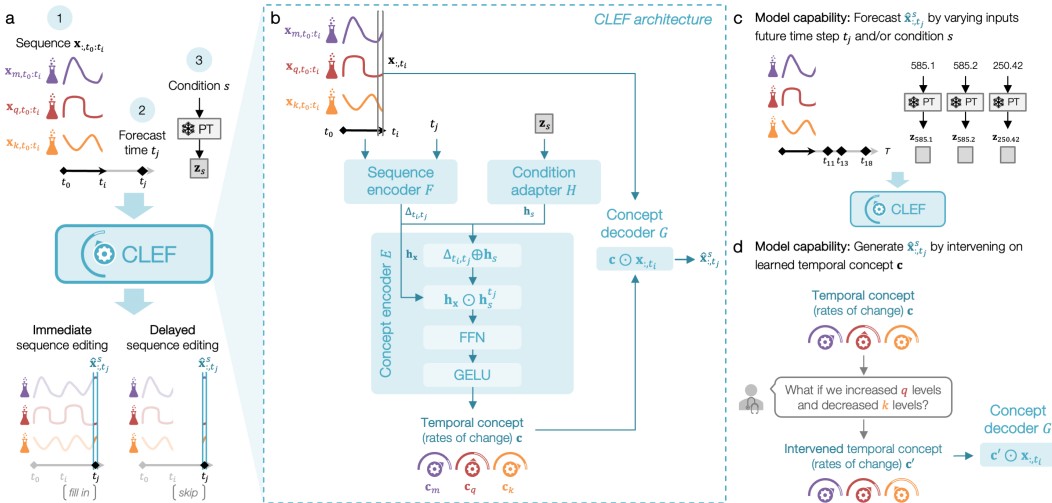

**Figure 2:** Overview of CLEF's architecture and capabilities. **(a)** Given a sequence, forecast time, and condition embedding from a frozen pretrained (PT) embedding model, CLEF generates a sequence via immediate or delayed sequence editing. **(b)** CLEF is composed of a sequence encoder, condition adapter, concept encoder, and concept decoder. CLEF has two key capabilities: **(c)** forecast sequences at any future time and under any condition (e.g., medical codes), and **(d)** generate sequences by intervening on CLEF's learned temporal concepts.

$\mathbf{x}_t^{(i)} \in \mathbb{R}^{d_x}$ (e.g., gene expression, laboratory test measurements) and categorical conditions $\mathbf{s}_t^{(i)}$ (e.g., transcription factor activation, clinical intervention). The outcome of the condition is measured by the covariates (e.g., activating a transcription factor affects a cell's gene expression, prescribing a medication affects a patient's laboratory test profile). We omit entity index $(i)$ unless needed.

**Definition 3.1** (Sequence editing). Sequence editing is the local sample-level modification of sequence $\mathbf{x}$ to autoregressively generate $\hat{\mathbf{x}}_{:,t_j}$ under condition $s$ given at time $t_j - \epsilon$. Time gap $\epsilon$ indicates that $\hat{\mathbf{x}}_{:,t_j}$ is measured a negligible amount of time after $s$ is applied[1]; for notation, we omit $\epsilon$ unless needed. There are two types of controllable sequence editing (Fig. 2a):

- **Immediate sequence editing:** Given $\mathbf{x}_{:,t_0:t_i}$ and $s$ to occur at $t_{i+1}$, forecast $\hat{\mathbf{x}}_{:,t_{i+1}}$
- **Delayed sequence editing:** Given $\mathbf{x}_{:,t_0:t_i}$ and $s$ to occur at $t_j \geq t_{i+1}$, forecast $\hat{\mathbf{x}}_{:,t_j}$

Examples of immediate sequence editing include generating trajectories after perturbing cells *now* or performing surgery on patients *today* (Sec. 5.1). In contrast, delayed sequence editing generates trajectories after perturbing cells *in ten days* or performing surgery on patients *next year* (Sec. 5.2).

**Definition 3.2** (Temporal concept). Temporal concept $\mathbf{c}$ approximates the trajectory (or rate of change of each variable in sequence $\mathbf{x}$) between a pair of time steps $t_j > t_i$ such that $\mathbf{x}_{:,t_j} = \mathbf{c} \odot \mathbf{x}_{:,t_i}$.

**Definition 3.3** (Controllable sequence editing). Concept encoder $E$ and decoder $G$ can leverage temporal concepts $\mathbf{c}$ to perform controllable sequence editing if the following are satisfied.

- Condition $s$ on $\mathbf{x}_{:,t_0:t_i}$ at time step $t_j$ learns $\mathbf{c}$ that accurately forecasts $\hat{\mathbf{x}}_{:,t_j}^s$ such that $\hat{\mathbf{x}}_{:,t_j}^s \simeq \mathbf{x}_{:,t_j}^s$.
- For an alternative condition $a \neq s$ on $\mathbf{x}_{:,t_0:t_i}$ at $t_j$, the method learns a distinct $\mathbf{c}' \neq \mathbf{c}$ that forecasts $\hat{\mathbf{x}}_{:,t_j}^a$ such that $\hat{\mathbf{x}}_{:,t_j}^a \neq \hat{\mathbf{x}}_{:,t_j}^s$ and, if known, $\hat{\mathbf{x}}_{:,t_j}^a \simeq \mathbf{x}_{:,t_j}^a$.

**Problem Statement 3.1** (CLEF). Given a sequence encoder $F$, condition adapter $H$, concept encoder $E$, and concept decoder $G$ trained on a longitudinal dataset $\mathcal{D}$, CLEF learns temporal concept $\mathbf{c} = E\big(F(\mathbf{x}_{:,t_0:t_i}, t_j), H(s)\big)$ to forecast $\hat{\mathbf{x}}_{:,t_j}^s = G(\mathbf{x}_{:,t_i}, \mathbf{c})$ for any $\mathbf{x}_{:,t_0:t_i} \in \mathcal{D}$, $t_j > t_i$, and $s$.

$$\hat{\mathbf{x}}_{:,t_j}^s = G\big(\mathbf{x}_{:,t_i}, E(F(\mathbf{x}_{:,t_0:t_i}, t_j), H(s))\big) \tag{1}$$

---

[1]We assume that the condition $s$ always occurs shortly before the measured covariates at time $t_j$. When $s$ is an intervention and our problem becomes counterfactual prediction (refer to Sec. 3.3 for a more rigorous discussion about the formulation of sequence editing in this context), our assumption is consistent with existing practice in the literature of counterfactual prediction.

## 3.2 CLEF ARCHITECTURE

CLEF's input are a continuous multivariate sequence $\mathbf{x}_{:,t_0:t_i} \in \mathbb{R}^V$ with $V$ measured variables, a condition $s$, and time $t_j > t_i$ for which to forecast $\hat{\mathbf{x}}^s_{:,t_j}$. CLEF consists of 4 major components: sequence encoder $F$, condition adapter $H$, concept encoder $E$, and concept decoder $G$ (Sec. A.3).

**Sequence encoder $F$.** The sequence encoder $F$ extracts features $\mathbf{x}_{:,t_0:t_i}$ such that $\mathbf{h_x} = F(\mathbf{x}_{:,t_0:t_i})$. Any sequential data encoder, including a pretrained multivariate foundation model, can be used. The time encoder in $F$ generates a time positional embedding $\mathbf{h}_t$ for any time $t$ via element-wise summation of the year (sinusoidal), month, date, and hour embeddings. It is also used to compute the time delta embedding $\Delta_{t_i,t_j} = \mathbf{h}_{t_j} - \mathbf{h}_{t_i}$ of time steps $t_i$ and $t_j$ for the concept encoder $E$.

**Condition adapter $H$.** The condition token, or embedding $\mathbf{z}_s$ corresponding to the input condition $s$, is either retrieved from a frozen pretrained embedding model (denoted as PT in Fig. 2a) or features of a condition/intervention. The condition adapter $H$ projects $\mathbf{z}_s$ into a hidden representation $\mathbf{h}_s = H(s)$.

**Concept encoder $E$.** Given the hidden representations generated by $F$ and $H$, concept encoder $E$ learns temporal concepts $\mathbf{c} = E(\mathbf{h_x}, \Delta_{t_i,t_j}, \mathbf{h}_s)$. First, the time delta embedding $\Delta_{t_i,t_j}$ is combined via summation with $\mathbf{h}_s$ to generate a time- and condition-specific embedding $\mathbf{h}_s^{t_j} = \Delta_{t_i,t_j} \oplus \mathbf{h}_s$. Then, $\mathbf{c}$ is learned via an element-wise multiplication of $\mathbf{h_x}$ and $\mathbf{h}_s^{t_j}$, an optional linear projection (FNN), and a GELU activation to approximate the trajectory between $t_i$ and $t_j$.

$$\mathbf{c} = \text{GELU}(\text{FFN}(\mathbf{h_x} \odot \mathbf{h}_s^{t_j})) \qquad (2)$$

**Concept decoder $G$.** The concept decoder $G$ forecasts $\hat{\mathbf{x}}^s_{:,t_j}$ by performing element-wise multiplication of the latest time $t_i$ of the input sequence $\mathbf{x}_{:,t_0:t_i}$ (denoted as $\mathbf{x}_{:,t_i}$) and the learned concept $\mathbf{c}$

$$\hat{\mathbf{x}}^s_{:,t_j} = \mathbf{c} \odot \mathbf{x}_{:,t_i} \qquad (3)$$

This decoder is less sensitive to covariates with different units of measure (e.g., blood sodium in meq/L vs. white blood cell in $10^9$/L). Also, applying $\mathbf{c}$ in a single step (via $\odot$) allows users to directly intervene on $\mathbf{c}$ and simulate the effects of the intervention as a counterfactual trajectory (Sec. 5.6).

**Objective function $\mathcal{L}$** quantifies the reconstruction error of predicted $\hat{\mathbf{x}}^s_{:,t_j}$ from ground truth $\mathbf{x}^s_{:,t_j}$ (via Huber or MSE). It may include balancing loss functions for counterfactual prediction only (Sec. D.1).

## 3.3 CLEF'S CONNECTION TO COUNTERFACTUAL PREDICTION

Let $\mathbf{x}_{:,t_j}$ refer to the outcomes observed at $t_j$ after treatment $s$ is given. Our problem can be viewed as counterfactual prediction when there is no treatment assigned between $t_i$ and $t_j$ except $s$.

Formally, under the potential outcomes framework (Neyman, 1923; Rubin, 1978) and its extension to time-varying treatments and outcomes (Robins & Hernan, 2008), the potential counterfactual outcomes over time are identifiable from the observational data $\mathcal{D}$ under three standard assumptions: consistency, positivity, and sequential ignorability (Sec. B). Thus, CLEF (via temporal concepts $\mathbf{c}$) predicts counterfactuals under the additional Assumption 3.4:

**Assumption 3.4** (Conditional mean function estimation). For time steps $t_j > t_i$, temporal concepts $\mathbf{c}$ learned based on the next treatment $\mathbf{s}_{t_j}$, historical treatments $\mathbf{s}_{t_0:t_i}$, and historical covariates $\mathbf{x}_{:,t_0:t_i}$ capture (balanced) representations such that the concept decoder $\mathbf{c}(\mathbf{s}_{t_j}, \mathbf{s}_{t_0:t_i}, \mathbf{x}_{:,t_0:t_i}) \odot \mathbf{x}_{:,t_i}$ approximates the conditional mean function $\mathbb{E}[\mathbf{x}_{:,t_j+\epsilon}(\mathbf{s}_{t_j}, \mathbf{s}_{t_0:t_i})|\mathbf{s}_{t_0:t_i}, \mathbf{x}_{:,t_0:t_i}]$.

In the following, we elaborate on why it can be reasonable to view CLEF as an accurate counterfactual prediction model by satisfying Assumption 3.4.

**Estimating counterfactuals.** We estimate future counterfactual outcomes over time, formulated as

$$\mathbb{E}(\mathbf{x}_{:,t_j+\epsilon}(\mathbf{s}_{t_j}, \mathbf{s}_{t_0:t_i})|\mathbf{s}_{t_0:t_i}, \mathbf{x}_{:,t_0:t_i}) \qquad (4)$$

by learning a function $g(\tau, \mathbf{s}_{t_j}, \mathbf{s}_{t_0:t_i}, \mathbf{x}_{:,t_0:t_i}) = G(\mathbf{x}_{:,t_i}, E(F(\mathbf{x}_{:,t_0:t_i}, t_j), H(\mathbf{s}_{t_j})))$ with projection horizon $\tau = (t_j + \epsilon) - t_i \geq 1$ for $\tau$-step ahead prediction (Eq. 1; Sec. 3.2). Indeed, the key to reliable counterfactual prediction is the accurate estimation of Eq. 4 to adjust for bias introduced by time-varying confounders (Robins & Hernan, 2008). In particular, our design of $g(\cdot)$ estimates

Eq. 4 well (refer to Sec. 5.4 for empirical results) due to the effective learning of temporal concepts (Def. 3.2) and the strong representation power of the encoders (Sec. 3.2).

**Balancing representations via CLEF (Sec. D.4).** Since the historical covariates and next treatment are encoded independently by $F$ and $H$, the learned representations are treatment-invariant (or balanced), following the discussions in existing balanced representation learning architectures (e.g., CRN (Bica et al., 2020), CT (Melnychuk et al., 2022)). Further, by Assumption 3.4, our designed structure isolates the causal effect of the treatment from other spurious factors, enabling reliable counterfactual estimation (Zhang et al., 2024a).

## 4 EXPERIMENTAL SETUP

### 4.1 DATASETS

CLEF is evaluated on **8 biomedical and financial datasets on conditional and counterfactual generation** (Fig. 3; Sec. C; Tab. 1). We introduce benchmarking datasets, **WOT** (conditional generation) and **WOT-CF** (counterfactual generation), of single-cell transcriptomic profiles of developmental time courses of cells. We also construct two new real-world patient datasets of irregularly-measured routine laboratory tests from **eICU** (Pollard et al., 2018) and **MIMIC-IV** (Johnson et al., 2024a; 2023; Goldberger et al., 2000).

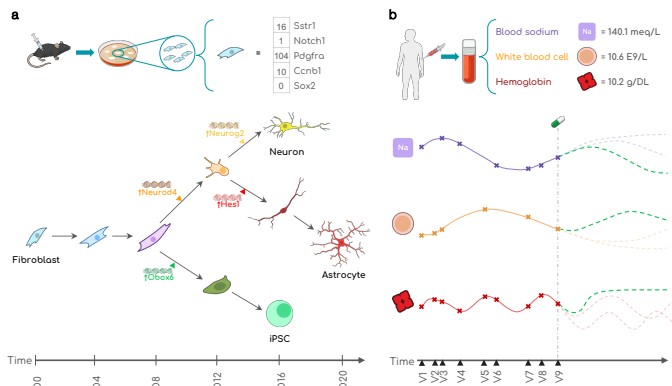

**Figure 3:** CLEF is evaluated on 7 datasets of **(a)** cellular development and **(b)** patient health trajectories. Illustrations from NIAID NIH BIOART.

We evaluate counterfactual outcomes estimation on three established benchmarks related to **tumor growth** (Geng et al., 2017) and patient intensive care units (**ICU**) (Johnson et al., 2016) trajectories for $\tau$-step ahead prediction; trajectories with time-varying confounding $\gamma$ are simulated (Yang et al., 2023). We evaluate conditional generation on real-world store sales trajectories: **M5** (Makridakis et al., 2022; Huang et al., 2024; Wang et al., 2025).

### 4.2 SETUP

CLEF is evaluated on 3 tasks: immediate and delayed sequence editing (Def. 3.1) and counterfactual prediction (Sec. 3.3). We use standard metrics (MAE, RMSE, $R^2$) to compare ground truth $\mathbf{x}^s_{:,t_j}$ and predicted $\hat{\mathbf{x}}^s_{:,t_j}$. **Experiments are designed to isolate temporal concepts' contribution to predictive performance** (e.g., CLEF and non-CLEF differ only in the components needed to learn temporal concepts; Sec. D.2). Refer to Sec. C-D for experimental setup and implementation details.

**Baselines.** We evaluate CLEF against SOTA conditional generation and counterfactual prediction models, which do not learn temporal concepts; **each baseline has a CLEF counterpart**. *Conditional generation (4):* We adopt the SOTA conditional sequence generation setup with 3 sequential encoders: Transformer (Waswani et al., 2017; Narasimhan et al., 2024; Jing et al., 2024; Zhang et al., 2023a); xLSTM (Beck et al., 2024); and time series foundation model, MOMENT (Goswami et al., 2024). We evaluate against traditional time series model, Vector Autoregression (VAR) (Lütkepohl, 2005). *Counterfactual prediction (5):* We adopt the SOTA counterfactual prediction setup using 2 architectures (i.e., Counterfactual Recurrent Network (CRN) (Bica et al., 2020) and Causal Transformer (CT) (Melnychuk et al., 2022)) with and without balancing loss functions (i.e., gradient reversal (GR) (Bica et al., 2020), counterfactual domain confusion (CDC) (Melnychuk et al., 2022)).

**Ablations.** SimpleLinear is an ablation in which temporal concepts are all ones (i.e., not learned nor meaningful), inspired by traditional linear models that excel when $\mathbf{x}_{t_j} \simeq \mathbf{x}_{t_i}$ (Toner & Darlow, 2024; Ahlmann-Eltze et al., 2025). We evaluate CLEF with and without an FFN layer in $E$ (Sec. E). LowR

is an ablated decoder $(\mathbf{I} + \mathbf{W})(\mathbf{c} \odot \mathbf{x})$ where $\mathbf{W}$ is low-rank (with rank $= 4, 8, 16$). To demonstrate the benefit of single-step generation, we perform an experiment in which we arbitrarily add three intermediate steps between $t_i$ and $t_j$ before finally predicting the observed $\mathbf{x}_{:,t_j}$ (Tab. 9).

## 5 RESULTS

We extensively evaluate the impact of CLEF's learned temporal concepts on controllable sequence editing. **R1-R3:** How well does CLEF perform in (R1) immediate and (R2) delayed sequence editing, and (R3) generalize to unseen/new sequences? **R4:** How does CLEF perform in counterfactual outcomes estimation? **R5:** Can CLEF perform zero-shot conditional generation of counterfactual sequences? **R6:** How can CLEF be leveraged for real-world patient trajectory simulations?

### 5.1 R1: IMMEDIATE SEQUENCE EDITING ON OBSERVED SEQUENCES

Immediate sequence editing involves forecasting the next time step of a trajectory after applying a condition. The defining feature is that the condition occurs in the present moment, and its effects are reflected in the next observation of the sequence. This setting is relevant when the condition has an instantaneous impact (e.g., administering a drug to a cell *now*, performing surgery on a patient *today*).

CLEF models consistently perform competitively against baseline models across all datasets (Fig. 4a, 8-9; Tab. 7, 15-16). SimpleLinear, which assumes no temporal changes, performs comparably in some cases, but CLEF outperforms it on datasets where short-term dynamics are more complex. On WOT, CLEF outperforms or performs comparably to the time series forecasting model, VAR. This is particularly exciting given recent findings that lin-

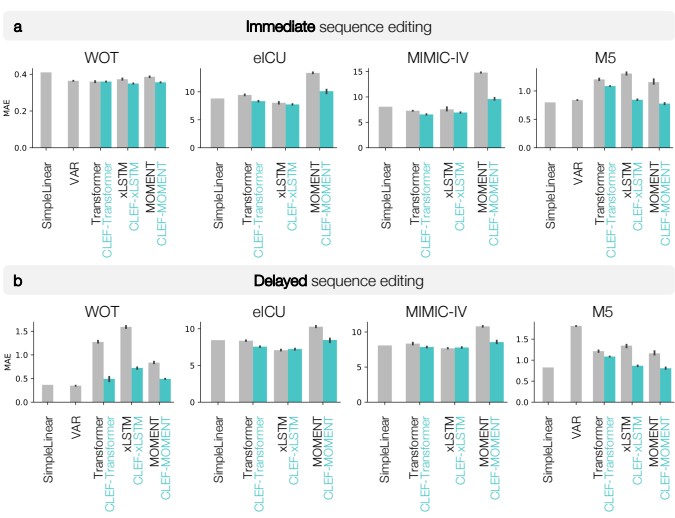

**Figure 4:** Benchmarking CLEF, baselines, and ablations on **(a)** immediate and **(b)** delayed sequence editing on observed sequences. Lower MAE is better. Models are trained on 3 seeds using a standard cell-, patient-, or store-centric random split; error bars show 95% CI. Not shown for visualization purposes are VAR's performance on eICU and MIMIC-IV: on immediate sequence editing, MAE for eICU and MIMIC-IV are 55982.74 and 886.05; on delayed sequence editing, MAE for eICU and MIMIC-IV are $3.02 \times 10^{39}$ and $8.62 \times 10^{23}$.

ear models can achieve competitive or better forecasting performance than neural network models (Toner & Darlow, 2024; Ahlmann-Eltze et al., 2025). Further, CLEF models tend to yield less error (MAE) on both preserved and edited variables of a sequence than non-CLEF models (Tab. 5). These results highlight CLEF's ability to accurately edit sequences at the desired times while preserving unaffected portions of the sequence.

Regardless of the sequence encoder used with CLEF, these models tend to outperform or perform comparably to non-CLEF models (Fig. 4a). However, CLEF's performance can be affected by the ability of the sequence encoder to capture the temporal dynamics of the input sequences. For instance, models with the MOMENT encoder generally yield the highest MAE in all three biomedical datasets (Fig. 4a). Still, CLEF-MOMENT models have lower MAE than their non-CLEF counterparts.

### 5.2 R2: DELAYED SEQUENCE EDITING ON OBSERVED SEQUENCES

Delayed sequence editing forecasts a trajectory at a specified future time step under a given condition while preserving sequence consistency. Unlike immediate editing, the condition takes effect at the

designated future time, requiring models to project forward without introducing compounding errors. Examples include applying a drug to a cell in *ten days* or scheduling a patient's surgery for *next year*.

CLEF performs competitively against SimpleLinear (ablation) and VAR on eICU, MIMIC-IV, and M5 (Fig. 4b, 8-9; Tab. 8). CLEF-transformer and CLEF-xLSTM achieve lower MAE than SimpleLinear, whereas non-CLEF transformer and MOMENT baselines perform comparably or worse. As in immediate sequence editing, models with MOMENT as the sequence encoder (i.e., using temporal concepts with MOMENT) yield the highest MAE on the biomedical sequences. However, incorporating CLEF with MOMENT reduces the MAE to levels comparable to the MAE of SimpleLinear and VAR.

On WOT, SimpleLinear and VAR outperform neural network models in delayed sequence editing (Fig. 4b). This suggests that cellular trajectories exhibit small and possibly noisy changes at each time step, favoring linear models (Ahlmann-Eltze et al., 2025; Toner & Darlow, 2024). Also, given the relatively small number of training trajectories compared to the high-dimensional state space, nonlinear models may overfit to noise more readily than linear models. Still, CLEF significantly reduces the MAE of non-CLEF models, demonstrating its effectiveness as a regularizer that mitigates short-term noise while preserving long-term trends. Further, as in immediate sequence editing, CLEF better preserves unedited variables than non-CLEF models (Tab. 5).

A benefit of delayed sequence editing is the ability to perform a single-step generation of any time $t_j$ in the future. To empirically demonstrate that single-step generation can help avoid compounding autoregressive error, we perform an experiment in which we arbitrarily add three intermediate steps between $t_i$ and $t_j$ before finally predicting the observed $\mathbf{x}_{:,t_j}$ (Tab. 9). On real-world patient datasets, we find that adding intermediate steps leads to compounding autoregressive error. There are a few possible explanations. The patient datasets have irregular and large time intervals (e.g., hours, days, weeks, months, years). Because MIMIC-IV and eICU capture patients in the intensive care unit, these patients' lab test measurements (e.g., white blood cell count, glucose) can change rapidly. Arbitrarily adding intermediate steps may introduce noise that compounds and degrades predictions at $t_j$.

## 5.3  R3: GENERALIZATION TO NEW PATIENT TRAJECTORIES VIA CONDITIONAL GENERATION

We assess CLEF's generalizability to new patient sequences. We create challenging data splits where the test sets have minimal similarity to the training data (Sec. C.2). CLEF models exhibit stronger generalization than non-CLEF models on both eICU and MIMIC-IV (Fig. 11-12; Tab. 6). For immediate and delayed sequence editing on eICU, CLEF-transformer and CLEF-xLSTM maintain stable and strong performance even as train/test divergence increases. In contrast, their non-CLEF counterparts degrade significantly. Although baseline MOMENT models show relatively stable performance across train/test splits in delayed sequence editing, they generalize poorly compared to CLEF-MOMENT models. Despite similar performance between xLSTM and CLEF-xLSTM in delayed sequence editing on both patient datasets (Fig. 4b), CLEF-xLSTM demonstrates superior generalizability, highlighting the effectiveness of CLEF in adapting to unseen data distributions.

## 5.4  R4: COUNTERFACTUAL OUTCOMES ESTIMATION

Following the setup of established benchmarks (Bica et al., 2020; Melnychuk et al., 2022) (Sec. D.4), we evaluate CLEF on counterfactual outcomes estimation of synthetic tumor growth and semi-synthetic ICU (Fig. 16) trajectories.

On the tumor growth and ICU trajectories, for which we have ground truth counterfactual sequences, CLEF consistently performs better or competitively against non-CLEF models in $\tau$-step ahead prediction (Fig. 5, 13-16). With low time-varying confounding ($\gamma < 3$), CLEF-CT with CDC loss and CLEF-CRN with GR loss perform comparably to or better than their non-CLEF counterparts. When time-varying confounding is high ($\gamma \geq 3$), CLEF-CT with CDC loss and CLEF-CRN with GR loss outperform their non-CLEF counterparts. For all levels of confounding bias, CLEF-CRN with CDC loss outperforms their non-CLEF counterparts. Notably, CLEF-CT and CLEF-CRN without any balancing loss (i.e., neither GR nor CDC; violet-red) are the best performing CT/CRN models. While studies have shown a trade-off between prediction accuracy and balanced representations (Huang et al., 2024; Wang et al., 2025), this finding is consistent with improved balanced representations and empirically supports Assumption 3.4. In other words, CLEF's strong performance without

any balancing loss suggests that the *temporal concepts learn balanced representations* that are not predictive of the assigned treatment and approximate the conditional mean function (Eq. 4; Sec. 3.3).

Beyond predictive accuracy metrics, we additionally evaluate the treatment predictability of CLEF and non-CLEF's learned representations (Tab. 10-12). Concretely, we compute the binary cross entropy (BCE) loss for predicting the treatment from the learned representations of CLEF and non-CLEF models. Because "balanced" representations should be treatment-invariant, higher BCE loss is better. For all three datasets of tumor growth and ICU trajectories, we find that CLEF models have comparable or higher (i.e., better) BCE loss in predicting the treatment compared to non-CLEF models. These results suggest that CLEF models learn representations that are treatment invariant.

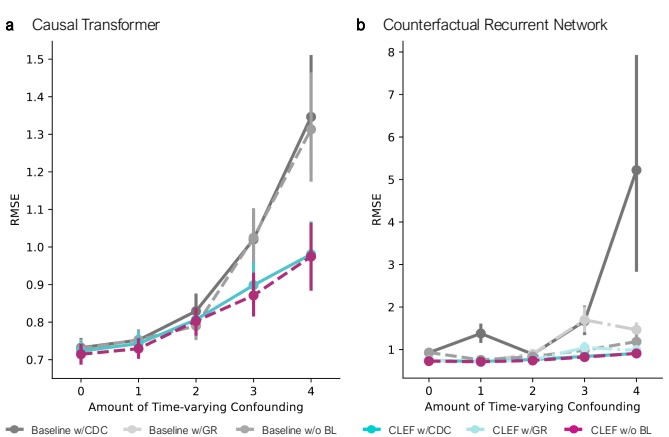

**Figure 5:** Counterfactual $\tau$-step ahead prediction on tumor growth (single-sliding treatment) with different amounts of time-varying confounding. Models are trained on 5 seeds; error bars show 95% CI.

## 5.5    R5: ZERO-SHOT CONDITIONAL GENERATION OF COUNTERFACTUAL TRAJECTORIES

We evaluate CLEF on zero-shot conditional generation of counterfactual cellular trajectories (Fig. 6, 17). With the WOT-CF dataset, models are trained on the "original" cellular trajectories and evaluated on the "counterfactual" cellular trajectories in a zero-shot setting.

By learning temporal concepts, CLEF *consistently outperforms* non-CLEF in immediate and delayed sequence editing (Fig. 17). We examine the predictions for

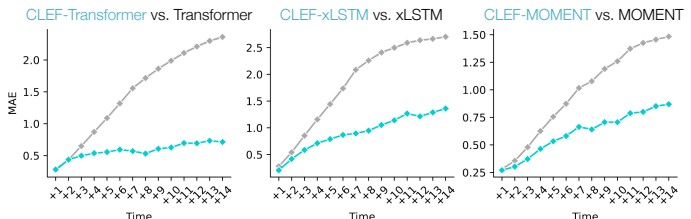

**Figure 6:** Zero-shot conditional generation of counterfactual cellular trajectories via delayed sequence editing. Shown are MAE of predictions per time step for counterfactual sequences of length 23 (most common sequence length) starting at time step 10 (earliest divergence time step of a counterfactual). Models are trained on 3 seeds; error bars show 95% CI.

trajectories of length 23, the most common sequence length in WOT-CF (Fig. 6). Since $t_i = 10$ is the earliest divergence time step, we input the first nine time steps $\mathbf{x}_{:,0:9}$, the counterfactual condition, and $t_j \in [10, 23]$. Comparing the generated and ground truth counterfactual sequences, we find that CLEF outperforms non-CLEF models after time step 10, which is when the trajectories begin to diverge.

## 5.6    R6: CASE STUDIES USING REAL-WORLD PATIENT DATASETS

Unlike conditional generation methods that rely on condition tokens to guide generation (Narasimhan et al., 2024; Jing et al., 2024; Zhang et al., 2023a), CLEF allows *direct edits to the generated outputs* via temporal concept intervention to produce counterfactual sequences (Sec. C.2.4). Instead of relying on predefined conditions, CLEF can precisely modify the values of specific lab tests to explore their longitudinal effects. We conduct case studies on two independent cohorts of patients with type 1 diabetes mellitus (T1D) (Quattrin et al., 2023) (Sec. C.2.3).

**Setup (Sec. C.2.4).** For each patient, we intervene on the temporal concepts corresponding to specific lab tests to simulate the "reversal" or "worsening" of symptoms, thereby generating "healthier" or "more severe" trajectories. CLEF-generated counterfactual patients are compared to the observed sequences from matched healthy individuals, other healthy individuals, and other T1D patients. We

hypothesize that clinically meaningful edits produce "healthier" (i.e., more similar to healthy patients) or "sicker" (i.e., more similar to other T1D patients) trajectories.

**Results.** First, we modify CLEF's concepts to halve glucose levels, aligning them closer to normal physiological ranges. Such counterfactual trajectories exhibit higher $R^2$ similarity with healthy individuals compared to other T1D patients (Fig. 7a), suggesting that CLEF *effectively generates trajectories indicative of a healthier state*. Next, we simulate a worsening condition by doubling glucose levels. These CLEF-generated trajectories show higher $R^2$ similarity with other T1D patients than with healthy individuals (Fig. 7a), as expected based on clinical evidence. Further, unlike CLEF, SimpleLinear (ablation) cannot generate trajectories that resemble the trajectories of healthier or sicker patients, depending on the intervention (Tab. 13-14).

Beyond examining the *direct effects* of the interventions on CLEF's concepts, we examine the *indirect changes* in CLEF-generated patients' lab values resulting from glucose modifications. In both eICU-T1D and MIMIC-IV-T1D cohorts, lowering glucose also leads to a reduction in white blood cell (WBC) count (Fig. 7b, 18a). This aligns with clinical knowledge, as T1D is an autoimmune disorder where immune activity, including WBC levels, plays a critical role (Quattrin et al., 2023). When we intervene on CLEF to reduce WBC levels instead of glucose, glucose levels also decrease across both cohorts (Fig. 18b,c), reinforcing the interdependence of these physiological markers. Finally, we show that modifying multiple lab tests simultaneously

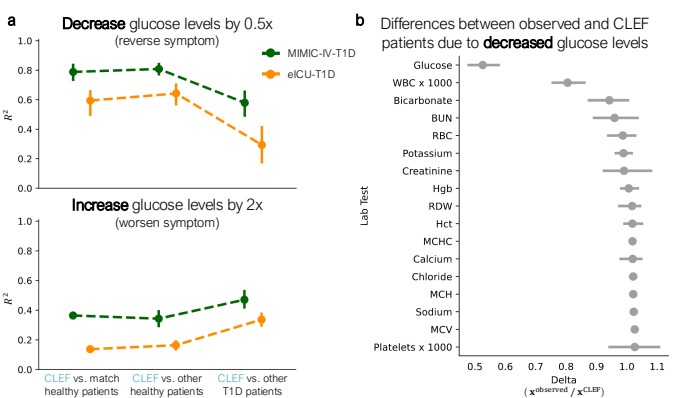

**Figure 7:** CLEF-generated counterfactual patients via intervention on temporal concepts. We intervene on CLEF to **(a)** halve or double a T1D patient's glucose levels to infer a "healthier" or "sicker" counterfactual patient. Higher $R^2$ indicates that patient pairs are more similar. **(b)** Observed and CLEF patients from the eICU-T1D cohort are compared to quantify the differences between their lab test trajectories as a result of the intervention to halve T1D patients' glucose levels. Delta $\left(x^{\text{observed}}/x^{\text{CLEF}}\right) = 1$ indicates no difference between the observed and CLEF patients. Error bars show 95% CI. Refer to Tables 13-14 for ablation analyses.

can produce compounding effects. When we intervene on CLEF to reduce both glucose and WBC levels, the resulting CLEF-generated patients resemble healthy individuals even more closely than other T1D patients, suggesting that CLEF can *capture the joint impact of multiple simultaneous edits on a patient* (Fig. 18d). Altogether, we demonstrate that direct temporal concept edits via CLEF enables actionable interpretability and tangible *in silico* hypothesis exploration.

## 6 CONCLUSION

CLEF is a flexible approach that *learns temporal concepts* for conditional sequence generation and potential outcomes prediction under specific conditions. Extensive experiments show that temporal concepts introduced in CLEF contribute to overall model performance. Controlling for model, time, and space complexity, temporal concepts generally yield *faster convergence* (Sec. D.5). Across 8 biological, medical, and financial datasets, CLEF excels in the conditional generation of longitudinal sequences, making *precise local edits while preserving global integrity*. CLEF also has *stronger generalizability* to new sequences than non-CLEF counterparts. Under counterfactual inference assumptions, CLEF *accurately estimates counterfactual outcomes over time*, outperforming baselines in settings with high time-varying confounding bias. CLEF even outperforms SOTA conditional generation models in *zero-shot counterfactual generation*. Further, we show that *interventions directly* on CLEF's temporal concepts can generate counterfactual patients such that their trajectories are shifted toward healthier outcomes. This capability has the potential to help discover clinical interventions that could alleviate a patient's symptoms. Limitations and future directions are discussed in Sec. F. We believe that CLEF's controllable sequence editing can help realize the promise of virtual cells and patients to facilitate large-scale *in silico* experimentation of molecules, cells, and tissues.

## ETHICS STATEMENT

By introducing a flexible and interpretable approach to conditional sequence generation, CLEF bridges the gap between language model-style conditional generation and structured, time-sensitive sequence editing, with implications for decision support in medical and scientific applications. Like all generative AI models, CLEF (and its derivatives) should be used solely for the benefit of society. In this study, we show that CLEF can generate alternative cellular trajectories and simulate the reversal or progression of symptoms to model healthier or sicker patient outcomes. However, this work (and any derivatives) should never be used to induce harmful cellular states (e.g., activating transcription factors to drive a cell toward a pathological state) or negatively impact patient care (e.g., neglecting necessary clinical interventions or recommending harmful treatments). Our goal is to help researchers understand the underlying mechanisms of disease to improve public health. Any misuse of this work poses risks to patient well-being. Therefore, the ability to intervene on CLEF's generated outputs should be leveraged to assess the model's robustness and correctness for ethical and responsible use.

## REPRODUCIBILITY STATEMENT

We provide code and instructions to implement CLEF, baselines, and ablations and to reproduce the experiments in this paper: `https://github.com/mims-harvard/CLEF`. In the Appendix, Sec. C provides details about data construction, data preparation, and experimental setup; and Sec. D describes the implementation and training of all models. We do not share data or model weights that may contain sensitive patient information.

## ACKNOWLEDGMENTS

M.L. and M.Z. are supported by the Berkowitz Family Living Laboratory at Harvard Medical School and the Clalit Research Institute. Y.E. is supported by grant T32 HG002295 from the National Human Genome Research Institute and the NDSEG Fellowship. We gratefully acknowledge the support of NIH R01-HD108794, NSF CAREER 2339524, US DoD FA8702-15-D-0001, ARPA-H BDF program, awards from Chan Zuckerberg Initiative, Bill & Melinda Gates Foundation INV-079038, Amazon Faculty Research, Google Research Scholar Program, AstraZeneca Research, Roche Alliance with Distinguished Scientists, Sanofi iDEA-iTECH, Pfizer Research, John and Virginia Kaneb Fellowship at Harvard Medical School, Biswas Computational Biology Initiative in partnership with the Milken Institute, Harvard Medical School Dean's Innovation Fund for the Use of Artificial Intelligence, Harvard Data Science Initiative, and Kempner Institute for the Study of Natural and Artificial Intelligence at Harvard University. Any opinions, findings, conclusions or recommendations expressed in this material are those of the authors and do not necessarily reflect the views of the funders. We thank Ruth Johnson, Boyang Fu, and Grey Kuling for their helpful feedback.

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

# APPENDIX

## A EXTENDED RELATED WORK

### A.1 LEVERAGING TRAJECTORIES AS INDUCTIVE BIASES

Understanding sequential data as trajectories (e.g., increasing, decreasing, constant) is more natural for human interpretation than individual values (Kacprzyk et al., 2024). Many models on temporal data extract dynamic motifs as inductive biases to improve their interpretability (Kacprzyk et al., 2024; Goswami et al., 2024; Cao et al., 2024). Such temporal patterns can be used for prompting large pretrained models to perform time series forecasting (Cao et al., 2024), suggesting that trajectories can capture more universal and transferrable insights about the temporal dynamics in time series data. Trajectories have yet to be adopted for conditional or counterfactual sequence generation.

**Relevance to CLEF.** Temporal concepts $\mathbf{c}$ (Def. 3.2) represent trajectories (or rates of change). The concept decoder $G$ leverages temporal concepts $\mathbf{c}$ and covariates at the latest time step $\mathbf{x}_{:,t_i}$ to generate the remainder of the sequence (Sec. 3.2). To understand how temporal concepts enable CLEF models to preserve global consistency: One can think of the latest covariates $\mathbf{x}_{:,t_i}$ as a set of reference values for each covariate, and these values are modified based on the desired forecast time $t_j$ and condition token $\mathbf{z}_s$. Such modifications are captured by temporal concepts $\mathbf{c}$, which represent the rates of change (or trajectories) for each covariate from time steps $t_i$ to $t_j$.

### A.2 BUILDING DIGITAL TWINS

Building virtual representations of cells and patients (commonly referred to as virtual cells, virtual patients, or digital twins) has the potential to facilitate preventative and personalized medicine (Li et al., 2025; Bunne et al., 2024). Medical digital twins (e.g., an artificial lung or pancreas, automated insulin delivery systems, and cardiac twins) have demonstrated clinical utility (Li et al., 2025; Kovatchev et al., 2025; Qian et al., 2025). There is a wide range of methods for building digital twins, such as mechanistic models (e.g., physics-informed self-supervised learning approach (Kuang et al., 2024)), neural models (e.g., finetuned large language and vision models (Makarov et al., 2025; Awasthi et al., 2025)), and hybrid models (e.g., framework with mechanistic and neural components (Holt et al., 2024)).

**Relevance to CLEF.** CLEF is a machine learning-based neural model. It is a flexible architecture to enable conditional sequence generation (Def. 3.1, Problem Statement 3.1) as well as counterfactual prediction (Sec. 3.3, Sec. D.4) of continuous multivariate sequences.

### A.3 ADDITIONAL DETAILS

**Controllable text generation.** Controllable text generation (CTG), designed specifically to edit natural language sequences, has been extensively studied (Zhang et al., 2023a). **They excel in *immediate sequence editing***: predicting the next token or readout in the sequence under a given condition (Niu et al., 2024; Gu et al., 2025; Zhou et al., 2024; Chatzi et al., 2025; Zhang et al., 2023a; Bhattacharjee et al., 2024). For example, if asked to predict the next word in the sentence "Once upon a time, there lived a boy" under the condition that the genre is horror, a CTG model may respond with "alone" to convey vulnerability and loneliness. However, **CTG models are unable to perform *delayed sequence editing***: modifying a trajectory in the far-future. The distinction is important: the focus is on when the edit occurs, not necessarily when its effects manifest. Whereas immediate sequence editing applies a condition *now* (e.g., administering insulin *today*), delayed sequence editing schedules a condition for the *future* (e.g., starting a chemotherapy regimen *in six weeks*). Existing CTG models cannot effectively utilize the given context to skip ahead to the future; instead, they would need to be run repeatedly to fill in the temporal gap without any guarantee of satisfying the desired condition. As a result, **CTG models are insufficient for other sequence types (i.e., not natural language) for which both immediate and delayed sequence editing are necessary**, such as cell development and patient health trajectories.

**Delayed sequence editing vs. long-horizon forecasting.** While long-horizon forecasting and delayed sequence editing both predict the sequence or covariates at a future time $t_j$, delayed sequence editing

does not require autoregressive predictions from $t_i$ to $t_j$, which can lead to accumulation of error. Instead, delayed sequence editing allows skipping directly to $t_j$ from $t_i$ in a single step.

**Intuition for CLEF's conditional sequence generation architecture.** We leverage the state-of-the-art conditional sequence generation setup (Waswani et al., 2017; Narasimhan et al., 2024; Jing et al., 2024; Zhang et al., 2023a; Beck et al., 2024; Valevski et al., 2025). The *sequence encoder* extracts features from historical covariates to learn a hidden representation that captures relevant information for the generation task. Any encoder (e.g., pretrained multivariate foundation model) can be used with CLEF. The *condition adapter* projects the condition token to a shared latent space with the sequence and time representations. Because condition tokens are generated by a pretrained foundation model (e.g., ESM2, a protein language model that learns on protein sequences), they capture information that allows CLEF to generalize to conditions that have not been observed in the training dataset (e.g., based on shared evolutionary information between protein sequences). The ***concept encoder*** (**CLEF only**) learns a representation (temporal concepts) that captures information about the condition at the next time step, historical conditions, and historical covariates. The final GELU activation layer transforms its input into values that are $\gtrsim 0$, which are interpreted by the concept decoder as the trajectories (or rates of change) of covariates between time steps $t_i$ and $t_j$. The ***concept decoder*** (**CLEF only**) generates a sequence by applying the learned temporal concepts to the covariates at the last time step $t_i$. Element-wise multiplication between the learned temporal concepts and the covariates at $t_i$ is a suitable operation because it is less sensitive to covariates with different units of measurements, which are commonly observed in clinical sequences (Sec. C.2, C.4). Further, applying the temporal concepts in a single step (via element-wise multiplication) allows users to directly intervene on the temporal concepts and simulate the effects of the intervention as a counterfactual trajectory (Sec. C.2.4). *With these components*, CLEF can generate sequences based on high-dimensional sequences at any future time point and condition. *Without the sequence encoder*, it would be computationally challenging to operate directly on the input historical sequences. *Without the condition adapter*, CLEF and state-of-the-art conditional generation models cannot generalize well to unseen conditions in the training dataset. *Without the concept encoder or decoder*, the model may inaccurately generate sequences (refer to Sections 5.1-5.3, and 5.5 for empirical results).

**Other usage of counterfactuals.** (1) There is extensive work on generating counterfactuals for static data (e.g., single time-step perturbation measured via gene expression profiles or images) (Louizos et al., 2017; Yoon et al., 2018; Lotfollahi et al., 2023; Wu et al., 2024; 2025). In this work, we focus on longitudinal trajectories. (2) Counterfactual prediction has been used as an additional task to improve the predictions' interpretability and accuracy (Yan & Wang, 2023; Hao et al., 2023; Wang et al., 2023; Liu et al., 2025), such as leveraging causal alignment to produce reliable diagnoses (Liu et al., 2025). While CLEF's temporal concepts can be intervened upon to interpret model outputs, counterfactual prediction is not an auxiliary task to improve CLEF's interpretability and performance.

**Excluded baselines for estimating counterfactual outcomes over time.** Causal CPC (El Bouchat-taoui et al., 2024), Mamba-CDSP (Wang et al., 2025), and GMCG (Ahn & Vashist, 2025) can estimate counterfactual outcomes over time, but are excluded due to unavailable code. While BNCDE (Hess et al., 2024) can estimate counterfactual outcomes over time, it is designed to forecast outcomes as well as uncertainty (rather than single-point estimates, which is the focus of Sec. 3.3). As such, extending BNCDE (Hess et al., 2024) with CLEF is not directly feasible. CF-GODE (Jiang et al., 2023) can estimate continuous-time counterfactual outcomes, but is excluded due to unavailable code.

**Distinction from biological sequence design.** Methods for biological sequence design (Stanton et al., 2022; M Ghari et al., 2024; Jain et al., 2023) are not comparable to CLEF because there is no temporal aspect in the data. The "sequence editing" by these models focuses on positional changes, not temporal. These methods are more related to text generation, where time is not a requirement.

**Distinction from reinforcement learning.** In contrast to reinforcement learning approaches (Oh et al., 2025), CLEF is trained via a fundamentally different objective. The objective of reinforcement learning is to learn a policy (i.e., output the optimal next action given the current/history of states) (Oh et al., 2025). On the other hand, CLEF is akin to a forward transition model (i.e., predict future state given history of states and actions/conditions). **CLEF is complementary to reinforcement learning.**

## B  ASSUMPTIONS FOR CAUSAL IDENTIFICATION

Under the potential outcomes framework (Neyman, 1923; Rubin, 1978) and its extension to time-varying treatments and outcomes (Robins & Hernan, 2008), the potential counterfactual outcomes over time (i.e., $\tau$-step ahead, where $\tau = t_j - t_i$, potential outcome conditioned on history from Eq. 4) are identifiable from factual observational data under three standard assumptions: consistency, positivity, and sequential ignorability.

**Assumption B.1** (Consistency). Let $\mathbf{s}$ be the given sequence of treatments for a patient, consisting of historical treatments $\mathbf{s}_{t_0:t_i}$ and next treatment $\mathbf{s}_{t_j}$. The potential outcome is consistent with the observed (factual) outcome $\mathbf{x}_{:,t_j}(\mathbf{s}) = \mathbf{x}_{:,t_j}$.

**Assumption B.2** (Positivity). There is always a non-zero probability of receiving (or not) a treatment for all the history space over time (Imai & Van Dyk, 2004): If $P(\mathbf{s}_{t_0:t_i}, \mathbf{x}_{:,t_0:t_i}) > 0$, then $0 < P(\mathbf{s}_{t_j}|\mathbf{s}_{t_0:t_i}, \mathbf{x}_{:,t_0:t_i}) < 1$ for all $\mathbf{s}_{t_0:t_i}$. This assumption is also referred to as (sequential) overlap (Bica et al., 2020; Melnychuk et al., 2022).

**Assumption B.3** (Sequential ignorability). The current treatment is independent of the potential outcome, conditioning on the observed history: $\mathbf{s}_{t_j} \perp\!\!\!\perp \mathbf{x}_{:,t_j}(\mathbf{s}_{t_j})|\mathbf{s}_{t_0:t_i}, \mathbf{x}_{:,t_0:t_i}$. This implies that there are no unobserved confounders that affect both treatment and outcome.

While Assumptions B.2 and B.3 are standard across all methods that estimate treatment effects, they may not always be satisfied in real-world settings (Robins et al., 2000; Pearl, 2009; Ying et al., 2025).

**Corollary B.4** (G-computation). *Assumptions B.1-B.3 provide sufficient identifiability conditions for Eq. 4 (i.e., with G-computation (Li et al., 2021)). However, it requires estimating conditional distributions of time-varying covariates (Melnychuk et al., 2022). Since this could be challenging given a finite dataset size and high dimensionality of covariates, we refrain from the explicit usage of G-computation (Melnychuk et al., 2022).*

Note that the standard setup for counterfactual prediction assumes a fixed time grid and normalized covariates (Bica et al., 2020; Melnychuk et al., 2022). As such, the standardized data preprocessing pipeline entails forward and backward filling for missing values and standard normalization of continuous time-varying features (Bica et al., 2020; Melnychuk et al., 2022). With the model architecture shown in Fig. 2, these preprocessing steps are not necessary, thereby better reflecting real-world data. Still, Assumptions B.1-B.3 hold for our models depicted in Fig. 2.

## C  DATA & EXPERIMENTAL SETUP

We provide further details about data construction, data preparation, and experimental setup. Sections C.1-C.2 and Tab. 1 describe the novel conditional sequence generation benchmark datasets. Sections C.3-C.4 discuss the standard synthetic and semi-synthetic benchmark datasets for counterfactual outcomes estimation. Each section also contains the corresponding experimental setup. We share code and instructions in our GitHub repository to reproduce the experiments in this paper: `https://github.com/mims-harvard/CLEF`.

**Overview of novel datasets.** To study cellular development, fibroblast cells derived from mice can be artificially reprogrammed into various other cell states *in vitro*. A cell's state is defined by its gene expression. Throughout reprogramming, a cell activates transcription factor (TF) genes at different time points to change its gene expression, thereby influencing its developmental trajectory. In Fig. 3a, a mouse fibroblast is being reprogrammed over the span of 20 days (D0-D20); color and shape represent cell state. On day 8, if the cell activates the Obox6 TF, the cell is on the path toward becoming an induced pluripotent stem cell (iPSC); whereas if it activates the Neurod4 TF, it is on the path toward becoming a neuron or astrocyte. The health of a human patient is often monitored through lab tests (e.g. blood sodium level, white blood cell count). As shown in Fig. 3b, the history of lab results across multiple patient visits (V1-V9) as well as candidate clinical interventions (e.g., medication) can be used to infer the most likely future trajectory of the patient's health.

### C.1  CELLULAR DEVELOPMENTAL TRAJECTORIES

Here, we describe the process of (1) simulating single-cell transcriptomic profiles of developmental time courses for individual cells and (2) preparing these trajectories for modeling.

**Table 1: Dataset statistics for conditional sequence generation benchmarks.** We construct three core datasets for benchmarking conditional sequence generation: WOT (cellular developmental trajectories), eICU (patient lab tests), and MIMIC-IV (patient lab tests). We also construct a paired counterfactual cellular trajectories dataset, WOT-CF. $N$ is the number of sequences (i.e., cellular developmental trajectories, patient lab test trajectories), $V$ is the number of measured variables (i.e., gene expression, lab test), and $L$ is the length of the sequences.

| Dataset | $N$ | $V$ | Mean $L$ | Max $L$ |
|---------|-----|-----|----------|---------|
| WOT | $3,000$ | $1,480$ | $27.03 \pm 6.04$ | $37$ |
| WOT-CF | $2,560$ | $1,480$ | $27.31 \pm 6.11$ | $37$ |
| eICU | $108,346$ | $17$ | $20.27 \pm 25.23$ | $858$ |
| MIMIC-IV | $156,310$ | $16$ | $15.56 \pm 24.43$ | $949$ |

### C.1.1 SIMULATING TRAJECTORIES

Cellular reprogramming experiments help elucidate cellular development (Schiebinger et al., 2019). In these wet-lab experiments, cells are manipulated and allowed to progress for a specific period of time before they undergo RNA sequencing (RNA-seq), and we analyze the resulting RNA-seq data to observe their new cellular profiles (Schiebinger et al., 2019). RNA-seq is a destructive process for the cell, meaning that the same cell cannot be sequenced at two different time points. Computational models are thus necessary to infer the trajectory of a cell.

**Waddington-OT dataset and model.** Waddington-OT (Schiebinger et al., 2019) is a popular approach to reconstruct the landscape of cellular reprogramming using optimal transport (OT). There are two components in Waddington-OT: (1) *a single-cell RNA-seq (scRNA-seq) dataset* of mouse cells from a reprogramming experiment, and (2) *an OT-based trajectory inference model* fitted on the scRNA-seq dataset. The scRNA-seq dataset consists of 251,203 mouse cells profiled from 37 time points (0.5-day intervals) during an 18-day reprogramming experiment starting from mouse embryonic fibroblasts. The trajectory inference model consists of transport matrices $\pi_{t_k, t_{k+1}}$ with dimensions $N \times M$ that relate all cells $\mathbf{x}_{t_k}^1, ..., \mathbf{x}_{t_k}^n$ profiled at time $t_k$ to all cells $\mathbf{x}_{t_{k+1}}^1, ..., \mathbf{x}_{t_{k+1}}^m$ profiled at time $t_{k+1}$. An entry at row $i$ and column $j$ of $\pi_{t_k, t_{k+1}}$ corresponds to the probability that $\mathbf{x}_{t_{k+1}}^j$ is a descendant cell of $\mathbf{x}_{t_k}^i$, as determined using optimal transport (Chizat et al., 2018). Every cell in the scRNA-seq dataset is either pre-labeled as one of the 13 provided cell sets (i.e., induced pluripotent stem, stromal, epithelial, mesenchymal-epithelial transition, trophoblast, spongiotrophoblast, trophoblast progenitor, oligodendrocyte progenitor, neuron, radial glial, spiral artery trophoblast giant, astrocyte, other neural) or unlabeled. We cluster the unlabeled cells using Leiden clustering via `scanpy` (Wolf et al., 2018) at a resolution of 1, and define the resulting 27 unlabeled clusters as unique cell sets. As a result, each cell in the dataset belongs one and only one cell set.

**Simulating cell state trajectories.** We define "cell state" as the transcriptomic profile of a cell. Here, a transcriptomic profile is the log-normalized RNA-seq counts of the top $1,479$ most highly variable genes. To create a simulated trajectory of cell states for an individual cell undergoing reprogramming, we randomly and uniformly sample a cell profiled at time step $t_0$ (Day 0.0) from the Waddington-OT scRNA-seq dataset, and generate via the transport matrix $\pi_{t_0, t_1}$ a probability distribution $\mathbb{P}_{t_1}$ over possible descendant cells $\mathbf{x}_{t_1}^1, ..., \mathbf{x}_{t_1}^m$ at time step $t_1$ (Day 0.5). We sample a cell from this distribution, and repeat the process until we reach either Day 18.0 or a terminal state (i.e., neural, stromal, or induced pluripotent stem cell). After generating a trajectory composed of cells from the Waddington-OT scRNA-seq dataset through this process, we retrieve the transcriptomic profile of each cell to compose $\mathbf{x}_{:, t_0:t_T}$, where $T$ is the length of the trajectory.

**Inferring conditions.** A condition $s_{t_i}$ is defined as the activation of a transcription factor (TF) that leads a cell to transition from state $\mathbf{x}_{t_i}$ to descendant state $\mathbf{x}_{t_{i+1}}$. To infer such conditions, we perform differential expression analysis between cells from the same cell set as $\mathbf{x}_{t_i}$ (i.e., $\mathbf{x}^a \in A$) and cells from the same cell set as $\mathbf{x}_{t_{i+1}}$ (i.e., $\mathbf{x}^b \in B$). Using the `wot.tmap.diff_exp` function (via the `Waddington-OT` library), we identify the top TF that was significantly upregulated in $\mathbf{x}^a \in A$ compared to $\mathbf{x}^b \in B$. If no TFs are differentially expressed, then the condition is "None." We retroactively perform this analysis on all pairs of consecutive cell states in a cell state trajectory $\mathbf{x}_{:, t_0:t_T}$ to obtain the full trajectory containing both cell states and TF conditions: $\tau = \{\mathbf{x}_{t_0}, s_{t_0}, \mathbf{x}_{t_1}, s_{t_1}, \cdots, s_{t_{T-1}}, \mathbf{x}_{t_T}\}$. In other words, $\tau$ represents a simulated trajectory of an

individual cell undergoing the reprogramming process. Condition embeddings $\mathbf{z}_s \in \mathbb{R}^{5120}$ are obtained from the (frozen) pretrained ESM-2 embedding model (Lin et al., 2023).

**Generating matched counterfactual trajectories.** We additionally create pairs of matched counterfactual trajectories to evaluate a model's performance in zero-shot counterfactual generation. Each pair consists of an "original" trajectory $\tau_{og}$ and a "counterfactual" trajectory $\tau_{cf}$. First, we generate $\tau_{og}$ using the Waddington-OT model. Then, given a divergence time step $D$, the first $D$ time steps of $\tau_{og}$ are carried over to $\tau_{cf}$ such that the first $D$ cell states and conditions of $\tau_{og}$ and $\tau_{cf}$ are exactly the same. The remaining states and conditions of $\tau_{cf}$ are sampled independently from $\tau_{og}$, resulting in an alternative future trajectory based on an alternative condition at time step $D$. The paired trajectories are simulated such that the alternative condition is different from the original, and the alternative condition leads to a different cell set at $D + 1$. All trajectories are unique.

**Implementation note:** Since CLEF learns time embeddings based on the year, month, date, and hour of a timestamp, we convert the time steps of each cell into timestamps. We set the starting time $t_0$ as timestamp `2000/01/01 00:00:00`, and add $10 \times t_i$ hours to the converted timestamp of $t_{i-i}$.

### C.1.2 EXPERIMENTAL SETUP

**Generating data splits.** There are three cell sets (i.e., groups of cells with the same cell state label) that consist of cells from Day 0.0 in our post-clustering version of the Waddington-OT dataset. We refer to these cell sets as "start clusters" because all initial cell states are sampled from one of these cell sets. Since the choice of start cluster can influence the likelihood of a cell's trajectory reaching certain terminal fates, we split our cellular trajectories into train, validation, and test sets based on their start cluster. This cell-centric data split allows us to evaluate how well a model can generalize to different distributions of trajectories. Start cluster #1 is in the train set, start cluster #3 is in the validation set, and start cluster #2 is in the test set.

**Zero-shot counterfactual generation.** The data split for zero-shot counterfactual generation is constructed such that the original trajectories $\tau_{og}$ are in the train or validation sets, and the counterfactual trajectories $\tau_{cf}$ are in the test set.

## C.2 PATIENT LAB TESTS

Here, we describe the process of (1) preprocessing electronic health records to extract longitudinal routine lab tests data and (2) preparing these trajectories for modeling.

### C.2.1 CONSTRUCTING ROUTINE LAB TEST TRAJECTORIES

We leverage two publicly available medical datasets: eICU (Pollard et al., 2018) and MIMIC-IV (Johnson et al., 2024a; 2023; Goldberger et al., 2000). Both datasets are under the PhysioNet Credentialed Health Data License 1.5.0 (PhysioNet). The retrieval process includes registering as a credentialed user on PhysioNet, completing the CITI "Data or Specimens Only Research" training, and signing the necessary data use agreements.

We process each dataset (i.e., eICU, MIMIC-IV) separately with the following steps. First, we extract the routine lab tests only (annotation available only in MIMIC-IV) and the most commonly ordered lab tests (i.e., lab tests that appear in at least 80% of patients). Next, we keep patients for whom we have at least one of each lab test. If there are multiple measurements of a lab test at the same time step (i.e., year, month, date, hour, minute, and seconds), we take the mean of its values. We extract patients with more than one visit (or time step).

We define patients' conditions as medical codes, specifically International Classification of Diseases (ICD), of their diagnosis. Both eICU and MIMIC-IV use ICD-9 and ICD-10 codes. We extract the medical codes and their timestamps (multiple medical codes at a single time step is possible). Since the timestamps of diagnostic codes and lab tests are not necessarily the same (and there are fewer entries of diagnostic codes than lab orders), we merge them with a tolerance range of 12 hours (eICU) or two days (MIMIC-IV). We obtain (frozen) condition embeddings $\mathbf{z}_s \in \mathbb{R}^{128}$ (retrieved on December 22, 2024) from an embedding model that has been pretrained on a clinical knowledge graph (Johnson et al., 2024b). The clinical knowledge graph is constructed by integrating six existing databases of clinical vocabularies used in electronic health records: International Classification of

Diseases (ICD), Anatomical Therapeutic Chemical (ATC) Classification, Systemized Nomenclature of Medicine - Clinical Terms (SNOMED CT), Current Procedural Terminology (CPT), Logical Observation Identifiers Names and Codes (LOINC), and phecodes (Johnson et al., 2024b).

### C.2.2 GENERATING DATA SPLITS

We generate a standard patient-centric random split for benchmarking model performance (**R1-R2**), and a series of increasingly challenging data splits via SPECTRA (Ektefaie et al., 2024) to evaluate model generalizability (**R3**). We describe in detail the process of constructing SPECTRA data splits.

SPECTRA (Ektefaie et al., 2024) creates a series of splits with decreasing cross-split overlap or similarity between the train and test sets. By training and testing models on these splits, we can assess model performance as a function of cross-split overlap (Fig. 10). SPECTRA refers to this relationship as the spectral performance curve, which provides insight into how well a model generalizes to less similar data. When a new dataset split is encountered, it can be plotted as a point on this curve. The area under the spectral performance curve (AUSPC) serves as a metric of model generalizability and enables comparisons across models (Tab. 6).

To generate a split with SPECTRA, a similarity definition and a SPECTRA parameter (SP) value between 0 and 1 are required. SP controls the level of cross-split overlap (Fig. 10): values closer to 0 create splits resembling classical random splits, while values closer to 1 produce stricter splits with minimal or no overlap between train and test sets. For example, at an input of 1, no similar samples are shared between the train and test sets.

For eICU and MIMIC-IV, we define two patients as similar if: (1) they are of the same gender, (2) they are born in the same decade, and (3) they share at least one ICD-9 or ICD-10 category. We exclude ICD-9 and ICD-10 codes that are present in more than 50% of patients to avoid overly generic features. SPECTRA systematically prunes similar patients to produce splits. For this study, we generate 20 splits with SP values that are evenly spaced between 0 and 1 (Fig. 10). Given a train and test set, cross-split overlap is defined as the proportion of samples in the train set that are similar to at least one sample in the test set (Fig. 10).

### C.2.3 CONSTRUCTING COHORTS OF PATIENTS WITH TYPE 1 DIABETES MELLITUS

We conduct case studies on two independent cohorts of patients with type 1 diabetes mellitus (T1D), a chronic autoimmune disease in which the immune system attacks insulin-producing cells in the pancreas (Quattrin et al., 2023). From our processed eICU and MIMIC-IV datasets, we construct two cohorts of T1D patients and matched healthy individuals.

**Procedure.** To define a type 1 diabetes mellitus (T1D) patient cohort in eICU and MIMIC-IV, we identify patients with T1D and matched healthy individuals. A patient has T1D if the ICD-10 code `E10` (or the equivalent ICD-9 code `250`) is present in the electronic health records. Matched healthy patients are defined by three criteria. First, the patient must not contain any of the following ICD-10 (and ICD-9 equivalent) codes: `E11`, `E13`, `E12`, `E08`, `E09`, `R73`, and `O24`. An initial healthy patient cohort is constructed using these filtering codes. Next, we identify frequently co-occurring ICD codes between the initial set of patients and patients with T1D to filter out generic ICD codes (threshold = 20). Finally, healthy patients are matched with a T1D patient if: they are of the same gender, they are born in the same decade, and they share at least 50% of ICD codes.

**Data statistics.** eICU-T1D contains 59 T1D patients and 579 matched healthy controls, while MIMIC-IV-T1D includes 25 T1D patients and 226 matched healthy controls.

### C.2.4 EXPERIMENTAL SETUP FOR TYPE 1 DIABETES MELLITUS CASE STUDY

We evaluate CLEF's ability to simulate counterfactual patient trajectories through temporal concept intervention. This is analogous to intervening on concept bottleneck models by editing concept values and propagating the changes to the final prediction (Koh et al., 2020). This capability is particularly useful when condition tokens are insufficient (e.g., prescribing medication dosage). Editing concept values allow users (e.g., clinicians) to simulate potential trajectories as a result of the precise edits.

We conduct case studies on two independent patient cohorts with type 1 diabetes mellitus (T1D). For each patient, we intervene on the temporal concepts corresponding to specific lab tests to simulate the

"reversal" or "worsening" of symptoms, thereby generating "healthier" or "more severe" trajectories. Formally, given temporal concept $\mathbf{c}$ learned from $\mathbf{x}_{:,t_0:t_i}$ and an optional condition $s$, we modify $\mathbf{c}^I \neq \mathbf{c}$ such that at least one element satisfies $\mathbf{c}_k \neq \mathbf{c}_k^I$. Then, CLEF simulates future trajectories of length $T = 10$. We then compare these counterfactual trajectories (i.e., CLEF-generated patients) against observed sequences from matched healthy individuals, other healthy individuals, and other T1D patients. Our hypothesis is that clinically meaningful edits will produce "healthier" (i.e., more similar to healthy patients) or "sicker" (i.e., more similar to other T1D patients) trajectories.

## C.3 SYNTHETIC TUMOR GROWTH TRAJECTORIES

The tumor growth simulation model (Geng et al., 2017) produces trajectories of tumor volume (i.e., one-dimensional outcome) after cancer diagnosis. There are two binary treatments (i.e., radiotherapy $\mathbf{A}_t^r$ and chemotherapy $\mathbf{A}_t^c$) at time $t$, and the possible treatments are: $\{(\mathbf{A}_t^c = 0, \mathbf{A}_t^r = 0), (\mathbf{A}_t^c = 1, \mathbf{A}_t^r = 0), (\mathbf{A}_t^c = 0, \mathbf{A}_t^r = 1), (\mathbf{A}_t^c = 1, \mathbf{A}_t^r = 1)\}$. For $\tau$-step ahead prediction, we simulate synthetic tumor growth trajectories under single-sliding treatment (i.e., shift the treatment over a window) (Bica et al., 2020; Melnychuk et al., 2022) and random trajectories (i.e., randomly assign treatments) settings (Melnychuk et al., 2022). Importantly, the ground-truth counterfactual trajectories are known. We limit the length of trajectories to a maximum of 60 time steps. For each setting, we generate trajectories with different amounts of time-varying confounding $\gamma \in [0, 1, 2, 3, 4]$, each with 10,000 trajectories for training, 1,000 for validation, and 1,000 for testing.

We follow the data simulation process and experimental setup as described in Appendix J and GitHub repository of the original Causal Transformer publication (Melnychuk et al., 2022).

## C.4 SEMI-SYNTHETIC PATIENT TRAJECTORIES

MIMIC-III-CF is a semi-synthetic dataset based on patient data from real-world intensive care units (Wang et al., 2020; Johnson et al., 2016). The data are aggregated at hourly levels, with forward and backward filling for missing values and standard normalization of the continuous time-varying features (Wang et al., 2020; Johnson et al., 2016; Bica et al., 2020; Melnychuk et al., 2022). Patients have 25 vital signs as time-varying covariates and 3 static covariates (gender, ethnicity, age). Untreated trajectories of outcomes are first simulated under endogenous and exogenous dependencies, and then treatments are sequentially applied (Melnychuk et al., 2022). There are 3 synthetic binary treatments and 2 synthetic outcomes (Melnychuk et al., 2022). Importantly, the ground-truth counterfactual trajectories are known. We limit the length of trajectories to a maximum of 60 time steps. We generate 1,000 patients into train, validation, and test subsets via a 60%, 20%, and 20% split. For $\tau$-step ahead prediction with $\tau_{\max} = 10$, we sample 10 random trajectories for each patient per time step.

We follow the data simulation process and experimental setup as described in Appendix K and GitHub repository of the original Causal Transformer publication (Melnychuk et al., 2022).

## C.5 SALES TRAJECTORIES

Sales trajectories (M5 Forecasting) are obtained from daily transaction data of Walmart stores across three US states (Makridakis et al., 2022). Following (Huang et al., 2024) and (Wang et al., 2025), the objective of the model is to predict the future unit sales, and the condition is defined by the product price (Huang et al., 2024; Wang et al., 2025). There are 1,942 time points on 3,049 products from 10 stores (4 in California, 3 in Texas, and 3 in Wisconsin). The dataset is split by state: train, validate, and test on California, Texas, and Wisconsin, respectively. As the dataset does not contain any ground truth counterfactual trajectories, M5 is only used for conditional generation of observed sequences.

## D IMPLEMENTATION DETAILS

We provide code and instructions to implement CLEF, baselines, and ablations: `https://github.com/mims-harvard/CLEF`. To implement baselines, we follow the authors' recommendations on model design and hyperparameter selection from the original publications. We do not share data or model weights that may contain sensitive patient information.

## D.1 OBJECTIVE FUNCTIONS

For conditional generation, we use Huber loss, where $\mathbf{a} = \mathbf{x}^s_{:,t_j} - \hat{\mathbf{x}}^s_{:,t_j}$ and $\delta = 1$,

$$\mathcal{L}(\mathbf{x}^s_{:,t_j}, \hat{\mathbf{x}}^s_{:,t_j}) = \begin{cases} 0.5\mathbf{a}^2, & \text{if } |\mathbf{a}| \leq \delta \\ \delta(|\mathbf{a}| - 0.5\delta), & \text{otherwise} \end{cases} \tag{5}$$

We use PyTorch's implementation. To briefly explain each component of Huber loss:

- $\delta$ is used to switch between mean squared error (MSE) and mean absolute error (MAE).
- The $0.5\mathbf{a}^2$ term (MSE) is a quadratic component that penalizes outliers when errors are $\leq \delta$.
- The $\delta(|\mathbf{a}| - 0.5\delta)$ term (MAE) is a linear component that does not over-penalize large errors when errors are larger than $\delta$.

We also train CLEF via another commonly used objective function for forecasting (i.e., MSE loss) and objective functions designed specifically for counterfactual prediction (i.e., gradient reversal, counterfactual domain confusion; Sec. D.4) (Bica et al., 2020; Melnychuk et al., 2022).

## D.2 EXPERIMENTS TO EVALUATE TEMPORAL CONCEPTS

We intentionally design temporal concepts to isolate their contribution to predictive performance. In our formulation, temporal concepts are learned from an aggregation of historical data, the future time point, and the desired condition by concept encoder $E$ (Eq. 2). We evaluate multiple ways of defining $E$: via the state-of-the-art setups for conditional sequence generation (Sec. D.3) and counterfactual outcomes estimation (Sec. D.4) with different sequential encoders. Also, temporal concepts are applied directly to the latest time step in the concept decoder $F$ (Eq. 3) to generate the future state. Alternative model architectural designs for learning temporal concepts and applying conditions (or interventions) to the model may obfuscate the contribution of temporal concepts to predictive performance. For example, feeding the intervention directly to the decoder would bypass the temporal concept mechanism, meaning that the concepts would capture only the passage of time rather than the effect of the intervention. Similarly, feeding the future state directly to the decoder would introduce an additional function applied after the temporal concepts, making it difficult to directly control the edit by the specified concept (because of the add-on decoder).

To further isolate the contribution of temporal concepts to predictive performance, CLEF and non-CLEF models differ only in the components needed to learn temporal concepts (i.e., concept encoder and decoder). In other words, CLEF models share the same sequence encoder and condition adapter as their non-CLEF counterparts but replace the forecasting decoder in SOTA baselines (Sec. A.3, D.3-D.4) with a concept encoder and decoder to leverage temporal concepts.

Altogether, our formulation keeps the architecture minimal and interpretable, **ensuring that performance gains can be attributed directly to temporal concepts**.

## D.3 BASELINES FOR CONDITIONAL SEQUENCE GENERATION

We benchmark CLEF against the state-of-the-art conditional sequence generation setup with 3 sequential data encoders (Fig. 2): Transformer (Waswani et al., 2017; Narasimhan et al., 2024; Jing et al., 2024; Zhang et al., 2023a); xLSTM (Beck et al., 2024); and state-of-the-art time series foundation model, MOMENT (Goswami et al., 2024). For MOMENT, we finetune an adapter for the 1024-dimensional embeddings from the frozen `MOMENT-1-large` embedding model.

## D.4 CLEF EXTENSIONS AND BASELINES FOR COUNTERFACTUAL OUTCOMES ESTIMATION

Due to its versatility, CLEF can be leveraged by state-of-the-art machine learning models designed to estimate counterfactual outcomes (Bica et al., 2020; Melnychuk et al., 2022). Counterfactual Recurrent Network (CRN) (Bica et al., 2020) and Causal Transformer (CT) (Melnychuk et al., 2022) demonstrate state-of-the-art performance in the established benchmarks (Bica et al., 2020; Melnychuk et al., 2022). To implement CLEF-CRN and CLEF-CT, the GELU activation layer from the concept encoder $E$ (Eq. 2) and the concept decoder $G$ (Eq. 3) of CLEF are appended to outcome

predictor network (denoted as $G_Y$ where $Y$ is the outcome of the given treatment in the original publications (Bica et al., 2020; Melnychuk et al., 2022)) of CRN and CT. Following the original CRN and CT publications, we minimize the factual outcome loss (i.e., output of $G_Y$) via mean squared error (MSE) (Bica et al., 2020; Melnychuk et al., 2022).

We evaluate CLEF against their non-CLEF counterparts with and without balancing loss functions (i.e., gradient reversal (GR) (Bica et al., 2020), counterfactual domain confusion (CDC) loss (Melnychuk et al., 2022)). This results in 5 distinct state-of-the-art baselines: CRN with GR loss (i.e., original CRN implementation) (Bica et al., 2020); CRN with CDC loss (Melnychuk et al., 2022); CRN without balancing loss (Melnychuk et al., 2022); CT with CDC loss (i.e., original CT implementation) (Melnychuk et al., 2022); and CT without balancing loss (Melnychuk et al., 2022).

### D.5 MODEL TRAINING

CLEF models do not require any additional resources than non-CLEF models. All CLEF models have **comparable number of parameters** (Tab. 2) and **time complexity** (Tab. 3) as their non-CLEF counterparts. In 67% of cases, the CLEF model's best checkpoint occurs earlier than its non-CLEF counterpart, indicating **faster convergence** (Tab. 4).

Models are **trained on a single GPU** (i.e., NVIDIA A100 or H100). For the model with the largest number of parameters (i.e. CLEF-MOMENT with FFN=1 on the M5 dataset; Tab. 2), 24GB of GPU memory is allocated and the maximum utilization is 100%. For the model with the largest number of trainable parameters (i.e., CLEF-xLSTM with FFN=1 on the M5 dataset; Tab. 2), 12GB of memory is allocated and the maximum utilization is 86%.

**Table 2: Model parameters.** The number of all (denoted as **A**) or trainable (denoted as **T**) parameters is comparable between CLEF and non-CLEF counterparts. FFN refers to the optional FFN layer in the concept encoder; the number of layers $l_{\text{FFN}} \in [0, 1]$ is a hyperparameter.

| Dataset | Encoder | Baseline (A) | Baseline (T) | CLEF (A) | CLEF (T) |
|---|---|---|---|---|---|
| WOT | Transformer | 67002560 | 66947800 | FFN=0: 67008480
FFN=1: 69200360 | FFN=0: 66953720
FFN=1: 69145600 |
| WOT | xLSTM | 66170384 | 66115624 | FFN=0: 66176304
FFN=1: 68368184 | FFN=0: 66121544
FFN=1: 68313424 |
| WOT | MOMENT | 354933280 | 13638200 | FFN=0: 354939200
FFN=1: 357131080 | FFN=0: 13644120
FFN=1: 15836000 |
| eICU | Transformer | 52560 | 37116 | FFN=0: 52632
FFN=1: 52974 | FFN=0: 37188
FFN=1: 37530 |
| eICU | xLSTM | 78716 | 63272 | FFN=0: 78788
FFN=1: 79130 | FFN=0: 63344
FFN=1: 63686 |
| eICU | MOMENT | 341293924 | 38160 | FFN=0: 341293996
FFN=1: 341294338 | FFN=0: 38232
FFN=1: 38574 |
| MIMIC | Transformer | 48000 | 32816 | FFN=0: 48064
FFN=1: 48336 | FFN=0: 32880
FFN=1: 33152 |
| MIMIC | xLSTM | 79096 | 63912 | FFN=0: 79160
FFN=1: 79432 | FFN=0: 63976
FFN=1: 64248 |
| MIMIC | MOMENT | 341290816 | 35312 | FFN=0: 341290880
FFN=1: 341291152 | FFN=0: 35376
FFN=1: 35648 |
| M5 | Transformer | 260954950 | 260890900 | FFN=0: 260967150
FFN=1: 270272700 | FFN=0: 260903100
FFN=1: 270208650 |
| M5 | xLSTM | 280615102 | 280551052 | FFN=0: 280627302
FFN=1: 289932852 | FFN=0: 280563252
FFN=1: 289868802 |
| M5 | MOMENT | 372621770 | 31317400 | FFN=0: 372633970
FFN=1: 381939520 | FFN=0: 31329600
FFN=1: 40635150 |

**Table 3: Time complexity.** The training and evaluation times are comparable for the CLEF and non-CLEF models. Reported times are rounded to the nearest half hour.

| Dataset | Encoder | Baseline | CLEF |
|---------|---------|----------|------|
| WOT | Transformer | 1.0 hour | 1.0 hour |
| WOT | xLSTM | 1.0 hour | 1.0 hour |
| WOT | MOMENT | 4.0 hours | 4.0 hours |
| eICU | Transformer | 5.0 hours | 5.0 hours |
| eICU | xLSTM | 5.0 hours | 5.0 hours |
| eICU | MOMENT | 5.5 hours | 5.5 hours |
| MIMIC | Transformer | 7.0 hours | 7.0 hours |
| MIMIC | xLSTM | 7.5 hours | 7.5 hours |
| MIMIC | MOMENT | 7.5 hours | 7.5 hours |
| M5 | Transformer | 0.5 hour | 0.5 hour |
| M5 | xLSTM | 0.5 hour | 0.5 hour |
| M5 | MOMENT | 1 hour | 1 hour |

**Table 4: Model convergence.** Shown are the epochs of the best model checkpoints (index starting at 1) for each dataset. FFN refers to the optional FFN layer in the concept encoder; the number of layers $l_{FFN} \in [0, 1]$ is a hyperparameter. In 67% of cases, the CLEF model's best checkpoint occurs earlier than its non-CLEF counterpart, indicating faster convergence.

| Dataset | Encoder | Baseline | CLEF |
|---------|---------|----------|------|
| WOT | Transformer | 49 | FFN=0: 46
FFN=1: 38 |
| | xLSTM | 44 | FFN=0: 28
FFN=1: 20 |
| | MOMENT | 5 | FFN=0: 5
FFN=1: 5 |
| eICU | Transformer | 47 | FFN=0: 48
FFN=1: 45 |
| | xLSTM | 45 | FFN=0: 44
FFN=1: 44 |
| | MOMENT | 4 | FFN=0: 3
FFN=1: 2 |
| MIMIC | Transformer | 48 | FFN=0: 42
FFN=1: 42 |
| | xLSTM | 42 | FFN=0: 42
FFN=1: 48 |
| | MOMENT | 5 | FFN=0: 3
FFN=1: 1 |
| M5 | Transformer | 46 | FFN=0: 46
FFN=1: 46 |
| | xLSTM | 50 | FFN=0: 35
FFN=1: 7 |
| | MOMENT | 5 | FFN=0: 4
FFN=1: 5 |

## D.6 Hyperparameter Sweep

The selection of hyperparameters for the (conditional sequence generation) models trained from scratch are: dropout rate $\in [0.3, 0.4, 0.5, 0.6]$, learning rate $\in [0.001, 0.0001, 0.00001]$, and number of layers (or blocks in xLSTM) $\in [4, 8]$. Because the number of heads must be divisible by the number of features, the number of heads for eICU (18 lab tests) $\in [2, 3, 6, 9]$ and for others $\in [4, 8]$. For xLSTM, the additional hyperparameters are: 1D-convolution kernel size $\in [4, 5, 6]$ and QVK projection layer block size $\in [4, 8]$.

## D.7 Best Hyperparameters

**MIMIC-IV (patient trajectories) dataset.** The best hyperparameters for the (conditional sequence generation) models trained on the MIMIC-IV dataset are: dropout rate = 0.6, learning rate = 0.0001, number of layers (blocks in xLSTM) = 8, and number of heads = 4. For xLSTM models, 1D-convolution kernel size = 4 and QVK projection layer block size = 4. For CLEF models, the number of FNN in the concept encoder = 1 (Fig. 8-9).

**eICU (patient trajectories) dataset.** The best hyperparameters for the (conditional sequence generation) models trained on the eICU dataset are: dropout rate = 0.6, learning rate = 0.0001, number of layers (blocks in xLSTM) = 8, and number of heads = 6. For xLSTM models, the number of heads = 2, 1D-convolution kernel size = 4, and QVK projection layer block size = 4. For CLEF models, the number of FNN in the concept encoder = 1 (Fig. 8-9).

**WOT (cellular trajectories) dataset.** The best hyperparameters for the (conditional sequence generation) models trained on the WOT dataset are: dropout rate = 0.6, learning rate = 0.00001, number of layers (or blocks in xLSTM) = 4, and number of heads = 8. For xLSTM models, 1D-convolution kernel size = 4 and QVK projection layer block size = 8. For CLEF models, the number of FNN in the concept encoder = 0 (Fig. 8-9).

**Synthetic tumor growth and semi-synthetic patient trajectories datasets.** For the counterfactual prediction models (including CLEF and non-CLEF models), we follow the best hyperparameters as reported in the original publications of CT (Melnychuk et al., 2022) and CRN (Bica et al., 2020).

**M5 (store sales trajectories) dataset.** The best hyperparameters for the (conditional sequence generation) models trained on the M5 dataset are: dropout rate = 0.6, learning rate = 0.00001, number of layers (or blocks in xLSTM) = 4, and number of heads = 5. For xLSTM models, number of heads = 2, 1D-convolution kernel size = 6, and QVK projection layer block size = 8. For CLEF models, the number of FNN in the concept encoder = 1 (Fig. 8-9).

# E    ADDITIONAL FIGURES AND TABLES

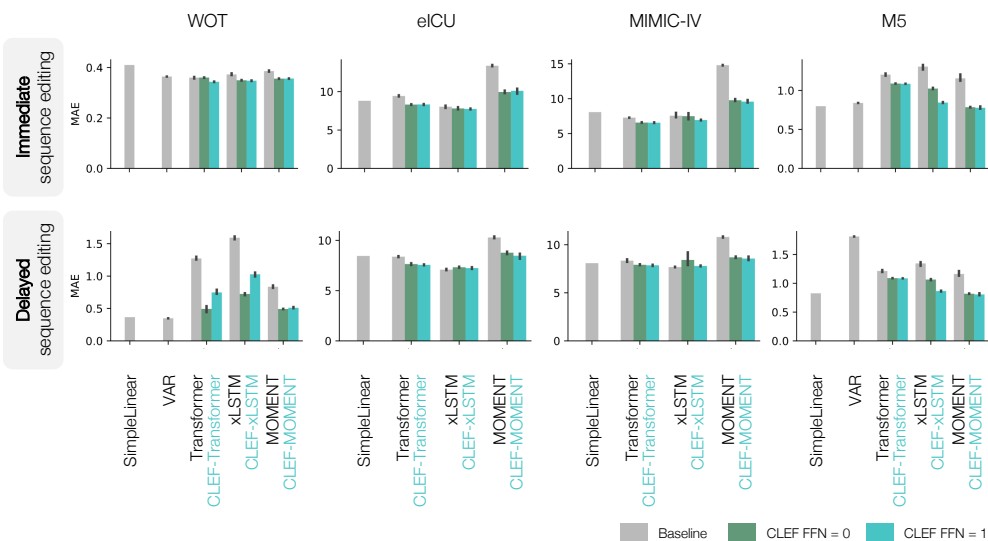

**Figure 8:** Benchmarking the performance of CLEF, baselines, and ablation models on **(a)** immediate and **(b)** delayed sequence editing of observed trajectories (Sec. C.1-C.2). Performance is measured by **MAE** (lower is better). Models are trained on 3 seeds using standard cell-, patient-, or store-centric random splits; error bars show 95% CI. Not shown for visualization purposes are VAR performance on eICU and MIMIC-IV datasets: on immediate sequence editing, MAE for eICU and MIMIC-IV are 55982.74 and 886.05, respectively; on delayed sequence editing, MAE for eICU and MIMIC-IV are $3.02 \times 10^{39}$ and $8.62 \times 10^{23}$, respectively.

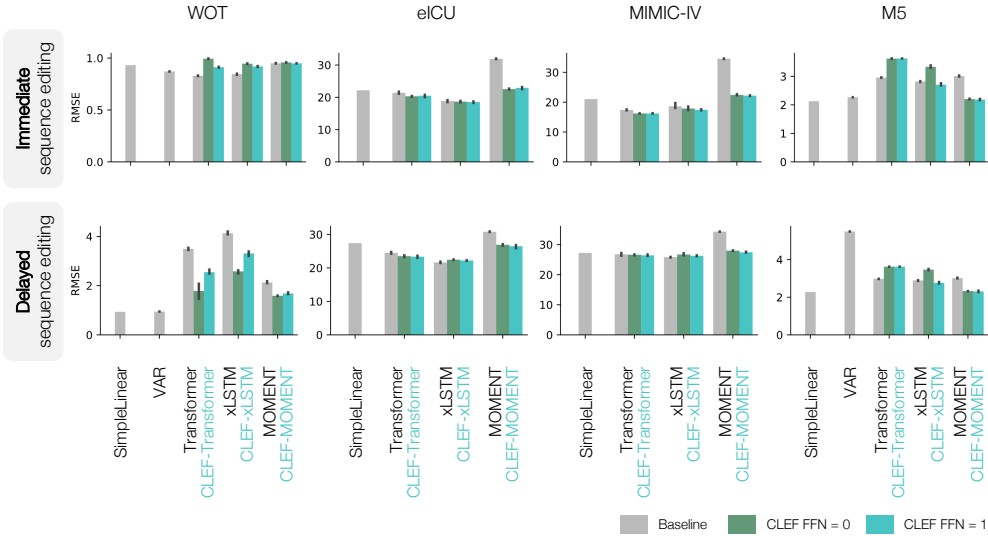

**Figure 9:** Benchmarking the performance of CLEF, baselines, and ablation models on **(a)** immediate and **(b)** delayed sequence editing of observed trajectories (Sec. C.1-C.2). Performance is measured by **RMSE** (lower is better). Models are trained on 3 seeds using standard cell-, patient-, or store-centric random splits; error bars show 95% CI. Not shown for visualization purposes are VAR performance on eICU and MIMIC-IV datasets: on immediate sequence editing, MAE for eICU and MIMIC-IV are 135003.67 and 1793.23, respectively; on delayed sequence editing, MAE for eICU and MIMIC-IV are $5.84 \times 10^{39}$ and $1.59 \times 10^{24}$, respectively.

**Table 5:** Performance of CLEF and non-CLEF models on preserving unedited variables while editing sequences. Models are evaluated on the WOT (cellular) dataset. Generally, CLEF models have less error (MAE) than non-CLEF models. Notably, CLEF models typically have less error (MAE) on "Not Edited" (i.e., preserved) variables than "Edited" variables, whereas the opposite is true for non-CLEF models.

| Model | Immediate | | Delayed | |
|---|---|---|---|---|
| | Edited | Not Edited | Edited | Not Edited |
| Transformer | 0.35860 ± 0.00308 | 0.36449 ± 0.00370 | 1.27342 ± 0.02205 | 1.32864 ± 0.03137 |
| CLEF-transformer | 0.36150 ± 0.00083 | 0.34431 ± 0.00094 | 0.49511 ± 0.04567 | 0.48322 ± 0.04640 |
| xLSTM | 0.37160 ± 0.00386 | 0.37278 ± 0.00188 | 1.59413 ± 0.02165 | 1.65325 ± 0.02717 |
| CLEF-xLSTM | 0.35000 ± 0.00107 | 0.33820 ± 0.00114 | 0.72733 ± 0.01621 | 0.70962 ± 0.01898 |
| MOMENT | 0.38030 ± 0.00231 | 0.42576 ± 0.00080 | 0.83916 ± 0.01876 | 0.81810 ± 0.00788 |
| CLEF-MOMENT | 0.35099 ± 0.00076 | 0.38743 ± 0.00025 | 0.49849 ± 0.00224 | 0.42301 ± 0.00231 |

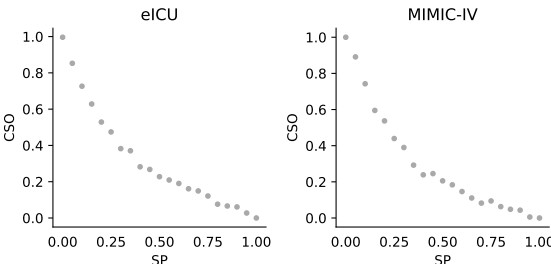

**Figure 10:** Cross-split overlap (CSO) as a function of SPECTRA parameter (SP) for eICU and MIMIC-IV datasets (Sec. C.2). CSO is defined as the number of samples in the test set that are similar to at least one sample in the train set. SP is an internal parameter used by SPECTRA to control the CSO of generated data splits. CSO decreases as SP increases. These data splits are used to evaluate conditional sequence generation models' generalizability to unseen patient trajectories.

**Table 6:** Generalizability of CLEF, baselines, and ablations on eICU and MIMIC-IV datasets (Sec. C.2) in immediate and delayed sequencing. Performance is measured by the area under the spectral performance curve (AUSPC) for MAE (Fig. 11) or RMSE (Fig. 12). Smaller AUSPC values indicate better performance. Models are trained on 3 seeds; standard deviation is reported.

| Model | eICU | | | | MIMIC-IV | | | |
|---|---|---|---|---|---|---|---|---|
| | Immediate | | Delayed | | Immediate | | Delayed | |
| | MAE | RMSE | MAE | RMSE | MAE | RMSE | MAE | RMSE |
| Transformer | 27.06 ± 0.98 | 59.83 ± 1.14 | 22.59 ± 1.21 | 50.29 ± 0.56 | 40.87 ± 0.15 | 71.77 ± 0.21 | 44.61 ± 0.19 | 80.38 ± 0.32 |
| + CLEF | 15.16 ± 1.09 | 32.95 ± 2.47 | 14.36 ± 1.07 | 34.27 ± 2.12 | 32.79 ± 1.41 | 57.76 ± 3.39 | 35.65 ± 1.73 | 65.10 ± 4.43 |
| + CLEF + FFN | 10.99 ± 0.31 | 27.57 ± 0.27 | 9.25 ± 0.60 | 27.69 ± 0.22 | 21.35 ± 3.16 | 36.92 ± 5.46 | 23.83 ± 3.26 | 44.11 ± 5.83 |
| xLSTM | 28.47 ± 0.63 | 62.28 ± 1.38 | 23.11 ± 0.91 | 52.53 ± 1.98 | 40.75 ± 0.30 | 71.90 ± 0.40 | 44.31 ± 0.24 | 80.38 ± 0.33 |
| + CLEF | 16.73 ± 2.16 | 35.43 ± 6.01 | 15.32 ± 2.10 | 34.68 ± 7.09 | 32.06 ± 1.13 | 53.42 ± 2.18 | 33.88 ± 1.98 | 57.73 ± 3.63 |
| + CLEF + FFN | 11.35 ± 0.11 | 28.09 ± 0.08 | 9.04 ± 0.18 | 26.21 ± 0.48 | 21.04 ± 2.32 | 37.50 ± 4.60 | 22.63 ± 2.61 | 42.12 ± 5.03 |
| MOMENT | 53.49 ± 0.03 | 90.54 ± 0.03 | 48.83 ± 0.02 | 82.50 ± 0.02 | 46.55 ± 0.01 | 77.22 ± 0.01 | 50.59 ± 0.02 | 85.72 ± 0.01 |
| + CLEF | 47.69 ± 0.33 | 82.18 ± 0.34 | 40.10 ± 0.44 | 72.70 ± 0.46 | 44.01 ± 0.35 | 73.83 ± 0.63 | 46.88 ± 0.38 | 81.20 ± 1.27 |
| + CLEF + FFN | 47.56 ± 1.60 | 82.81 ± 2.88 | 39.91 ± 1.65 | 72.54 ± 3.20 | 42.92 ± 0.52 | 70.72 ± 1.96 | 45.75 ± 0.65 | 77.35 ± 2.77 |

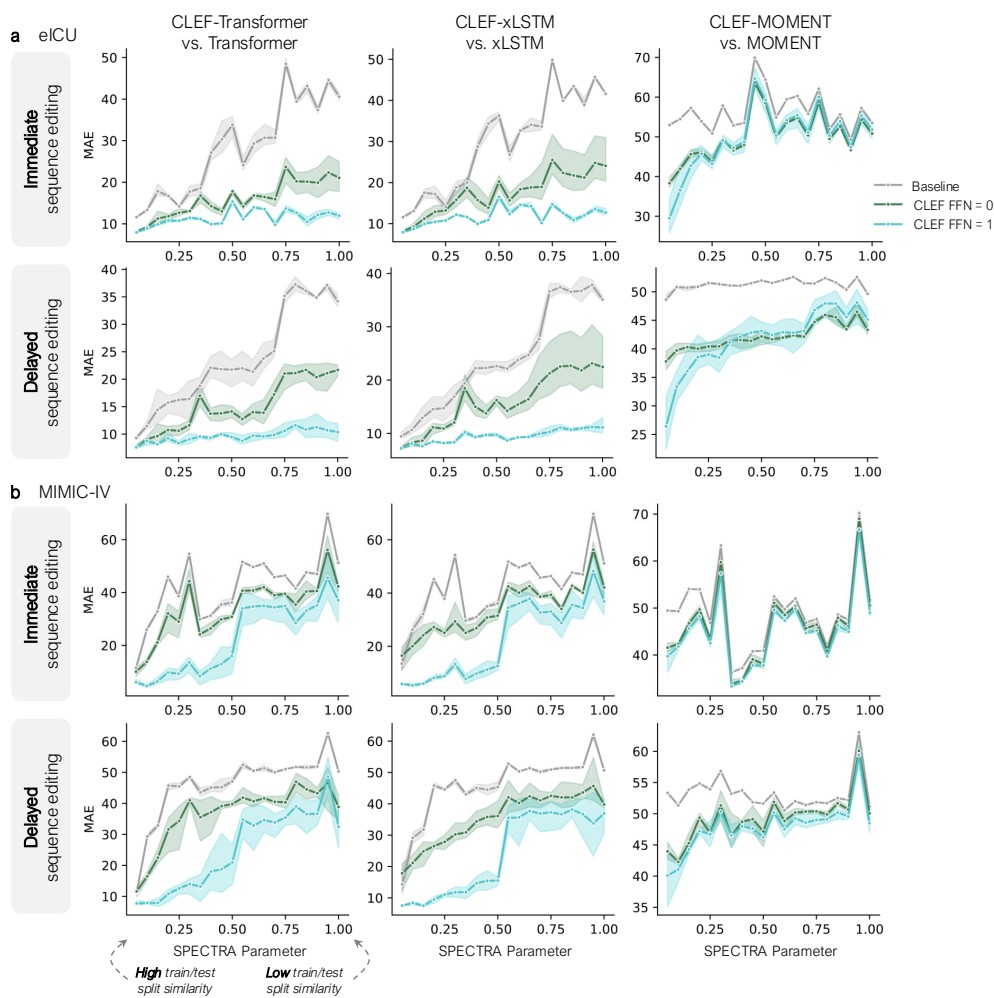

**Figure 11:** Generalizability of CLEF, baselines, and ablation models on **(a)** eICU and **(b)** MIMIC-IV patient datasets (Sec. C.2) in immediate and delayed sequence editing. Performance is measured by **MAE** (lower is better). Models are trained on 3 seeds; error bars show 95% CI. As the SPECTRA parameter increases, the train/test split similarity decreases (Fig. 10). AUSPC evaluation is in Tab. 6.

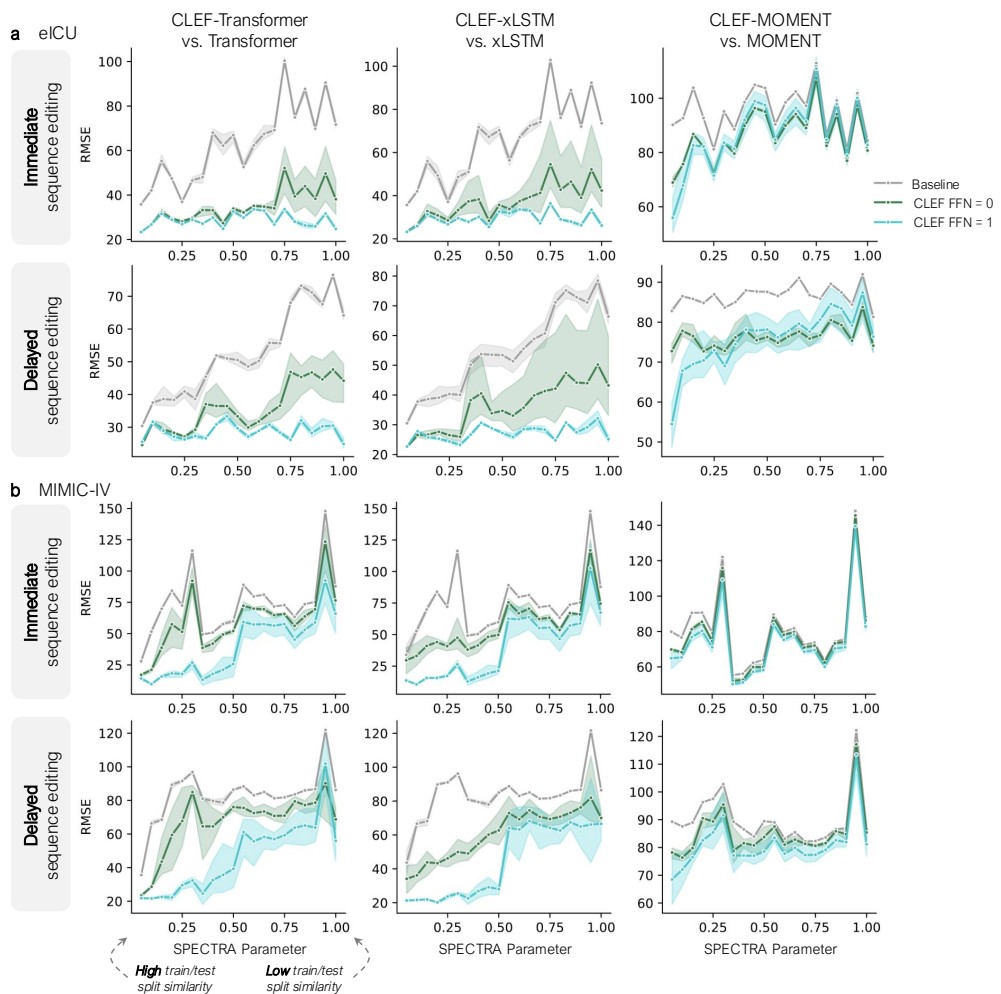

**Figure 12:** Generalizability of CLEF, baselines, and ablation models on **(a)** eICU and **(b)** MIMIC-IV patient datasets (Sec. C.2) in immediate and delayed sequence editing. Performance is measured by **RMSE** (lower is better). Models are trained on 3 seeds; error bars show 95% CI. As the SPECTRA parameter increases, the train/test split similarity decreases (Fig. 10). AUSPC evaluation is in Tab. 6.

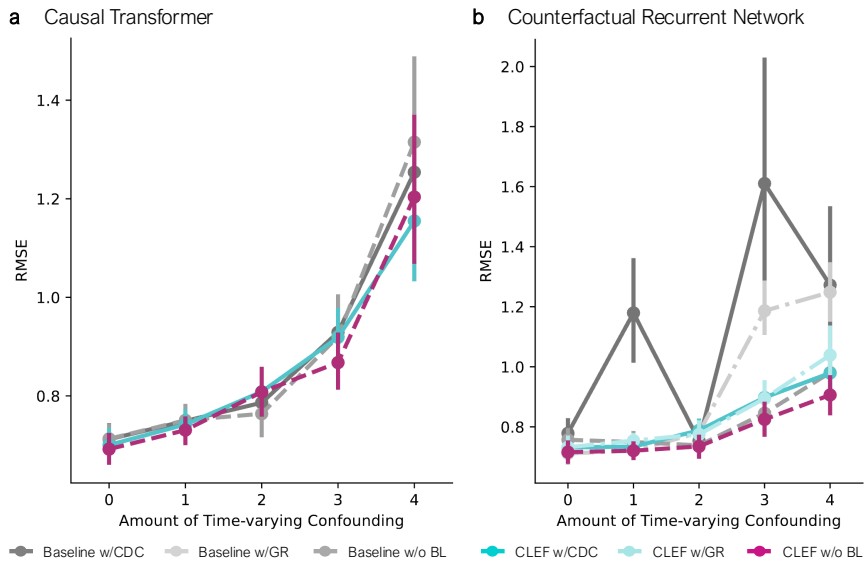

**Figure 13:** Counterfactual $\tau$-step ahead prediction on **tumor growth (random trajectories setting)** with different amounts of time-varying confounding $\gamma$ (Sec. C.3). GR refers to Gradient Reversal loss (Bica et al., 2020); CDC refers to Counterfactual Domain Confusion loss (Melnychuk et al., 2022); BL refers to Balancing Loss (i.e., GR or CDC). Models are trained on 5 seeds; error bars show 95% CI.

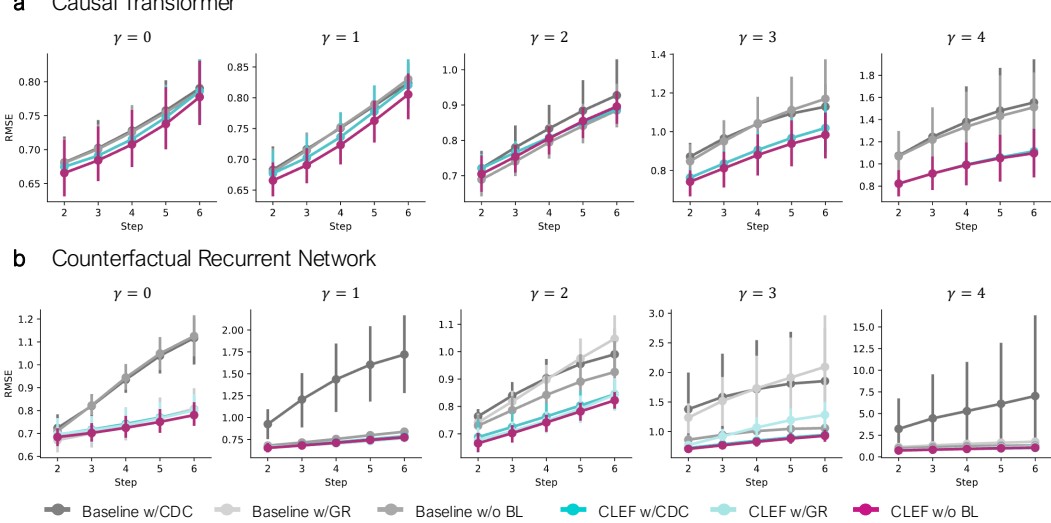

**Figure 14:** Counterfactual $\tau$-step ahead prediction on **tumor growth (single-sliding treatment)** with different amounts of time-varying confounding $\gamma$ (Sec. C.3). GR refers to Gradient Reversal loss (Bica et al., 2020); CDC refers to Counterfactual Domain Confusion loss (Melnychuk et al., 2022); BL refers to Balancing Loss (i.e., GR or CDC). Models are trained on 5 seeds; error bars show 95% CI.

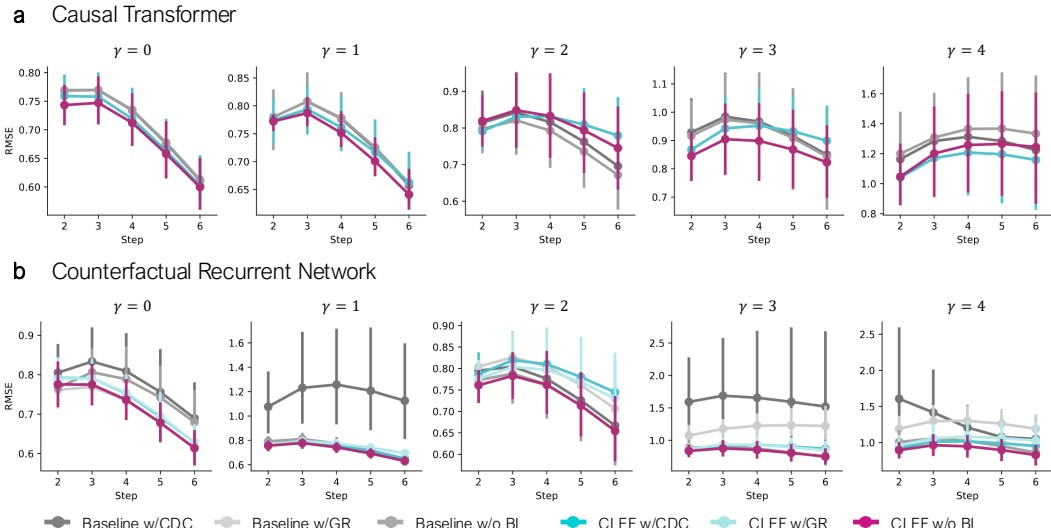

**Figure 15:** Counterfactual $\tau$-step ahead prediction on **tumor growth (random trajectories setting)** with different amounts of time-varying confounding $\gamma$ (Sec. C.3). GR refers to Gradient Reversal loss (Bica et al., 2020); CDC refers to Counterfactual Domain Confusion loss (Melnychuk et al., 2022); BL refers to Balancing Loss (i.e., GR or CDC). Models are trained on 5 seeds; error bars show 95% CI.

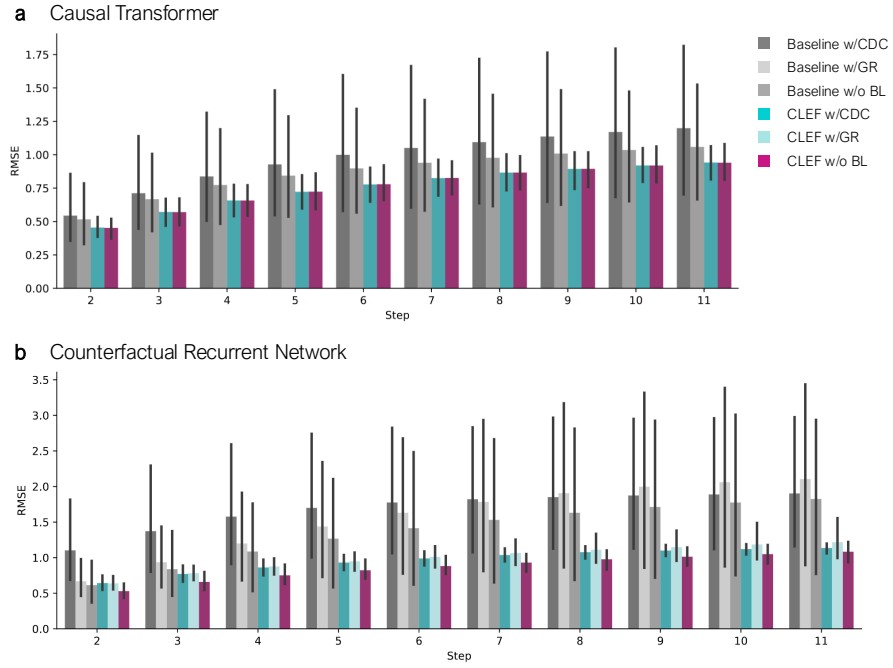

**Figure 16:** Counterfactual $\tau$-step ahead prediction on **semi-synthetic patient ICU trajectories** (Sec. C.4). GR refers to Gradient Reversal loss (Bica et al., 2020); CDC refers to Counterfactual Domain Confusion loss (Melnychuk et al., 2022); BL refers to Balancing Loss (i.e., GR or CDC). Models are trained on 5 seeds; error bars show 95% CI.

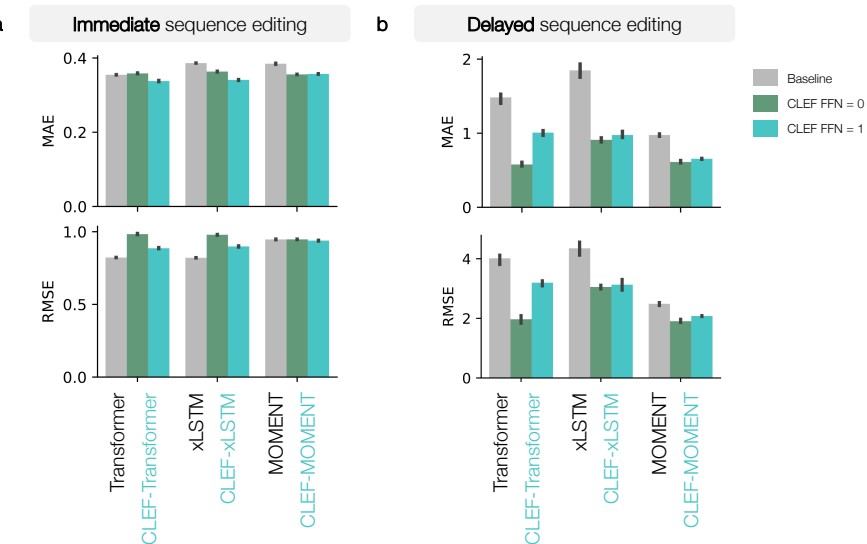

**Figure 17:** Benchmarking the performance of CLEF, baselines, and ablation models on zero-shot **(a)** immediate and **(b)** delayed counterfactual generation of cellular developmental trajectories (Sec. C.1). Performance is measured by MAE (top row) and RMSE (bottom row). Models are trained on 3 seeds; error bars show 95% CI.

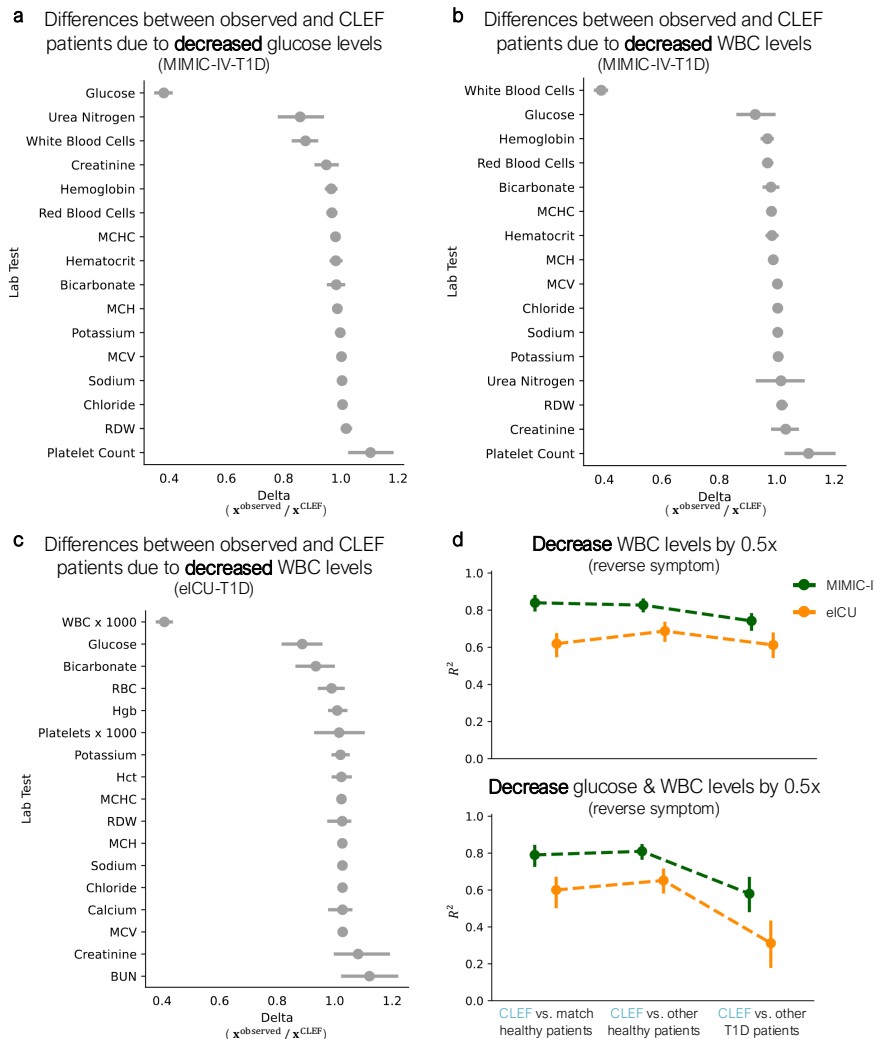

**Figure 18:** CLEF-generated patients via intervention on temporal concepts (Sec. C.2). Observed and CLEF patients are compared to quantify the differences between their lab test trajectories as a result of the intervention to halve the **(a)** glucose levels in T1D patients from MIMIC-IV-T1D, **(b)** WBC levels in T1D patients from MIMIC-IV-T1D, and **(c)** WBC levels in T1D patients from eICU-T1D. **(d)** After intervening on CLEF to halve WBC levels, we observe whether the resulting CLEF patients' trajectories are "healthier" or "sicker" compared to other patients in the real-world cohort (top). Further, we investigate whether the intervention effects are compounded when simultaneously reducing glucose and WBC levels by half (bottom). Error bars show 95% CI.

**Table 7:** As an ablation on the concept decoder, we implement LowR, a decoder $(\mathbf{I} + \mathbf{W})(\mathbf{c} \odot \mathbf{x})$ where $\mathbf{W}$ is low-rank (with rank $= 4, 8, 16$). We evaluate the models on **immediate** sequence editing of observed trajectories. Models are trained on 3 seeds using standard cell-, patient-, or store-centric random splits.

| Dataset | Model | MAE |
|---|---|---|
| MIMIC-IV | Transformer | $7.284 \pm 0.005$ |
| | LowR-transformer (Rank = 4) | $6.673 \pm 0.074$ |
| | LowR-transformer (Rank = 8) | $6.748 \pm 0.067$ |
| | LowR-transformer (Rank = 16) | $6.724 \pm 0.155$ |
| | CLEF-transformer (FFN = 0) | $6.577 \pm 0.038$ |
| | CLEF-transformer (FFN = 1) | $6.562 \pm 0.046$ |
| MIMIC-IV | xLSTM | $7.561 \pm 0.354$ |
| | LowR-xLSTM (Rank = 4) | $7.018 \pm 0.162$ |
| | LowR-xLSTM (Rank = 8) | $7.139 \pm 0.402$ |
| | LowR-xLSTM (Rank = 16) | $7.031 \pm 0.058$ |
| | CLEF-xLSTM (FFN = 0) | $7.493 \pm 0.722$ |
| | CLEF-xLSTM (FFN = 1) | $6.942 \pm 0.052$ |
| MIMIC-IV | MOMENT | $14.804 \pm 0.036$ |
| | LowR-MOMENT (Rank = 4) | $10.671 \pm 0.290$ |
| | LowR-MOMENT (Rank = 8) | $10.603 \pm 0.093$ |
| | LowR-MOMENT (Rank = 16) | $10.685 \pm 0.204$ |
| | CLEF-MOMENT (FFN = 0) | $9.779 \pm 0.148$ |
| | CLEF-MOMENT (FFN = 1) | $9.579 \pm 0.199$ |
| eICU | Transformer | $9.439 \pm 0.082$ |
| | LowR-transformer (Rank = 4) | $8.135 \pm 0.102$ |
| | LowR-transformer (Rank = 8) | $8.265 \pm 0.109$ |
| | LowR-transformer (Rank = 16) | $8.426 \pm 0.162$ |
| | CLEF-transformer (FFN = 0) | $8.319 \pm 0.038$ |
| | CLEF-transformer (FFN = 1) | $8.338 \pm 0.064$ |
| eICU | xLSTM | $8.041 \pm 0.142$ |
| | LowR-xLSTM (Rank = 4) | $7.731 \pm 0.054$ |
| | LowR-xLSTM (Rank = 8) | $8.018 \pm 0.062$ |
| | LowR-xLSTM (Rank = 16) | $8.001 \pm 0.111$ |
| | CLEF-xLSTM (FFN = 0) | $7.815 \pm 0.139$ |
| | CLEF-xLSTM (FFN = 1) | $7.751 \pm 0.050$ |
| eICU | MOMENT | $13.376 \pm 0.089$ |
| | LowR-MOMENT (Rank = 4) | $10.405 \pm 0.099$ |
| | LowR-MOMENT (Rank = 8) | $10.256 \pm 0.252$ |
| | LowR-MOMENT (Rank = 16) | $11.729 \pm 1.460$ |
| | CLEF-MOMENT (FFN = 0) | $9.955 \pm 0.164$ |
| | CLEF-MOMENT (FFN = 1) | $10.111 \pm 0.329$ |
| WOT | Transformer | $0.360 \pm 0.003$ |
| | LowR-transformer (Rank = 4) | $0.365 \pm 0.001$ |
| | LowR-transformer (Rank = 8) | $0.367 \pm 0.001$ |
| | LowR-transformer (Rank = 16) | $0.366 \pm 0.002$ |
| | CLEF-transformer (FFN = 0) | $0.360 \pm 0.001$ |
| | CLEF-transformer (FFN = 1) | $0.344 \pm 0.001$ |
| WOT | xLSTM | $0.373 \pm 0.004$ |
| | LowR-xLSTM (Rank = 4) | $0.362 \pm 0.002$ |
| | LowR-xLSTM (Rank = 8) | $0.363 \pm 0.001$ |
| | LowR-xLSTM (Rank = 16) | $0.365 \pm 0.001$ |
| | CLEF-xLSTM (FFN = 0) | $0.350 \pm 0.001$ |
| | CLEF-xLSTM (FFN = 1) | $0.348 \pm 0.001$ |
| WOT | MOMENT | $0.386 \pm 0.002$ |
| | LowR-MOMENT (Rank = 4) | $0.370 \pm 0.003$ |
| | LowR-MOMENT (Rank = 8) | $0.372 \pm 0.001$ |
| | LowR-MOMENT (Rank = 16) | $0.381 \pm 0.001$ |
| | CLEF-MOMENT (FFN = 0) | $0.356 \pm 0.001$ |
| | CLEF-MOMENT (FFN = 1) | $0.356 \pm 0.001$ |
| M5 | Transformer | $1.203 \pm 0.014$ |
| | LowR-transformer (Rank = 4) | $1.137 \pm 0.036$ |
| | LowR-transformer (Rank = 8) | $1.136 \pm 0.015$ |
| | LowR-transformer (Rank = 16) | $1.151 \pm 0.011$ |
| | CLEF-transformer (FFN = 0) | $1.089 \pm 0.001$ |
| | CLEF-transformer (FFN = 1) | $1.086 \pm 0.000$ |
| M5 | xLSTM | $1.306 \pm 0.032$ |
| | LowR-xLSTM (Rank = 4) | $1.029 \pm 0.011$ |
| | LowR-xLSTM (Rank = 8) | $1.039 \pm 0.007$ |
| | LowR-xLSTM (Rank = 16) | $1.077 \pm 0.028$ |
| | CLEF-xLSTM (FFN = 0) | $1.025 \pm 0.008$ |
| | CLEF-xLSTM (FFN = 1) | $0.845 \pm 0.006$ |
| M5 | MOMENT | $1.156 \pm 0.043$ |
| | LowR-MOMENT (Rank = 4) | $0.835 \pm 0.019$ |
| | LowR-MOMENT (Rank = 8) | $0.849 \pm 0.006$ |
| | LowR-MOMENT (Rank = 16) | $0.875 \pm 0.023$ |
| | CLEF-MOMENT (FFN = 0) | $0.786 \pm 0.003$ |
| | CLEF-MOMENT (FFN = 1) | $0.778 \pm 0.015$ |

**Table 8:** As an ablation on the concept decoder, we implement LowR, a decoder $(\mathbf{I} + \mathbf{W})(\mathbf{c} \odot \mathbf{x})$ where $\mathbf{W}$ is low-rank (with rank = 4, 8, 16). We evaluate the models on **delayed** sequence editing of observed trajectories. Models are trained on 3 seeds using standard cell-, patient-, or store-centric random splits.

| Dataset | Model | MAE |
|---|---|---|
| MIMIC-IV | Transformer | $8.346 \pm 0.129$ |
| | LowR-transformer (Rank = 4) | $7.818 \pm 0.045$ |
| | LowR-transformer (Rank = 8) | $7.943 \pm 0.041$ |
| | LowR-transformer (Rank = 16) | $7.928 \pm 0.027$ |
| | CLEF-transformer (FFN = 0) | $7.915 \pm 0.046$ |
| | CLEF-transformer (FFN = 1) | $7.863 \pm 0.065$ |
| MIMIC-IV | xLSTM | $7.678 \pm 0.029$ |
| | LowR-xLSTM (Rank = 4) | $7.742 \pm 0.075$ |
| | LowR-xLSTM (Rank = 8) | $7.853 \pm 0.026$ |
| | LowR-xLSTM (Rank = 16) | $7.823 \pm 0.025$ |
| | CLEF-xLSTM (FFN = 0) | $8.420 \pm 0.692$ |
| | CLEF-xLSTM (FFN = 1) | $7.794 \pm 0.042$ |
| MIMIC-IV | MOMENT | $10.807 \pm 0.054$ |
| | LowR-MOMENT (Rank = 4) | $8.928 \pm 0.362$ |
| | LowR-MOMENT (Rank = 8) | $8.926 \pm 0.221$ |
| | LowR-MOMENT (Rank = 16) | $9.001 \pm 0.493$ |
| | CLEF-MOMENT (FFN = 0) | $8.688 \pm 0.074$ |
| | CLEF-MOMENT (FFN = 1) | $8.563 \pm 0.182$ |
| eICU | Transformer | $8.377 \pm 0.047$ |
| | LowR-transformer (Rank = 4) | $7.465 \pm 0.201$ |
| | LowR-transformer (Rank = 8) | $7.567 \pm 0.070$ |
| | LowR-transformer (Rank = 16) | $7.788 \pm 0.128$ |
| | CLEF-transformer (FFN = 0) | $7.643 \pm 0.083$ |
| | CLEF-transformer (FFN = 1) | $7.566 \pm 0.040$ |
| eICU | xLSTM | $7.086 \pm 0.066$ |
| | LowR-xLSTM (Rank = 4) | $7.253 \pm 0.023$ |
| | LowR-xLSTM (Rank = 8) | $7.375 \pm 0.094$ |
| | LowR-xLSTM (Rank = 16) | $7.414 \pm 0.112$ |
| | CLEF-xLSTM (FFN = 0) | $7.324 \pm 0.048$ |
| | CLEF-xLSTM (FFN = 1) | $7.241 \pm 0.082$ |
| eICU | MOMENT | $10.289 \pm 0.086$ |
| | LowR-MOMENT (Rank = 4) | $8.873 \pm 0.157$ |
| | LowR-MOMENT (Rank = 8) | $9.074 \pm 0.352$ |
| | LowR-MOMENT (Rank = 16) | $9.608 \pm 1.133$ |
| | CLEF-MOMENT (FFN = 0) | $8.773 \pm 0.120$ |
| | CLEF-MOMENT (FFN = 1) | $8.472 \pm 0.259$ |
| WOT | Transformer | $1.273 \pm 0.022$ |
| | LowR-transformer (Rank = 4) | $0.552 \pm 0.049$ |
| | LowR-transformer (Rank = 8) | $0.547 \pm 0.003$ |
| | LowR-transformer (Rank = 16) | $0.525 \pm 0.057$ |
| | CLEF-transformer (FFN = 0) | $0.494 \pm 0.045$ |
| | CLEF-transformer (FFN = 1) | $0.748 \pm 0.035$ |
| WOT | xLSTM | $1.592 \pm 0.022$ |
| | LowR-xLSTM (Rank = 4) | $0.857 \pm 0.086$ |
| | LowR-xLSTM (Rank = 8) | $0.854 \pm 0.047$ |
| | LowR-xLSTM (Rank = 16) | $0.882 \pm 0.048$ |
| | CLEF-xLSTM (FFN = 0) | $0.724 \pm 0.016$ |
| | CLEF-xLSTM (FFN = 1) | $1.028 \pm 0.030$ |
| WOT | MOMENT | $0.836 \pm 0.018$ |
| | LowR-MOMENT (Rank = 4) | $0.534 \pm 0.010$ |
| | LowR-MOMENT (Rank = 8) | $0.542 \pm 0.005$ |
| | LowR-MOMENT (Rank = 16) | $0.576 \pm 0.003$ |
| | CLEF-MOMENT (FFN = 0) | $0.493 \pm 0.002$ |
| | CLEF-MOMENT (FFN = 1) | $0.512 \pm 0.009$ |
| M5 | Transformer | $1.214 \pm 0.016$ |
| | LowR-transformer (Rank = 4) | $1.133 \pm 0.036$ |
| | LowR-transformer (Rank = 8) | $1.137 \pm 0.010$ |
| | LowR-transformer (Rank = 16) | $1.149 \pm 0.012$ |
| | CLEF-transformer (FFN = 0) | $1.090 \pm 0.001$ |
| | CLEF-transformer (FFN = 1) | $1.086 \pm 0.000$ |
| M5 | xLSTM | $1.345 \pm 0.034$ |
| | LowR-xLSTM (Rank = 4) | $1.067 \pm 0.014$ |
| | LowR-xLSTM (Rank = 8) | $1.075 \pm 0.013$ |
| | LowR-xLSTM (Rank = 16) | $1.117 \pm 0.030$ |
| | CLEF-xLSTM (FFN = 0) | $1.066 \pm 0.009$ |
| | CLEF-xLSTM (FFN = 1) | $0.867 \pm 0.008$ |
| M5 | MOMENT | $1.162 \pm 0.044$ |
| | LowR-MOMENT (Rank = 4) | $0.858 \pm 0.015$ |
| | LowR-MOMENT (Rank = 8) | $0.876 \pm 0.006$ |
| | LowR-MOMENT (Rank = 16) | $0.907 \pm 0.026$ |
| | CLEF-MOMENT (FFN = 0) | $0.819 \pm 0.004$ |
| | CLEF-MOMENT (FFN = 1) | $0.807 \pm 0.017$ |

4️⃣

**Table 9:** To show the benefit of single-step generation, we arbitrarily add three intermediate steps between $t_i$ and $t_j$ before finally predicting the observed $\mathbf{x}_{:,t_j}$. Because we only have the ground truth for $\mathbf{x}_{:,t_j}$, we evaluate only on the predicted $\hat{\mathbf{x}}_{:,t_j}$. Models are trained on 3 seeds using standard cell-, patient-, or store-centric random splits.

| Dataset | Model | MAE |
|---------|-------|-----|
| MIMIC-IV | Transformer | $8.346 \pm 0.129$ |
|  | Transformer(intermediate) | $8.627 \pm 0.365$ |
|  | Clef-transformer | $7.915 \pm 0.046$ |
|  | Clef-transformer(intermediate) | $8.279 \pm 0.135$ |
| MIMIC-IV | xLSTM | $7.678 \pm 0.029$ |
|  | xLSTM(intermediate) | $8.281 \pm 0.414$ |
|  | Clef-xLSTM | $8.420 \pm 0.692$ |
|  | Clef-xLSTM(intermediate) | $8.684 \pm 0.767$ |
| MIMIC-IV | MOMENT | $10.807 \pm 0.054$ |
|  | MOMENT(intermediate) | $10.673 \pm 0.059$ |
|  | Clef-MOMENT | $8.688 \pm 0.074$ |
|  | Clef-MOMENT(intermediate) | $9.096 \pm 0.137$ |
| eICU | Transformer | $8.377 \pm 0.047$ |
|  | Transformer(intermediate) | $9.707 \pm 1.646$ |
|  | Clef-transformer | $7.643 \pm 0.083$ |
|  | Clef-transformer(intermediate) | $8.144 \pm 0.247$ |
| eICU | xLSTM | $7.086 \pm 0.066$ |
|  | xLSTM(intermediate) | $8.015 \pm 0.566$ |
|  | Clef-xLSTM | $7.324 \pm 0.048$ |
|  | Clef-xLSTM(intermediate) | $8.121 \pm 0.371$ |
| eICU | MOMENT | $10.289 \pm 0.086$ |
|  | MOMENT(intermediate) | $10.233 \pm 0.069$ |
|  | Clef-MOMENT | $8.773 \pm 0.120$ |
|  | Clef-MOMENT(intermediate) | $9.591 \pm 0.198$ |
| WOT | Transformer | $1.273 \pm 0.022$ |
|  | Transformer(intermediate) | $1.068 \pm 0.053$ |
|  | Clef-transformer | $0.494 \pm 0.045$ |
|  | Clef-transformer(intermediate) | $0.411 \pm 0.026$ |
| WOT | xLSTM | $1.592 \pm 0.022$ |
|  | xLSTM(intermediate) | $1.444 \pm 0.043$ |
|  | Clef-xLSTM | $0.724 \pm 0.016$ |
|  | Clef-xLSTM(intermediate) | $0.395 \pm 0.010$ |
| WOT | MOMENT | $0.836 \pm 0.018$ |
|  | MOMENT(intermediate) | $0.764 \pm 0.015$ |
|  | Clef-MOMENT | $0.493 \pm 0.002$ |
|  | Clef-MOMENT(intermediate) | $0.463 \pm 0.003$ |
| M5 | Transformer | $1.214 \pm 0.016$ |
|  | Transformer(intermediate) | $1.205 \pm 0.016$ |
|  | Clef-transformer | $1.090 \pm 0.001$ |
|  | Clef-transformer(intermediate) | $1.075 \pm 0.001$ |
| M5 | xLSTM | $1.345 \pm 0.034$ |
|  | xLSTM(intermediate) | $1.271 \pm 0.030$ |
|  | Clef-xLSTM | $1.066 \pm 0.009$ |
|  | Clef-xLSTM(intermediate) | $1.121 \pm 0.011$ |
| M5 | MOMENT | $1.162 \pm 0.044$ |
|  | MOMENT(intermediate) | $1.137 \pm 0.043$ |
|  | Clef-MOMENT | $0.819 \pm 0.004$ |
|  | Clef-MOMENT(intermediate) | $0.799 \pm 0.005$ |

**Table 10:** Binary cross entropy (BCE) loss for predicting the treatment from the learned representations of CLEF and non-CLEF models on **tumor growth (single-sliding treatment)** with different amounts of time-varying confounding $\gamma$ (Sec. C.3). Models are trained on 5 seeds.

| Model | BCE Loss |
|---|---|
| CT w/CDC | $1.499 \pm 0.099$ |
| **CLEF-CT w/CDC** | $1.506 \pm 0.096$ |
| CRN w/GR | $1.193 \pm 0.169$ |
| **CLEF-CRN w/GR** | $1.196 \pm 0.167$ |
| CRN w/CDC | $1.695 \pm 0.652$ |
| **CLEF-CRN w/CDC** | $1.597 \pm 0.187$ |

**Table 11:** Binary cross entropy (BCE) loss for predicting the treatment from the learned representations of CLEF and non-CLEF models on **tumor growth (random trajectories setting)** with different amounts of time-varying confounding $\gamma$ (Sec. C.3). Models are trained on 5 seeds.

| Model | BCE Loss |
|---|---|
| CT w/CDC | $1.497 \pm 0.089$ |
| **CLEF-CT w/CDC** | $1.509 \pm 0.098$ |
| CRN w/GR | $1.194 \pm 0.168$ |
| **CLEF-CRN w/GR** | $1.196 \pm 0.167$ |
| CRN w/CDC | $1.583 \pm 0.178$ |
| **CLEF-CRN w/CDC** | $1.598 \pm 0.188$ |

**Table 12:** Binary cross entropy (BCE) loss for predicting the treatment from the learned representations of CLEF and non-CLEF models on **semi-synthetic patient ICU trajectories** (Sec. C.4). Models are trained on 5 seeds.

| Model | BCE Loss |
|---|---|
| CT w/CDC | $1.656 \pm 0.631$ |
| **CLEF-CT w/CDC** | $1.300 \pm 0.375$ |
| CRN w/GR | $0.330 \pm 0.149$ |
| **CLEF-CRN w/GR** | $0.364 \pm 0.151$ |
| CRN w/CDC | $1.706 \pm 0.662$ |
| **CLEF-CRN w/CDC** | $1.650 \pm 0.640$ |

**Table 13:** Generated counterfactual eICU-T1D patients via intervention (i.e., decrease glucose levels by 0.5x, increase glucose levels by 2x) on temporal concepts from CLEF and SimpleLinear (ablation). We compare the generated trajectories against observed trajectories of matched healthy, other healthy, and other T1D patients. Higher $R^2$ indicates that patient pairs are more similar. CLEF's generated trajectories are more similar to those of the other T1D patients ($R^2 = 0.348 \pm 0.149$) than matched healthy ($R^2 = 0.147 \pm 0.111$) and other healthy ($R^2 = 0.175 \pm 0.106$) patients. On the other hand, SimpleLinear's generated trajectories have consistent and low similarity to those of matched healthy ($R^2 = 0.287 \pm 0.141$), other healthy ($R^2 = 0.263 \pm 0.159$), and other T1D ($R^2 = 0.240 \pm 0.087$) patients. Also, the $R^2$ between SimpleLinear's generated trajectories and the trajectories of the matched healthy, other healthy, and other T1D patients are comparable. On the other hand, the $R^2$ between CLEF's generated trajectories and healthy trajectories are significantly lower than the $R^2$ between CLEF's generated trajectories and other T1D patients'. Unlike CLEF, SimpleLinear cannot generate trajectories that resemble the trajectories of healthier or sicker patients, depending on the intervention.

| Model | Intervention | Matched Healthy | Other Healthy | Other T1D |
|---|---|---|---|---|
| **CLEF** | 0.5x glucose | $0.757 \pm 0.227$ | $0.737 \pm 0.126$ | $0.600 \pm 0.122$ |
| SimpleLinear | 0.5x glucose | $0.403 \pm 0.226$ | $0.390 \pm 0.233$ | $0.288 \pm 0.109$ |
| **CLEF** | 2x glucose | $0.147 \pm 0.111$ | $0.175 \pm 0.106$ | $0.348 \pm 0.149$ |
| SimpleLinear | 2x glucose | $0.287 \pm 0.141$ | $0.263 \pm 0.159$ | $0.240 \pm 0.087$ |

**Table 14:** Generated counterfactual MIMIC-IV-T1D patients via intervention (i.e., decrease glucose levels by 0.5x, increase glucose levels by 2x) on temporal concepts from CLEF and SimpleLinear (ablation). We compare the generated trajectories against observed trajectories of matched healthy, other healthy, and other T1D patients. Higher $R^2$ indicates that patient pairs are more similar. CLEF's generated trajectories are more similar to those of the other T1D patients ($R^2 = 0.472 \pm 0.134$) than matched healthy ($R^2 = 0.364 \pm 0.126$) and other healthy ($R^2 = 0.344 \pm 0.129$) patients, which is expected. On the other hand, SimpleLinear's trajectories are more similar to those of the matched healthy ($R^2 = 0.790 \pm 0.110$) and other healthy ($R^2 = 0.709 \pm 0.051$) patients than other T1D patients ($R^2 = 0.541 \pm 0.056$), which is unexpected. Unlike CLEF, SimpleLinear cannot generate trajectories that resemble the trajectories of healthier or sicker patients, depending on the intervention.

| Model | Intervention | Matched Healthy | Other Healthy | Other T1D |
|---|---|---|---|---|
| **CLEF** | 0.5x glucose | $0.838 \pm 0.140$ | $0.814 \pm 0.076$ | $0.692 \pm 0.071$ |
| SimpleLinear | 0.5x glucose | $0.713 \pm 0.166$ | $0.689 \pm 0.113$ | $0.564 \pm 0.082$ |
| **CLEF** | 2x glucose | $0.364 \pm 0.126$ | $0.344 \pm 0.129$ | $0.472 \pm 0.134$ |
| SimpleLinear | 2x glucose | $0.790 \pm 0.110$ | $0.709 \pm 0.051$ | $0.541 \pm 0.056$ |

**Table 15:** We provide qualitative trajectories for a patient. CLEF seems to generate lab test values that are closer to the observed lab test values than non-CLEF models.

| Lab | Time | Observed | CLEF | Non-CLEF |
|---|---|---|---|---|
| Hct | 1900-01-03 09:58:00 | 28.200 | 31.664 | 32.347 |
| | 1900-01-04 10:09:00 | 26.700 | 32.022 | 32.805 |
| | 1900-01-05 09:31:00 | 26.000 | 31.885 | 32.831 |
| | 1900-01-06 06:19:00 | 27.500 | 32.025 | 32.863 |
| | 1900-01-07 10:00:00 | 27.800 | 31.779 | 32.357 |
| | 1900-01-09 09:29:00 | 29.500 | 31.084 | 32.169 |
| | 1900-01-10 10:04:00 | 30.600 | 30.771 | 31.962 |
| Hgb | 1900-01-03 09:58:00 | 9.700 | 10.193 | 10.558 |
| | 1900-01-04 10:09:00 | 9.000 | 10.273 | 10.665 |
| | 1900-01-05 09:31:00 | 8.700 | 10.230 | 10.541 |
| | 1900-01-06 06:19:00 | 9.200 | 10.295 | 10.605 |
| | 1900-01-07 10:00:00 | 9.300 | 10.182 | 10.472 |
| | 1900-01-09 09:29:00 | 9.700 | 9.938 | 10.391 |
| | 1900-01-10 10:04:00 | 9.800 | 9.804 | 10.192 |
| MCH | 1900-01-03 09:58:00 | 31.700 | 29.848 | 27.969 |
| | 1900-01-04 10:09:00 | 31.400 | 29.904 | 27.874 |
| | 1900-01-05 09:31:00 | 31.000 | 29.626 | 27.617 |
| | 1900-01-06 06:19:00 | 30.700 | 29.649 | 27.596 |
| | 1900-01-07 10:00:00 | 31.200 | 29.467 | 27.667 |
| | 1900-01-09 09:29:00 | 30.700 | 29.874 | 28.041 |
| | 1900-01-10 10:04:00 | 30.100 | 29.315 | 27.816 |
| MCV | 1900-01-03 09:58:00 | 92.200 | 93.786 | 89.05 |
| | 1900-01-04 10:09:00 | 93.000 | 93.745 | 89.26 |
| | 1900-01-05 09:31:00 | 92.500 | 93.373 | 88.64 |
| | 1900-01-06 06:19:00 | 91.700 | 93.508 | 88.65 |
| | 1900-01-07 10:00:00 | 93.300 | 93.270 | 88.68 |
| | 1900-01-09 09:29:00 | 93.400 | 94.305 | 89.65 |
| | 1900-01-10 10:04:00 | 93.900 | 93.824 | 89.26 |
| Sodium | 1900-01-03 09:58:00 | 139.000 | 137.202 | 140.934 |
| | 1900-01-04 10:09:00 | 137.000 | 137.157 | 141.779 |
| | 1900-01-05 09:31:00 | 138.000 | 136.695 | 141.848 |
| | 1900-01-06 06:19:00 | 137.000 | 137.239 | 141.967 |
| | 1900-01-07 10:00:00 | 138.000 | 136.807 | 142.173 |
| | 1900-01-09 09:29:00 | 137.000 | 137.781 | 142.912 |
| | 1900-01-10 10:04:00 | 137.000 | 138.031 | 141.562 |

**Table 16:** We provide qualitative trajectories for a patient. CLEF seems to generate lab test values that are closer to the observed lab test values than non-CLEF models.

| Lab | Time | Observed | CLEF | Non-CLEF |
|---|---|---|---|---|
| Hct | 1900-01-07 12:31:00 | 34.400 | 32.783 | 32.298 |
| | 1900-01-08 10:29:00 | 33.800 | 32.530 | 31.573 |
| | 1900-01-09 11:58:00 | 34.500 | 32.110 | 31.828 |
| | 1900-01-10 18:08:00 | 36.600 | 32.288 | 31.464 |
| | 1900-01-11 11:24:00 | 33.300 | 32.224 | 31.802 |
| | 1900-01-12 11:28:00 | 35.100 | 32.138 | 31.043 |
| | 1900-01-13 10:21:00 | 36.400 | 32.361 | 31.413 |
| | 1900-01-14 11:10:00 | 33.600 | 31.970 | 31.080 |
| | 1900-01-15 09:25:00 | 34.300 | 31.684 | 30.439 |
| | 1900-01-16 10:14:00 | 33.400 | 31.706 | 31.286 |
| | 1900-01-17 09:56:00 | 35.800 | 31.411 | 30.515 |
| | 1900-01-18 10:13:00 | 34.400 | 31.610 | 31.164 |
| | 1900-01-19 09:25:00 | 32.300 | 31.057 | 30.230 |
| | 1900-01-20 10:47:00 | 31.100 | 31.215 | 30.727 |
| | 1900-01-21 11:07:00 | 31.300 | 30.864 | 29.923 |
| Hgb | 1900-01-07 12:31:00 | 11.300 | 10.967 | 10.607 |
| | 1900-01-08 10:29:00 | 11.100 | 10.788 | 10.367 |
| | 1900-01-09 11:58:00 | 11.500 | 10.629 | 10.446 |
| | 1900-01-10 18:08:00 | 12.200 | 10.736 | 10.296 |
| | 1900-01-11 11:24:00 | 11.100 | 10.695 | 10.388 |
| | 1900-01-12 11:28:00 | 11.400 | 10.674 | 10.203 |
| | 1900-01-13 10:21:00 | 12.200 | 10.755 | 10.281 |
| | 1900-01-14 11:10:00 | 10.900 | 10.605 | 10.253 |
| | 1900-01-15 09:25:00 | 10.900 | 10.527 | 10.008 |
| | 1900-01-16 10:14:00 | 10.900 | 10.542 | 10.178 |
| | 1900-01-17 09:56:00 | 11.500 | 10.425 | 9.923 |
| | 1900-01-18 10:13:00 | 11.000 | 10.495 | 10.108 |
| | 1900-01-19 09:25:00 | 10.400 | 10.290 | 9.748 |
| | 1900-01-20 10:47:00 | 10.000 | 10.334 | 9.848 |
| | 1900-01-21 11:07:00 | 9.800 | 10.198 | 9.611 |
| MCH | 1900-01-07 12:31:00 | 30.100 | 29.502 | 27.910 |
| | 1900-01-08 10:29:00 | 30.000 | 29.252 | 27.722 |
| | 1900-01-09 11:58:00 | 30.700 | 29.300 | 27.923 |
| | 1900-01-10 18:08:00 | 30.600 | 29.222 | 27.676 |
| | 1900-01-11 11:24:00 | 30.200 | 29.299 | 27.440 |
| | 1900-01-12 11:28:00 | 29.800 | 29.160 | 27.412 |
| | 1900-01-13 10:21:00 | 30.300 | 29.183 | 27.374 |
| | 1900-01-14 11:10:00 | 30.100 | 29.106 | 26.954 |
| | 1900-01-15 09:25:00 | 29.600 | 29.062 | 27.089 |
| | 1900-01-16 10:14:00 | 30.200 | 29.197 | 27.007 |
| | 1900-01-17 09:56:00 | 29.900 | 29.024 | 26.864 |
| | 1900-01-18 10:13:00 | 29.900 | 29.153 | 26.852 |
| | 1900-01-19 09:25:00 | 30.100 | 29.044 | 26.847 |
| | 1900-01-20 10:47:00 | 29.900 | 29.046 | 26.571 |
| | 1900-01-21 11:07:00 | 29.900 | 28.978 | 26.824 |
| MCV | 1900-01-07 12:31:00 | 91.700 | 87.581 | 85.512 |
| | 1900-01-08 10:29:00 | 91.400 | 88.247 | 84.877 |
| | 1900-01-09 11:58:00 | 92.000 | 88.324 | 85.455 |
| | 1900-01-10 18:08:00 | 91.700 | 88.097 | 84.901 |
| | 1900-01-11 11:24:00 | 90.500 | 88.038 | 84.469 |
| | 1900-01-12 11:28:00 | 91.900 | 87.922 | 84.194 |
| | 1900-01-13 10:21:00 | 90.500 | 87.972 | 84.065 |
| | 1900-01-14 11:10:00 | 92.800 | 87.738 | 83.119 |
| | 1900-01-15 09:25:00 | 93.200 | 87.634 | 82.869 |
| | 1900-01-16 10:14:00 | 92.500 | 87.803 | 83.296 |
| | 1900-01-17 09:56:00 | 93.000 | 87.629 | 82.255 |
| | 1900-01-18 10:13:00 | 93.500 | 87.828 | 82.804 |
| | 1900-01-19 09:25:00 | 93.400 | 87.821 | 82.391 |
| | 1900-01-20 10:47:00 | 92.800 | 87.924 | 82.099 |
| | 1900-01-21 11:07:00 | 95.400 | 88.035 | 82.560 |
| Sodium | 1900-01-07 12:31:00 | 135.000 | 136.593 | 139.930 |
| | 1900-01-08 10:29:00 | 137.000 | 135.900 | 139.291 |
| | 1900-01-09 11:58:00 | 136.000 | 137.241 | 141.514 |
| | 1900-01-10 18:08:00 | 136.000 | 137.631 | 142.512 |
| | 1900-01-11 11:24:00 | 136.000 | 138.199 | 142.694 |
| | 1900-01-12 11:28:00 | 135.000 | 138.312 | 143.094 |
| | 1900-01-13 10:21:00 | 133.000 | 138.811 | 142.580 |
| | 1900-01-14 11:10:00 | 135.000 | 138.777 | 141.410 |
| | 1900-01-15 09:25:00 | 132.000 | 138.393 | 141.368 |
| | 1900-01-16 10:14:00 | 134.000 | 139.036 | 141.228 |
| | 1900-01-17 09:56:00 | 135.000 | 138.747 | 139.891 |
| | 1900-01-18 10:13:00 | 137.000 | 138.992 | 140.015 |
| | 1900-01-19 09:25:00 | 136.000 | 138.943 | 139.224 |
| | 1900-01-20 10:47:00 | 135.000 | 138.693 | 138.322 |
| | 1900-01-21 11:07:00 | 138.000 | 138.965 | 138.743 |

# F LIMITATIONS & FUTURE DIRECTIONS

There are a few key limitations of CLEF. Firstly, we define temporal concepts such that each element represents a unique measured variable in the sequence (e.g., gene expression, lab test). Instead, it may be beneficial to learn higher-order relationships between the measured variables or across time as abstract hierarchical concepts (LCM et al., 2024; Kacprzyk et al., 2024). Secondly, while CLEF is able to generate counterfactual sequences for any condition, including those it may not have seen during training, CLEF could potentially improve with additional guidance from a real-world causal model for the system or domain of interest (Chatzi et al., 2025). Since defining such a real-world causal graph is a major challenge, one promising future direction could be to enable user interventions, such as those performed in our T1D case studies, to finetune CLEF. There are many opportunities to improve CLEF. While elegant, the multiplicative decoder may impose a linearity assumption. Also, in the concept decoder, it can be a limitation to use the last time step $\mathbf{x}_{:,t_i}$ for decoding the next time step if there are missing or zero values. Straightforward solutions are to take the mean of the historical context or impute the missing values in the historical data before forecasting $t_j$. In addition to temporal editing of trajectories, future work may extend CLEF for spatial and positional editing.

