# OpenReview forum: "Controllable Sequence Editing for Biological and Clinical Trajectories"
_ICLR.cc/2026/Conference — ICLR 2026 Poster_

### Official Review · Reviewer_hatS · 2025-10-28

**Soundness:** 2
**Presentation:** 3
**Contribution:** 2
**Rating:** 4
**Confidence:** 4

**Summary:**

The paper studies controllable time‑series editing and proposes CLEF, a framework for conditional generation of time‑series given an intervention time and value. The model encodes past history, time difference, and the intervention into a temporal context embedding that modulates the extent of output change. The method is evaluated on conditional and counterfactual generation tasks across eight datasets from multiple domains, including cellular dynamics, healthcare, and sales.

**Strengths:**

* The work addresses counterfactual outcome estimation, an important problem in computational biology and healthcare.
* The introduction motivates the problem well, and Fig. 1 is informative. The writing is generally clear. Figures are descriptive and polished, although they are currently too small to read without zooming, so they should be enlarged.
* The model design is simple and clean.
* The method is evaluated on a substantial number of datasets from three domains; however, the experimental setup is difficult to follow from the main text at times (see Weaknesses).

**Weaknesses:**

The paper exhibits several weaknesses: the proposed method shows unexpectedly poor performance relative to a simple linear baseline, the experimental setup lacks clarity in several places, and the results as presented in the Experiments section are not fully aligned with the abstract. I am willing to reconsider my score if these concerns are addressed convincingly.

**(i) Performance relative to simple baselines**

This is the main weakness. In R1 and R2, performance is unexpectedly poor relative to a simple linear baseline: the linear baseline appears better on average than non‑CLEF models and comparable to CLEF models. This raises concerns about applicability and practicality. If these models do not outperform simply using the previous time point, the value of training and deploying them is unclear.

**(ii) Ambiguity in experimental setup and model variants**

It is unclear how CLEF and non‑CLEF models differ. Lines 297–298 state that “CLEF and non‑CLEF differ only in the components needed to learn temporal concepts,” which is ambiguous, and Appendix D is difficult to follow. Please clarify whether non‑CLEF models (e.g., Transformer, xLSTM) are implemented exactly as in their original papers or adapted in some way. It is surprising that these models underperform the simple baseline on average in Fig. 4.

**(iii) Abstract–experiment misalignment and selective reporting**

The abstract’s presentation of results does not accurately reflect the evidence shown in the Experiments section. It is unclear where the abstract’s claims are reported in the main results. Reporting “accuracy gain up to” an arbitrary or worst baseline is not a fair summary. In Fig. 4, the simple baseline appears on par with the proposed complex models. Examples:
* Lines 21–22: “… immediate sequence editing accuracy by up to 36.74% (MAE).”
* Lines 22–23: “… delayed sequence editing by up to 65.71% (MAE).”

**(iv) Task definition: R5 vs. R4**

The definition of R5 (zero‑shot conditional generation of counterfactual trajectories) is not clear. The distinction from counterfactual outcome estimation in R4 should be specified.

**(v) Intervention mechanics in R6**

For R6, it is unclear how interventions on temporal concepts are performed to decrease or increase glucose levels. As a sanity check, can authors construct a simple baseline also for this task?

**Questions:**

* Question: Have the authors considered/experimented with other concept encoder and concept decoder architectures?
* Suggestion: Lack of qualitative trajectory examples. The paper (and related counterfactual outcome estimation work [Bica+20; Seedat+22; Melnychuk+22]) does not provide example estimation trajectories. Including sample time‑series from each dataset with baseline and proposed model estimates would illustrate data characteristics and model behavior.
* Suggestion: The paragraph about "controllable text generation" in the Introduction (starting ln. 59) does not seem directly relevant to the problem at hand. The authors could reconsider removing it.

---

> ### Author Response · Authors · 2025-11-21
> **Rebuttal (Part 1 of 7)**
>
> Thank you for your insightful comments and valuable feedback! We appreciate your acknowledgement that the “introduction motivates the problem well,” “Fig. 1 is informative,” the “writing is generally clear,” and that the “figures are descriptive and polished.” We are happy that you have found that the “model design is simple and clean” and that “the method is evaluated on a substantial number of datasets from three domains.”
>
> We have carefully addressed your questions regarding the clarification of (Q1) baselines, (Q2) the performance reported in the abstract, (Q3) the task definitions of R4 versus R5, and (Q4) the intervention mechanics in R6. We have also addressed your minor questions on (Q5) clarifying the choice of concept encoder and decoder architectures, (Q6) providing a few sampled trajectories for qualitative analysis, (Q7) removing a portion of the introduction, and (Q8) improving figure readability.
>
> We hope that our responses answer all of your questions, and we kindly ask you to consider raising your score. Please let us know if you still have concerns after reviewing our responses. Thank you again!
>
> ---
>
> # Q1: Clarification of baselines
>
> To clarify the implementation of CLEF and non-CLEF models: **CLEF models share the same sequence encoder and condition adapter** as their non-CLEF counterparts but **replace the forecasting decoder with a concept encoder and decoder** to leverage temporal concepts. We use the implementation of the Transformer layer from PyTorch, xLSTM from the original paper (Beck et al. 2024), and MOMENT from the original paper (Goswami et al. 2024). To ensure that the models are comparable, CLEF and non-CLEF models have similar architectures (except for the forecasting decoder and concept encoder/decoder), number of parameters (Appendix Table 2), and time complexity (Appendix Table 3).
>
> Experiments show that **adding temporal concepts to these SOTA architectures improves both performance (Figure 4; Appendix Figures 8-9) and generalizability (Appendix Tables 2; Appendix Figures 11-12)**. As you noted, the SimpleLinear baseline outperforms even the best non-CLEF models while performing slightly worse than the best CLEF models. These findings suggest that the SOTA architectures are insufficient for controllable sequence editing on cellular, patient, and sales trajectories. While there is room for improvement, CLEF demonstrates stronger performance than non-CLEF models with SOTA architectures and comparable to or better than SimpleLinear.
>
> **On the applicability of CLEF models:** In addition to yielding strong predictive performance and generalizability on immediate and delayed sequence editing, temporal concepts improve counterfactual prediction (Figure 5; Appendix Figures 13-16), enable stronger zero-shot conditional generation of counterfactual sequences (Figure 6; Appendix Figure 17), and allow direct edits to the generated outputs via temporal concept intervention to produce counterfactual sequences (Figure 7; Appendix Figure 18). Models that do not learn temporal concepts lack the capability to perform temporal concept intervention.
>
> ---
>
> # Q2: Clarification on reporting performance in the abstract
>
> The performance metrics reported are comparing CLEF models against their non-CLEF counterparts.
>
> - The performance gap reported for immediate sequence editing refers to the comparison between CLEF-xLSTM and xLSTM (non-CLEF) on the M5 dataset.
>
> - The performance gap reported for delayed sequence editing refers to the comparison between CLEF-xLSTM and xLSTM (non-CLEF) on the HVG dataset.
>
> **To make the comparison more clear, we have revised the abstract (L021-L025)** to report the average performance gap between CLEF and non-CLEF models across all datasets. We have also clarified that the comparison is between CLEF and non-CLEF counterparts.
>
> - “CLEF improves immediate sequence editing accuracy by **16.28% (MAE) on average against their non-CLEF counterparts**. Unlike prior models, CLEF enables one-step conditional generation at arbitrary future times, outperforming **their non-CLEF counterparts** in delayed sequence editing by **26.73% (MAE) on average**.”

---

> ### Author Response · Authors · 2025-11-21
> **Rebuttal (Part 2 of 7)**
>
> # Q3: Clarification of task definitions for R4 versus R5
>
> Thank you for your suggestion to clarify the distinction between the task definition for R4 versus R5! The task definition for R4 and R5 are similar: Predict the outcomes of a counterfactual action on the trajectory. **The key differences are:**
>
> - R4 evaluates on existing benchmark datasets (tumor growth and semi-synthetic patient ICU data) with a small number of actions/conditions (the tumor growth dataset has four possible actions/conditions, and the semi-synthetic patient ICU dataset has three possible actions/conditions). Each of these actions/conditions is observed in at least one factual trajectory during training. On the other hand, R5 evaluates models on our new dataset, WOT-CF, that consists of paired factual-counterfactual cellular trajectories. There are 297 unique actions/conditions, and each of the actions/conditions is not guaranteed to be observed in at least one factual trajectory during training.
>
> - R4 demonstrates the ability to leverage temporal concepts for improving the performance of CT and CRN in counterfactual outcomes estimation. On the other hand, R5 showcases the ability to leverage temporal concepts for enabling any conditional generation model to perform zero-shot conditional generation of counterfactual trajectories (under the counterfactual assumptions in Section 3.3).
>
> ---
>
> # Q4: Clarification of intervention mechanics in R6 (Part 1/2)
>
> Thank you for your suggestions to clarify how the interventions on temporal concepts are performed, and to compare the CLEF-generated trajectories against the SimpleLinear baseline.
>
> Formally, given temporal concept $c$ learned from $x_{:,t_0:t_i}$ and an optional condition $s$, we modify $c^I \neq c$ such that at least one element satisfies $c_k \neq c_k^I$, where $k$ represents the temporal concept for glucose.
> - To increase the glucose levels by 2x, $c_k = 2$.
> - To decrease the glucose levels by 0.5x, $c_k = 0.5$.
>
> ## eICU
>
> We modify CLEF’s and SimpleLinear’s concepts to halve glucose levels, aligning them closer to normal physiological ranges. Such counterfactual trajectories should exhibit higher $R^2$ similarity with healthy individuals compared to other T1D patients.
>
> **Table 1: Decrease glucose levels by 0.5x.** We compare CLEF-generated trajectories against observed trajectories of matched healthy, other healthy, and other type 1 diabetic (T1D) patients. Higher $R^2$ indicates that patient pairs are more similar.
>
> | Model | Matched Healthy | Other Healthy | Other T1D |
> | --- | --- | --- | --- |
> | CLEF | 0.757 +/- 0.227 | 0.737 +/- 0.126 | 0.600 +/- 0.122 |
> | SimpleLinear | 0.403 +/- 0.226 |  0.390 +/- 0.233 | 0.288 +/- 0.109 |
>
> CLEF’s generated trajectories are more similar to those of the matched healthy ($R^2 = 0.757 +/- 0.227$) and other healthy ($R^2 = 0.737 +/- 0.126$) patients than other T1D patients ($R^2 = 0.600 +/- 0.122$). On the other hand, SimpleLinear’s generated trajectories have low similarity to those of matched healthy ($R^2 = 0.403 +/- 0.226$), other healthy ($R^2 = 0.390 +/- 0.233$), and other T1D patients ($R^2 = 0.288 +/- 0.109$). Also, the $R^2$ between SimpleLinear’s generated trajectories and the trajectories of matched healthy, other healthy, and other T1D patients are comparable. On the other hand, the $R^2$ between CLEF’s generated trajectories and healthy trajectories are significantly higher than the $R^2$ between CLEF’s generated trajectories and other T1D patients’ trajectories.
>
> Next, we simulate a worsening condition by doubling glucose levels. Such counterfactual trajectories should exhibit higher $R^2$ similarity with other T1D patients compared to healthy individuals.
>
> **Table 2: Increase glucose levels by 2x.** We compare CLEF-generated trajectories against observed trajectories of matched healthy, other healthy, and other type 1 diabetic (T1D) patients. Higher $R^2$ indicates that patient pairs are more similar.
>
> | Model | Matched Healthy | Other Healthy | Other T1D |
> | --- | --- | --- | --- |
> | CLEF | 0.147 +/- 0.111 | 0.175 +/- 0.106 | 0.348 +/- 0.149 |
> | SimpleLinear | 0.287 +/- 0.141 | 0.263 +/- 0.159 | 0.240 +/- 0.087 |
>
> CLEF’s generated trajectories are more similar to those of the other T1D patients ($R^2 = 0.348 +/- 0.149$) than matched healthy ($R^2 = 0.147 +/- 0.111$) and other healthy ($R^2 = 0.175 +/- 0.106$) patients. On the other hand, SimpleLinear’s generated trajectories have consistent and low similarity to those of matched healthy ($R^2 = 0.287 +/- 0.141$), other healthy ($R^2 = 0.263 +/- 0.159$), and other T1D ($R^2 = 0.240 +/- 0.087$) patients. Also, the $R^2$ between SimpleLinear’s generated trajectories and the trajectories of the matched healthy, other healthy, and other T1D patients are comparable. On the other hand, the $R^2$ between CLEF’s generated trajectories and healthy trajectories are significantly lower than the $R^2$ between CLEF’s generated trajectories and other T1D patients’.

---

> ### Author Response · Authors · 2025-11-21
> **Rebuttal (Part 3 of 7)**
>
> # Q4: Clarification of intervention mechanics in R6 (Part 2/2)
>
> (Continued)
>
> ## MIMIC-IV
>
> We modify CLEF’s and SimpleLinear’s concepts to halve glucose levels, aligning them closer to normal physiological ranges. Such counterfactual trajectories should exhibit higher $R^2$ similarity with healthy individuals compared to other T1D patients.
>
> **Table 1: Decrease glucose levels by 0.5x.** We compare CLEF-generated trajectories against observed trajectories of matched healthy, other healthy, and other type 1 diabetic (T1D) patients. Higher $R^2$ indicates that patient pairs are more similar.
>
> | Model | Matched Healthy | Other Healthy | Other T1D |
> | --- | --- | --- | --- |
> | CLEF | 0.838 +/- 0.140 | 0.814 +/- 0.076 | 0.692 +/- 0.071 |
> | SimpleLinear | 0.713 +/- 0.166 | 0.689 +/- 0.113 | 0.564 +/- 0.082 |
>
> Both CLEF’s and SimpleLinear’s trajectories are more similar to those of the matched healthy and other healthy patients than other T1D patients.
>
> Next, we simulate a worsening condition by doubling glucose levels. Such counterfactual trajectories should exhibit higher $R^2$ similarity with other T1D patients compared to healthy individuals.
>
> **Table 2: Increase glucose levels by 2x.** We compare CLEF-generated trajectories against observed trajectories of matched healthy, other healthy, and other type 1 diabetic (T1D) patients. Higher $R^2$ indicates that patient pairs are more similar.
>
> | Model | Matched Healthy | Other Healthy | Other T1D |
> | --- | --- | --- | --- |
> | CLEF | 0.364 +/- 0.126 | 0.344 +/- 0.129 | 0.472 +/- 0.134 |
> | SimpleLinear | 0.790 +/- 0.110 | 0.709 +/- 0.051 | 0.541 +/- 0.056 |
>
> CLEF’s generated trajectories are more similar to those of the other T1D patients ($R^2 = 0.472 +/- 0.134$) than matched healthy ($R^2 = 0.364 +/- 0.126$) and other healthy ($R^2 = 0.344 +/- 0.129$) patients, which is expected. On the other hand, SimpleLinear’s trajectories are more similar to those of the matched healthy ($R^2 = 0.790 +/- 0.110$) and other healthy ($R^2 = 0.709 +/- 0.051$) patients than other T1D patients ($R^2 = 0.541 +/- 0.056$), which is unexpected.
>
> **Unlike CLEF, SimpleLinear cannot generate trajectories that resemble the trajectories of healthier or sicker patients, depending on the intervention.**
>
> ---
>
> # Q5: Choice of concept encoder and decoder architectures (Part 1/3)
>
> Thank you for your question about our choice of concept decoder and decoder architectures! **We respond in two parts:**
>
> 1. We provide an intuition about the design of temporal concepts, which informs our choice of concept encoder and decoder architectures.
>
> 2. We perform an ablation experiment with an alternative concept decoder to evaluate the strengths and weaknesses of our concept decoder architecture.
>
>
>
> ## Intuition about the design of temporal concepts
>
> We intentionally design temporal concepts to isolate their contribution to predictive performance.
>
> - In our formulation, temporal concepts are learned from an aggregation of historical data, the future time point, and the desired condition by concept encoder $E$ (Equation 2). We evaluate multiple ways of defining $E$: via the state-of-the-art setups for conditional sequence generation (Appendix D3) and counterfactual outcomes estimation (Appendix D4) with different sequential encoders.
>
> - Temporal concepts are applied directly to the latest time step in the concept decoder $F$ (Equation 3) to generate the future state. Alternative model architectural designs for learning temporal concepts and applying conditions (or interventions) to the model may obfuscate the contribution of temporal concepts to predictive performance. For example, feeding the intervention directly to the decoder would bypass the temporal concept mechanism, meaning that the concepts would capture only the passage of time rather than the effect of the intervention. Similarly, feeding the future state directly to the decoder would introduce an additional function applied after the temporal concepts, making it difficult to directly control the edit by the specified concept (because of the add-on decoder).

---

> > ### Author Response · Authors · 2025-11-21
> > **Rebuttal (Part 4 of 7)**
> >
> > # Q5: Choice of concept encoder and decoder architectures (Part 2/3)
> >
> > (Continued)
> >
> > ## Ablation experiments (Part 1/2)
> >
> > We have implemented a decoder $(I + W)(c \odot x)$ where $W$ is low-rank, with rank $= 4, 8, 16$. We refer to models with this decoder using the prefix “LowR.” We evaluate LowR models against CLEF and non-CLEF models on WOT, eICU, MIMIC-IV, and M5 for **immediate sequence editing**.
> >
> >
> > | Dataset | Model | MAE |
> > | --- | --- | --- |
> > | MIMIC-IV | Transformer                  |  7.284 +/- 0.005 |
> > | | LowR-Transformer (Rank = 4)  |  6.673 +/- 0.074 |
> > | | LowR-Transformer (Rank = 8)  |  6.748 +/- 0.067 |
> > | | LowR-Transformer (Rank = 16) |  6.724 +/- 0.155 |
> > | | **CLEF-Transformer (FFN = 0)**   |  **6.577 +/- 0.038** |
> > | | **CLEF-Transformer (FFN = 1)**   |  **6.562 +/- 0.046** |
> > | MIMIC-IV | xLSTM                  |  7.561 +/- 0.354 |
> > | | LowR-xLSTM (Rank = 4)  |  7.018 +/- 0.162 |
> > | | LowR-xLSTM (Rank = 8)  |  7.139 +/- 0.402 |
> > | | LowR-xLSTM (Rank = 16) |  7.031 +/- 0.058 |
> > | | CLEF-xLSTM (FFN = 0)   |  7.493 +/- 0.722 |
> > | | **CLEF-xLSTM (FFN = 1)**   |  **6.942 +/- 0.052** |
> > | MIMIC-IV | MOMENT                  |  14.804 +/- 0.036 |
> > | | LowR-MOMENT (Rank = 4)  |  10.671 +/- 0.290 |
> > | | LowR-MOMENT (Rank = 8)  |  10.603 +/- 0.093 |
> > | | LowR-MOMENT (Rank = 16) |  10.685 +/- 0.204 |
> > | | **CLEF-MOMENT (FFN = 0)**   |  **9.779 +/- 0.148** |
> > | | **CLEF-MOMENT (FFN = 1)**   |  **9.579 +/- 0.199** |
> > | eICU | Transformer                   | 9.439 +/- 0.082 |
> > | | **LowR-Transformer (Rank = 4)**   | **8.135 +/- 0.102** |
> > | | **LowR-Transformer (Rank = 8)**   | **8.265 +/- 0.109** |
> > | | LowR-Transformer (Rank = 16)  | 8.426 +/- 0.162 |
> > | | **CLEF-Transformer (FFN = 0)**    | **8.319 +/- 0.038** |
> > | | **CLEF-Transformer (FFN = 1)**    | **8.338 +/- 0.064** |
> > | eICU | xLSTM                  | 8.041 +/- 0.142 |
> > | | **LowR-xLSTM (Rank = 4)**  | **7.731 +/- 0.054** |
> > | | LowR-xLSTM (Rank = 8)  | 8.018 +/- 0.062 |
> > | | LowR-xLSTM (Rank = 16) | 8.001 +/- 0.111 |
> > | | **CLEF-xLSTM (FFN = 0)**   | **7.815 +/- 0.139** |
> > | | **CLEF-xLSTM (FFN = 1)**   | **7.751 +/- 0.050** |
> > | eICU | MOMENT                  | 13.376 +/- 0.089 |
> > | | LowR-MOMENT (Rank = 4)  | 10.405 +/- 0.099 |
> > | | LowR-MOMENT (Rank = 8)  | 10.256 +/- 0.252 |
> > | | LowR-MOMENT (Rank = 16) | 11.729 +/- 1.460 |
> > | | **CLEF-MOMENT (FFN = 0)**   | **9.955 +/- 0.164** |
> > | | CLEF-MOMENT (FFN = 1)   | 10.111 +/- 0.329 |
> > | WOT | Transformer                   | 0.360 +/- 0.003 |
> > | | LowR-Transformer (Rank = 4)   | 0.365 +/- 0.001 |
> > | | LowR-Transformer (Rank = 8)   | 0.367 +/- 0.001 |
> > | | LowR-Transformer (Rank = 16)  | 0.366 +/- 0.002 |
> > | | CLEF-Transformer (FFN = 0)    | 0.360 +/- 0.001 |
> > | | **CLEF-Transformer (FFN = 1)**    | **0.344 +/- 0.001** |
> > | WOT | xLSTM                  | 0.373 +/- 0.004 |
> > | | LowR-xLSTM (Rank = 4)  | 0.362 +/- 0.002 |
> > | | LowR-xLSTM (Rank = 8)  | 0.363 +/- 0.001 |
> > | | LowR-xLSTM (Rank = 16) | 0.365 +/- 0.001 |
> > | | CLEF-xLSTM (FFN = 0)   | 0.350 +/- 0.001 |
> > | | **CLEF-xLSTM (FFN = 1)**   | **0.348 +/- 0.001** |
> > | WOT | MOMENT                  | 0.386 +/- 0.002 |
> > | | LowR-MOMENT (Rank = 4)  | 0.370 +/- 0.003 |
> > | | LowR-MOMENT (Rank = 8)  | 0.372 +/- 0.001 |
> > | | LowR-MOMENT (Rank = 16) | 0.381 +/- 0.001 |
> > | | **CLEF-MOMENT (FFN = 0)**   | **0.356 +/- 0.001** |
> > | | **CLEF-MOMENT (FFN = 1)**   | **0.356 +/- 0.001** |
> > | M5 | Transformer                  | 1.203 +/- 0.014 |
> > | | LowR-Transformer (Rank = 4)  | 1.137 +/- 0.036 |
> > | | LowR-Transformer (Rank = 8)  | 1.136 +/- 0.015 |
> > | | LowR-Transformer (Rank = 16) | 1.151 +/- 0.011 |
> > | | **CLEF-Transformer (FFN = 0)**   | **1.089 +/- 0.001** |
> > | | **CLEF-Transformer (FFN = 1)**   | **1.086 +/- 0.000** |
> > | M5 | xLSTM                  | 1.306 +/- 0.032 |
> > | | LowR-xLSTM (Rank = 4)  | 1.029 +/- 0.011 |
> > | | LowR-xLSTM (Rank = 8)  | 1.039 +/- 0.007 |
> > | | LowR-xLSTM (Rank = 16) | 1.077 +/- 0.028 |
> > | | CLEF-xLSTM (FFN = 0)   | 1.025 +/- 0.008 |
> > | | **CLEF-xLSTM (FFN = 1)**   | **0.845 +/- 0.006** |
> > | M5 | MOMENT                  | 1.156 +/- 0.043 |
> > | | LowR-MOMENT (Rank = 4)  | 0.835 +/- 0.019 |
> > | | LowR-MOMENT (Rank = 8)  | 0.849 +/- 0.006 |
> > | | LowR-MOMENT (Rank = 16) | 0.875 +/- 0.023 |
> > | | **CLEF-MOMENT (FFN = 0)**   | **0.786 +/- 0.003** |
> > | | **CLEF-MOMENT (FFN = 1)**   | **0.778 +/- 0.015** |
> >
> > We find that CLEF models yield comparable or lower MAE (i.e., better) than non-CLEF and LowR models across all datasets. These results suggest that CLEF’s concept decoder, $c \odot x$, is **appropriate for capturing the dynamics of the trajectories. To supplement these experiments, we have noted the limitations of this decoder and opportunities for future improvements.**

---

> > > ### Author Response · Authors · 2025-11-21
> > > **Rebuttal (Part 5 of 7)**
> > >
> > > # Q5: Choice of concept encoder and decoder architectures (Part 3/3)
> > >
> > > (Continued)
> > >
> > > ## Ablation experiments (Part 2/2)
> > >
> > > (Continued) We evaluate LowR models against CLEF and non-CLEF models on WOT, eICU, MIMIC-IV, and M5 for **delayed sequence editing**.
> > >
> > >
> > >
> > > | Dataset | Model | MAE |
> > > | --- | --- | --- |
> > > | MIMIC-IV | Transformer                        | 8.346 +/- 0.129 |
> > > | | **LowR-Transformer (Rank = 4)**        | **7.818 +/- 0.045** |
> > > | | LowR-Transformer (Rank = 8)        | 7.943 +/- 0.041 |
> > > | | LowR-Transformer (Rank = 16)       | 7.928 +/- 0.027 |
> > > | | CLEF-Transformer (FFN = 0)         | 7.915 +/- 0.046 |
> > > | | **CLEF-Transformer (FFN = 1)**         | **7.863 +/- 0.065** |
> > > | MIMIC-IV | xLSTM                      | 7.678 +/- 0.029 |
> > > | | **LowR-xLSTM (Rank = 4)**      | **7.742 +/- 0.075** |
> > > |  | LowR-xLSTM (Rank = 8)      | 7.853 +/- 0.026 |
> > > |  | LowR-xLSTM (Rank = 16)     | 7.823 +/- 0.025 |
> > > |  | CLEF-xLSTM (FFN = 0)       | 8.420 +/- 0.692 |
> > > |  | **CLEF-xLSTM (FFN = 1)**       | **7.794 +/- 0.042** |
> > > | MIMIC-IV | MOMENT                    | 10.807 +/- 0.054 |
> > > |  | LowR-MOMENT (Rank = 4)    | 8.928 +/- 0.362 |
> > > |  | LowR-MOMENT (Rank = 8)    | 8.926 +/- 0.221 |
> > > |  | LowR-MOMENT (Rank = 16)   | 9.001 +/- 0.493 |
> > > |  | **CLEF-MOMENT (FFN = 0)**     | **8.688 +/- 0.074** |
> > > |  | **CLEF-MOMENT (FFN = 1)**     | **8.563 +/- 0.182** |
> > > | eICU | Transformer                      | 8.377 +/- 0.047 |
> > > |  | **LowR-Transformer (Rank = 4)**      | **7.465 +/- 0.201** |
> > > |  | **LowR-Transformer (Rank = 8)**      | **7.567 +/- 0.070** |
> > > |  | LowR-Transformer (Rank = 16)     | 7.788 +/- 0.128 |
> > > |  | CLEF-Transformer (FFN = 0)       | 7.643 +/- 0.083 |
> > > |  | **CLEF-Transformer (FFN = 1)**       | **7.566 +/- 0.040** |
> > > | eICU | xLSTM                     | 7.086 +/- 0.066 |
> > > |  | **LowR-xLSTM (Rank = 4)**     | **7.253 +/- 0.023** |
> > > |  | LowR-xLSTM (Rank = 8)     | 7.375 +/- 0.094 |
> > > |  | LowR-xLSTM (Rank = 16)    | 7.414 +/- 0.112 |
> > > |  | CLEF-xLSTM (FFN = 0)      | 7.324 +/- 0.048 |
> > > |  | **CLEF-xLSTM (FFN = 1)**      | **7.241 +/- 0.082** |
> > > | eICU | MOMENT                    | 10.289 +/- 0.086 |
> > > |  | LowR-MOMENT (Rank = 4)    | 8.873 +/- 0.157 |
> > > |  | LowR-MOMENT (Rank = 8)    | 9.074 +/- 0.352 |
> > > |  | LowR-MOMENT (Rank = 16)   | 9.608 +/- 1.133 |
> > > |  | CLEF-MOMENT (FFN = 0)     | 8.773 +/- 0.120 |
> > > |  | **CLEF-MOMENT (FFN = 1)**     | **8.472 +/- 0.259** |
> > > | WOT | Transformer                      | 1.273 +/- 0.022 |
> > > |  | LowR-Transformer (Rank = 4)      | 0.552 +/- 0.049 |
> > > |  | LowR-Transformer (Rank = 8)      | 0.547 +/- 0.003 |
> > > |  | LowR-Transformer (Rank = 16)     | 0.525 +/- 0.057 |
> > > |  | **CLEF-Transformer (FFN = 0)**       | **0.494 +/- 0.045** |
> > > |  | CLEF-Transformer (FFN = 1)       | 0.748 +/- 0.035 |
> > > | WOT | xLSTM                      | 1.592 +/- 0.022 |
> > > |  | LowR-xLSTM (Rank = 4)      | 0.857 +/- 0.086 |
> > > |  | LowR-xLSTM (Rank = 8)      | 0.854 +/- 0.047 |
> > > |  | LowR-xLSTM (Rank = 16)     | 0.882 +/- 0.048 |
> > > |  | **CLEF-xLSTM (FFN = 0)**       | **0.724 +/- 0.016** |
> > > |  | CLEF-xLSTM (FFN = 1)       | 1.028 +/- 0.030 |
> > > | WOT | MOMENT                      | 0.836 +/- 0.018 |
> > > |  | LowR-MOMENT (Rank = 4)      | 0.534 +/- 0.010 |
> > > |  | LowR-MOMENT (Rank = 8)      | 0.542 +/- 0.005 |
> > > |  | LowR-MOMENT (Rank = 16)     | 0.576 +/- 0.003 |
> > > |  | **CLEF-MOMENT (FFN = 0)**       | **0.493 +/- 0.002** |
> > > |  | CLEF-MOMENT (FFN = 1)       | 0.512 +/- 0.009 |
> > > | M5 | Transformer                       | 1.214 +/- 0.016 |
> > > |  | LowR-Transformer (Rank = 4)       | 1.133 +/- 0.036 |
> > > |  | LowR-Transformer (Rank = 8)       | 1.137 +/- 0.010 |
> > > |  | LowR-Transformer (Rank = 16)      | 1.149 +/- 0.012 |
> > > |  | **CLEF-Transformer (FFN = 0)**        | **1.090 +/- 0.001** |
> > > |  | **CLEF-Transformer (FFN = 1)**        | **1.086 +/- 0.000** |
> > > | M5 | xLSTM                      | 1.345 +/- 0.034 |
> > > |  | LowR-xLSTM (Rank = 4)      | 1.067 +/- 0.014 |
> > > |  | LowR-xLSTM (Rank = 8)      | 1.075 +/- 0.013 |
> > > |  | LowR-xLSTM (Rank = 16)     | 1.117 +/- 0.030 |
> > > |  | CLEF-xLSTM (FFN = 0)       | 1.066 +/- 0.009 |
> > > |  | **CLEF-xLSTM (FFN = 1)**       | **0.867 +/- 0.008** |
> > > | M5 | MOMENT                    | 1.162 +/- 0.044 |
> > > |  | LowR-MOMENT (Rank = 4)    | 0.858 +/- 0.015 |
> > > |  | LowR-MOMENT (Rank = 8)    | 0.876 +/- 0.006 |
> > > |  | LowR-MOMENT (Rank = 16)   | 0.907 +/- 0.026 |
> > > |  | **CLEF-MOMENT (FFN = 0)**     | **0.819 +/- 0.004** |
> > > |  | **CLEF-MOMENT (FFN = 1)**     | **0.807 +/- 0.017** |
> > >
> > >
> > > We find that CLEF models yield comparable or lower MAE (i.e., better) than non-CLEF and LowR models across all datasets. These results suggest that CLEF’s concept decoder, $c \odot x$, is **appropriate for capturing the dynamics of the trajectories. To supplement these experiments, we have noted the limitations of this decoder and opportunities for future improvements.**

---

> ### Author Response · Authors · 2025-11-21
> **Rebuttal (Part 6 of 7)**
>
> # Q6: Qualitative trajectories (Part 1/2)
>
> Thank you for your suggestion to include “qualitative trajectory examples”! Due to the high dimensionality of the trajectories, we provide a subset of features (i.e., Hct, Hgb, MCH, MCV, Sodium) for two patient trajectories from the eICU dataset. Given the strong performance of CLEF-xLSTM and xLSTM, we compare their predicted trajectories against the observed trajectories.
>
> ## Patient 1
>
> | Lab | Time                | Observed    | CLEF   | non-CLEF |
> | --- | --- | --- | --- | --- |
> | Hct | 1900-01-03 09:58:00 | 28.200      | 31.664 | 32.347 |
> |  | 1900-01-04 10:09:00 | 26.700      | 32.022 | 32.805 |
> |  | 1900-01-05 09:31:00 | 26.000      | 31.885 | 32.831 |
> |  | 1900-01-06 06:19:00 | 27.500      | 32.025 | 32.863 |
> |  | 1900-01-07 10:00:00 | 27.800      | 31.779 | 32.357 |
> |  | 1900-01-09 09:29:00 | 29.500      | 31.084 | 32.169 |
> |  | 1900-01-10 10:04:00 | 30.600      | 30.771 | 31.962 |
> | Hgb | 1900-01-03 09:58:00 | 9.700 | 10.193 | 10.558 |
> |  | 1900-01-04 10:09:00 | 9.000 | 10.273 | 10.665 |
> |  | 1900-01-05 09:31:00 | 8.700 | 10.230 | 10.541 |
> |  | 1900-01-06 06:19:00 | 9.200 | 10.295 | 10.605 |
> |  | 1900-01-07 10:00:00 | 9.300 | 10.182 | 10.472 |
> |  | 1900-01-09 09:29:00 | 9.700 | 9.938  | 10.391 |
> |  | 1900-01-10 10:04:00 | 9.800 | 9.804  | 10.192 |
> | MCH | 1900-01-03 09:58:00 | 31.700 | 29.848 | 27.969 |
> |  | 1900-01-04 10:09:00 | 31.400 | 29.904 | 27.874 |
> |  | 1900-01-05 09:31:00 | 31.000 | 29.626 | 27.617 |
> |  | 1900-01-06 06:19:00 | 30.700 | 29.649 | 27.596 |
> |  | 1900-01-07 10:00:00 | 31.200 | 29.467 | 27.667 |
> |  | 1900-01-09 09:29:00 | 30.700 | 29.874 | 28.041 |
> |  | 1900-01-10 10:04:00 | 30.100 | 29.315 | 27.816 |
> | MCV | 1900-01-03 09:58:00 | 92.200 | 93.786 | 89.053
> |  | 1900-01-04 10:09:00 | 93.000 | 93.745 | 89.262
> |  | 1900-01-05 09:31:00 | 92.500 | 93.373 | 88.649
> |  | 1900-01-06 06:19:00 | 91.700 | 93.508 | 88.650
> |  | 1900-01-07 10:00:00 | 93.300 | 93.270 | 88.689
> |  | 1900-01-09 09:29:00 | 93.400 | 94.305 | 89.652
> |  | 1900-01-10 10:04:00 | 93.900 | 93.824 | 89.264
> | Sodium | 1900-01-03 09:58:00 | 139.000 | 137.202 | 140.934 |
> |  | 1900-01-04 10:09:00 | 137.000 | 137.157 | 141.779 |
> |  | 1900-01-05 09:31:00 | 138.000 | 136.695 | 141.848 |
> |  | 1900-01-06 06:19:00 | 137.000 | 137.239 | 141.967 |
> |  | 1900-01-07 10:00:00 | 138.000 | 136.807 | 142.173 |
> |  | 1900-01-09 09:29:00 | 137.000 | 137.781 | 142.912 |
> |  | 1900-01-10 10:04:00 | 137.000 | 138.031 | 141.562 |
>
> Based on this sampled trajectory, **CLEF seems to generate lab test values that are closer to the observed lab test values than non-CLEF models**.

---

> ### Author Response · Authors · 2025-11-21
> **Rebuttal (Part 7 of 7)**
>
> # Q6: Qualitative trajectories (Part 2/2)
>
> (Continued)
>
> ## Patient 2
>
> Based on this sampled trajectory, **CLEF seems to generate lab test values that are closer to the observed lab test values than non-CLEF models**.
>
>
> | Lab | Time                | Observed    | CLEF   | non-CLEF |
> | --- | --- | --- | --- | --- |
> | Hct | 1900-01-07 12:31:00 | 34.400 | 32.783 | 32.298 |
> |  | 1900-01-08 10:29:00 | 33.800 | 32.530 | 31.573 |
> |  | 1900-01-09 11:58:00 | 34.500 | 32.110 | 31.828 |
> |  | 1900-01-10 18:08:00 | 36.600 | 32.288 | 31.464 |
> |  | 1900-01-11 11:24:00 | 33.300 | 32.224 | 31.802 |
> |  | 1900-01-12 11:28:00 | 35.100 | 32.138 | 31.043 |
> |  | 1900-01-13 10:21:00 | 36.400 | 32.361 | 31.413 |
> |  | 1900-01-14 11:10:00 | 33.600 | 31.970 | 31.080 |
> |  | 1900-01-15 09:25:00 | 34.300 | 31.684 | 30.439 |
> |  | 1900-01-16 10:14:00 | 33.400 | 31.706 | 31.286 |
> |  | 1900-01-17 09:56:00 | 35.800 | 31.411 | 30.515 |
> |  | 1900-01-18 10:13:00 | 34.400 | 31.610 | 31.164 |
> |  | 1900-01-19 09:25:00 | 32.300 | 31.057 | 30.230 |
> |  | 1900-01-20 10:47:00 | 31.100 | 31.215 | 30.727 |
> |  | 1900-01-21 11:07:00 | 31.300 | 30.864 | 29.923 |
> | Hgb | 1900-01-07 12:31:00 | 11.300 | 10.967 | 10.607 |
> |  | 1900-01-08 10:29:00 | 11.100 | 10.788 | 10.367 |
> |  | 1900-01-09 11:58:00 | 11.500 | 10.629 | 10.446 |
> |  | 1900-01-10 18:08:00 | 12.200 | 10.736 | 10.296 |
> |  | 1900-01-11 11:24:00 | 11.100 | 10.695 | 10.388 |
> |  | 1900-01-12 11:28:00 | 11.400 | 10.674 | 10.203 |
> |  | 1900-01-13 10:21:00 | 12.200 | 10.755 | 10.281 |
> |  | 1900-01-14 11:10:00 | 10.900 | 10.605 | 10.253 |
> |  | 1900-01-15 09:25:00 | 10.900 | 10.527 | 10.008 |
> |  | 1900-01-16 10:14:00 | 10.900 | 10.542 | 10.178 |
> |  | 1900-01-17 09:56:00 | 11.500 | 10.425 | 9.923 |
> |  | 1900-01-18 10:13:00 | 11.000 | 10.495 | 10.108 |
> |  | 1900-01-19 09:25:00 | 10.400 | 10.290 | 9.748 |
> |  | 1900-01-20 10:47:00 | 10.000 | 10.334 | 9.848 |
> |  | 1900-01-21 11:07:00 | 9.800  | 10.198 | 9.611 |
> | MCH | 1900-01-07 12:31:00 | 30.100 | 29.502 | 27.910 |
> |  | 1900-01-08 10:29:00 | 30.000 | 29.252 | 27.722 |
> |  | 1900-01-09 11:58:00 | 30.700 | 29.300 | 27.923 |
> |  | 1900-01-10 18:08:00 | 30.600 | 29.222 | 27.676 |
> |  | 1900-01-11 11:24:00 | 30.200 | 29.299 | 27.440 |
> |  | 1900-01-12 11:28:00 | 29.800 | 29.160 | 27.412 |
> |  | 1900-01-13 10:21:00 | 30.300 | 29.183 | 27.374 |
> |  | 1900-01-14 11:10:00 | 30.100 | 29.106 | 26.954 |
> |  | 1900-01-15 09:25:00 | 29.600 | 29.062 | 27.089 |
> |  | 1900-01-16 10:14:00 | 30.200 | 29.197 | 27.007 |
> |  | 1900-01-17 09:56:00 | 29.900 | 29.024 | 26.864 |
> |  | 1900-01-18 10:13:00 | 29.900 | 29.153 | 26.852 |
> |  | 1900-01-19 09:25:00 | 30.100 | 29.044 | 26.847 |
> |  | 1900-01-20 10:47:00 | 29.900 | 29.046 | 26.571 |
> |  | 1900-01-21 11:07:00 | 29.900 | 28.978 | 26.824 |
> | MCV | 1900-01-07 12:31:00 | 91.700 | 87.581 | 85.512 |
> |  | 1900-01-08 10:29:00 | 91.400 | 88.247 | 84.877 |
> |  | 1900-01-09 11:58:00 | 92.000 | 88.324 | 85.455 |
> |  | 1900-01-10 18:08:00 | 91.700 | 88.097 | 84.901 |
> |  | 1900-01-11 11:24:00 | 90.500 | 88.038 | 84.469 |
> |  | 1900-01-12 11:28:00 | 91.900 | 87.922 | 84.194 |
> |  | 1900-01-13 10:21:00 | 90.500 | 87.972 | 84.065 |
> |  | 1900-01-14 11:10:00 | 92.800 | 87.738 | 83.119 |
> |  | 1900-01-15 09:25:00 | 93.200 | 87.634 | 82.869 |
> |  | 1900-01-16 10:14:00 | 92.500 | 87.803 | 83.296 |
> |  | 1900-01-17 09:56:00 | 93.000 | 87.629 | 82.255 |
> |  | 1900-01-18 10:13:00 | 93.500 | 87.828 | 82.804 |
> |  | 1900-01-19 09:25:00 | 93.400 | 87.821 | 82.391 |
> |  | 1900-01-20 10:47:00 | 92.800 | 87.924 | 82.099 |
> |  | 1900-01-21 11:07:00 | 95.400 | 88.035 | 82.560 |
> | Sodium | 1900-01-07 12:31:00 | 135.000 | 136.593 | 139.930 |
> |  | 1900-01-08 10:29:00 | 137.000 | 135.900 | 139.291 |
> |  | 1900-01-09 11:58:00 | 136.000 | 137.241 | 141.514 |
> |  | 1900-01-10 18:08:00 | 136.000 | 137.631 | 142.512 |
> |  | 1900-01-11 11:24:00 | 136.000 | 138.199 | 142.694 |
> |  | 1900-01-12 11:28:00 | 135.000 | 138.312 | 143.094 |
> |  | 1900-01-13 10:21:00 | 133.000 | 138.811 | 142.580 |
> |  | 1900-01-14 11:10:00 | 135.000 | 138.777 | 141.410 |
> |  | 1900-01-15 09:25:00 | 132.000 | 138.393 | 141.368 |
> |  | 1900-01-16 10:14:00 | 134.000 | 139.036 | 141.228 |
> |  | 1900-01-17 09:56:00 | 135.000 | 138.747 | 139.891 |
> |  | 1900-01-18 10:13:00 | 137.000 | 138.992 | 140.015 |
> |  | 1900-01-19 09:25:00 | 136.000 | 138.943 | 139.224 |
> |  | 1900-01-20 10:47:00 | 135.000 | 138.693 | 138.322 |
> |  | 1900-01-21 11:07:00 | 138.000 | 138.965 | 138.743 |
>
> ---
>
> # Q7: Revise introduction
>
> Thank you for your suggestion to revise the “controllable text generation” paragraph. We have moved it entirely to the Appendix’s related works section. Because controllable sequence generation has been most widely demonstrated with textual data, we have left a brief discussion in the introduction.
>
> ---
>
> # Q8: Improve figure readability
>
> Thank you for your suggestion to improve figure readability! The size of the figures is due to the space constraints of the submission. We have increased the size of the figures to improve readability.

---

> > ### Author Response · Authors · 2025-11-25
> > **Reminder: Following up on our rebuttal**
> >
> > Dear Reviewer hatS,
> >
> > We sincerely appreciate your insightful comments and the time you have taken to review our work. In response, we have carefully addressed each of your concerns and made the necessary revisions. We kindly ask if these revisions meet your expectations and if there are any further concerns or feedback you would like to discuss. We are more than happy to continue the conversation and address any additional points you may have.
> >
> > Thank you once again for your time and thoughtful consideration.
> >
> > Best regards,
> >
> > Authors of Submission #13156

---

### Official Review · Reviewer_v6m5 · 2025-11-01

**Soundness:** 3
**Presentation:** 3
**Contribution:** 3
**Rating:** 6
**Confidence:** 3

**Summary:**

The paper studies the problem of sequence editing and forecasting. The paper proposes a new method to achieve this goal. The proposed method is based on encoder decoder solution. Given a condition, the proposed method changes the sequence to meet the given condition. The paper splits the problem into two subproblems: immediate sequence editing and delayed sequence editing. Experiments are conducted on variety of biomedical and financial datasets.

**Strengths:**

- The paper addresses an interesting problem with significant practical applications.
- The paper is well-written and the proposed method is clearly motivated.
- Experimental section is thorough and convincing.

**Weaknesses:**

- The related work section could be expanded. Certain areas, such as reinforcement learning, are closely related to this problem but are not discussed in the current version.
- The paper could also benefit from including more baselines by adapting other existing and closely related methods to the experimental setup. For example, some methods proposed for biological, protein, and DNA sequence editing could be applied to the problem studied in this paper, even though they were originally designed to address slightly different tasks in other biological domains. However, this is not necessary for this paper whereas it can improve its strengths'.

**Questions:**

- After reading the paper, it is not entirely clear how the proposed solution and results differ between immediate sequence editing and delayed sequence editing. Could you elaborate on this? My understanding is that immediate sequence editing can be viewed as a special case of delayed sequence editing.
- There are existing approaches originally proposed for biological sequence design that could be applied to this problem. For example, Bayesian optimization such as the one introduced in “Accelerating Bayesian optimization for biological sequence design with denoising
autoencoders” (Stanton et al., 2022) could be considered. Another possible direction is to model the problem using reinforcement learning, where $x$ represents the state and $s$ denotes the value of that state. Note that the term state in this paper may differ from its usage in reinforcement learning. Furthermore, GFlowNets have been widely applied in biological sequence design and specifically in sequence editing as introduced by “GFlowNet-Assisted Biological Sequence Editing” (Ghari et al., 2024) and multi-objective generation as introduced by " Multi-objective GflowNets" (Jain et al., 2023). Although direct comparison with these methods is not necessary for this paper, I recommend including a discussion of such approaches to motivate future research.

---

> ### Author Response · Authors · 2025-11-21
> **Rebuttal**
>
> Thank you for your insightful comments and valuable feedback! We appreciate your acknowledgement that our paper “addresses an interesting problem with significant practical applications” and “is well-written and the proposed method is clearly motivated.” We are happy that you have found that the “experimental section is thorough and convincing.”
>
> We have carefully addressed your questions regarding (Q1) immediate versus delayed sequence editing and (Q2) an extended literature review.
>
> We hope that our responses answer all of your questions, and we kindly ask you to consider raising your score. Please let us know if you still have concerns after reviewing our responses. Thank you again!
>
> ---
>
> # Q1: Clarification on immediate versus delayed sequence editing
>
> Thank you for your question! To clarify the difference between immediate and delayed sequence editing:
>
> - Immediate sequence editing entails generating $t_j$ given data from $t_0$ to $t_i$, where $j = i + 1$
>
> - Delayed sequence editing entails generating $t_j$ given data from $t_0$ to $t_i$, where $j > i$.
>
> This means that immediate sequence editing can be viewed as a special case of delayed sequence editing, where $j = i + 1$.
>
> One of the benefits of delayed sequence editing is the ability to perform a single-step generation of any time $t_j$ in the future. To empirically support this claim, we perform an experiment in which we arbitrarily add three intermediate steps between $t_i$ and $t_j$ before finally predicting the observed $x_{t_j}$. Because we only have the ground truth for $x_{t_j}$, we evaluate only on the predicted $x_{t_j}$.
>
> | Dataset | Model | --- |
> | --- | --- | --- |
> | MIMIC-IV | Transformer                        | 8.346 +/- 0.129 |
> |  | Transformer(intermediate)          | 8.627 +/- 0.365 |
> |  | **CLEF-Transformer**         | **7.915 +/- 0.046** |
> |  | CLEF-Transformer(intermediate)     | 8.279 +/- 0.135 |
> | MIMIC-IV | **xLSTM**                      | **7.678 +/- 0.029** |
> |  | xLSTM(intermediate)        | 8.281 +/- 0.414 |
> |  | **CLEF-xLSTM**       | **8.420 +/- 0.692** |
> |  | CLEF-xLSTM(intermediate)   | 8.684 +/- 0.767 |
> | MIMIC-IV | MOMENT                    | 10.807 +/- 0.054 |
> |  | MOMENT(intermediate)      | 10.673 +/- 0.059 |
> |  | **CLEF-MOMENT**     | **8.688 +/- 0.074** |
> |  | CLEF-MOMENT(intermediate) | 9.096 +/- 0.137 |
> | eICU | Transformer                      | 8.377 +/- 0.047 |
> |  | Transformer(intermediate)        | 9.707 +/- 1.646 |
> |  | **CLEF-Transformer**       | **7.643 +/- 0.083** |
> |  | CLEF-Transformer(intermediate)   | 8.144 +/- 0.247 |
> | eICU | **xLSTM**                     | **7.086 +/- 0.066** |
> |  | xLSTM(intermediate)       | 8.015 +/- 0.566 |
> |  | **CLEF-xLSTM**      | **7.324 +/- 0.048** |
> |  | CLEF-xLSTM(intermediate)  | 8.121 +/- 0.371 |
> | eICU | MOMENT                    | 10.289 +/- 0.086 |
> |  | MOMENT(intermediate)      | 10.233 +/- 0.069 |
> |  | **CLEF-MOMENT**     | **8.773 +/- 0.120** |
> |  | CLEF-MOMENT(intermediate) | 9.591 +/- 0.198 |
>
> As expected, we find that **adding intermediate steps leads to compounding autoregressive error**. These results suggest that delayed sequence editing (i.e., single-step generation) can avoid compounding autoregressive error.
>
> ---
>
> # Q2: Extended literature review
>
> Thank you for your valuable suggestion to extend our literature review to further distinguish CLEF from existing work!
>
> To clarify, biological sequence design methods (e.g., Stanton et al., 2022, Ghari et al., 2024, Jain et al., 2023) are not comparable to CLEF because there is no temporal aspect in the data. The “sequence editing” by these models focuses on positional changes, not temporal. These methods are more related to text generation, where time is not a requirement. **We have added a brief discussion in our related works (Appendix) that clarifies these models’ distinction from CLEF. We have also mentioned as future work the opportunity to extend CLEF for spatial/positional and temporal editing of trajectories.**
>
> In contrast to reinforcement learning approaches, CLEF is trained via a fundamentally different objective. The objective of reinforcement learning is to learn a policy (i.e., output the optimal next action given the current/history of states). On the other hand, CLEF is akin to a forward transition model (i.e., predict future state given history of states and actions/conditions). **CLEF is complementary to reinforcement learning. We have added a brief discussion in our related works to clarify this distinction.**

---

> > ### Author Response · Authors · 2025-11-25
> > **Reminder: Following up on our rebuttal**
> >
> > Dear Reviewer v6m5,
> >
> > We sincerely appreciate your insightful comments and the time you have taken to review our work. In response, we have carefully addressed each of your concerns and made the necessary revisions. We kindly ask if these revisions meet your expectations and if there are any further concerns or feedback you would like to discuss. We are more than happy to continue the conversation and address any additional points you may have.
> >
> > Thank you once again for your time and thoughtful consideration.
> >
> > Best regards,
> >
> > Authors of Submission #13156

---

### Official Review · Reviewer_QAjY · 2025-11-02

**Soundness:** 3
**Presentation:** 3
**Contribution:** 3
**Rating:** 4
**Confidence:** 2

**Summary:**

The authors propose CLEF, a new framework for Controllable Sequence Editing, designed to modify longitudinal sequences (e.g., patient health trajectories, cellular development) based on a specified condition. [cite_start]The key contributions are twofold: 1) a new problem formulation, "delayed sequence editing," which involves applying a condition at a future time and generating the output in a single step, and 2) the CLEF model, which learns "temporal concepts" to represent the change from the last observed time to the future time.

The core mechanism of CLEF is to predict a future state as an element-wise product of the last observed state and the learned temporal concept. This concept $c$ is generated by a "concept encoder" that takes the encoded history ($h_x$), the encoded condition ($h_s$), and the time delta ($\Delta_{t_i, t_j}$) as input.

**Strengths:**

- The formalization of "delayed sequence editing"  as a one-step generation task is interesting contribution. This distinguishes the problem from standard auto-regressive forecasting
- Good empirical results CLEF-based models demonstrate consistent and often large improvements over their non-CLEF counterparts across all primary tasks: immediate editing , delayed editing , and zero-shot counterfactual generation

**Weaknesses:**

- Does the model assume that the entire, complex dynamic evolution of each specific variable (e.g., a single lab test) over an arbitrary time $\Delta t$ can be modeled as a single multiplicative scaling factor $c$ for that variable? This still seems dynamically and biologically implausible, as it ignores the coupled, differential nature of these systems.
- Btw if that is the case, then if any single variable in the last observed state, $x_{k, t_i}$, is 0, then the predicted value for that variable, $\hat{x}_{k, t_j}^s$, must also be 0 (since $c_k \odot 0 = 0$). This is a severe limitation for any data that is sparse or has variables that can cross zero.

**Questions:**

- See weaknesses
- Also I'd argue that c is more of a ratio than a rate of change

---

> ### Author Response · Authors · 2025-11-21
> **Rebuttal (Part 1 of 4)**
>
> Thank you for your insightful comments and valuable feedback! We appreciate your acknowledgement that “the formalization of ‘delayed sequence editing’ as a one-step generation task is [an] interesting contribution” and that it “distinguishes the problem from standard auto-regressive forecasting.” We are happy that you have found the paper to have “good empirical results,” highlighting that “CLEF-based models demonstrate consistent and often large improvements over their non-CLEF counterparts.”
>
> We have carefully addressed your questions regarding the (Q1) assumptions of the concept decoder and (Q2) limitations of the concept decoder.
>
> We hope that our responses answer all of your questions, and we kindly ask you to consider raising your score. Please let us know if you still have concerns after reviewing our responses. Thank you again!

---

> ### Author Response · Authors · 2025-11-21
> **Rebuttal (Part 2 of 4)**
>
> # Q1: Clarification on the assumptions of the concept decoder (Part 1/2)
>
> Thank you for raising the concern that the temporal concepts $c$ may be “[ignoring] the coupled, differential nature of these systems.” We have designed an ablation with a lightly coupled decoder. Specifically, we have implemented a decoder $(I + W)(c \odot x)$ where $W$ is low-rank, with rank $= 4, 8, 16$. We refer to models with this decoder using the prefix “LowR.” We evaluate LowR models against CLEF and non-CLEF models on WOT, eICU, MIMIC-IV, and M5 for **immediate sequence editing**.
>
> ## Immediate sequence editing
>
> | Dataset | Model | MAE |
> | --- | --- | --- |
> | MIMIC-IV | Transformer                  |  7.284 +/- 0.005 |
> | | LowR-Transformer (Rank = 4)  |  6.673 +/- 0.074 |
> | | LowR-Transformer (Rank = 8)  |  6.748 +/- 0.067 |
> | | LowR-Transformer (Rank = 16) |  6.724 +/- 0.155 |
> | | **CLEF-Transformer (FFN = 0)**   |  **6.577 +/- 0.038** |
> | | **CLEF-Transformer (FFN = 1)**   |  **6.562 +/- 0.046** |
> | MIMIC-IV | xLSTM                  |  7.561 +/- 0.354 |
> | | LowR-xLSTM (Rank = 4)  |  7.018 +/- 0.162 |
> | | LowR-xLSTM (Rank = 8)  |  7.139 +/- 0.402 |
> | | LowR-xLSTM (Rank = 16) |  7.031 +/- 0.058 |
> | | CLEF-xLSTM (FFN = 0)   |  7.493 +/- 0.722 |
> | | **CLEF-xLSTM (FFN = 1)**   |  **6.942 +/- 0.052** |
> | MIMIC-IV | MOMENT                  |  14.804 +/- 0.036 |
> | | LowR-MOMENT (Rank = 4)  |  10.671 +/- 0.290 |
> | | LowR-MOMENT (Rank = 8)  |  10.603 +/- 0.093 |
> | | LowR-MOMENT (Rank = 16) |  10.685 +/- 0.204 |
> | | **CLEF-MOMENT (FFN = 0)**   |  **9.779 +/- 0.148** |
> | | **CLEF-MOMENT (FFN = 1)**   |  **9.579 +/- 0.199** |
> | eICU | Transformer                   | 9.439 +/- 0.082 |
> | | **LowR-Transformer (Rank = 4)**   | **8.135 +/- 0.102** |
> | | **LowR-Transformer (Rank = 8)**   | **8.265 +/- 0.109** |
> | | LowR-Transformer (Rank = 16)  | 8.426 +/- 0.162 |
> | | **CLEF-Transformer (FFN = 0)**    | **8.319 +/- 0.038** |
> | | **CLEF-Transformer (FFN = 1)**    | **8.338 +/- 0.064** |
> | eICU | xLSTM                  | 8.041 +/- 0.142 |
> | | **LowR-xLSTM (Rank = 4)**  | **7.731 +/- 0.054** |
> | | LowR-xLSTM (Rank = 8)  | 8.018 +/- 0.062 |
> | | LowR-xLSTM (Rank = 16) | 8.001 +/- 0.111 |
> | | **CLEF-xLSTM (FFN = 0)**   | **7.815 +/- 0.139** |
> | | **CLEF-xLSTM (FFN = 1)**   | **7.751 +/- 0.050** |
> | eICU | MOMENT                  | 13.376 +/- 0.089 |
> | | LowR-MOMENT (Rank = 4)  | 10.405 +/- 0.099 |
> | | LowR-MOMENT (Rank = 8)  | 10.256 +/- 0.252 |
> | | LowR-MOMENT (Rank = 16) | 11.729 +/- 1.460 |
> | | **CLEF-MOMENT (FFN = 0)**   | **9.955 +/- 0.164** |
> | | CLEF-MOMENT (FFN = 1)   | 10.111 +/- 0.329 |
> | WOT | Transformer                   | 0.360 +/- 0.003 |
> | | LowR-Transformer (Rank = 4)   | 0.365 +/- 0.001 |
> | | LowR-Transformer (Rank = 8)   | 0.367 +/- 0.001 |
> | | LowR-Transformer (Rank = 16)  | 0.366 +/- 0.002 |
> | | CLEF-Transformer (FFN = 0)    | 0.360 +/- 0.001 |
> | | **CLEF-Transformer (FFN = 1)**    | **0.344 +/- 0.001** |
> | WOT | xLSTM                  | 0.373 +/- 0.004 |
> | | LowR-xLSTM (Rank = 4)  | 0.362 +/- 0.002 |
> | | LowR-xLSTM (Rank = 8)  | 0.363 +/- 0.001 |
> | | LowR-xLSTM (Rank = 16) | 0.365 +/- 0.001 |
> | | CLEF-xLSTM (FFN = 0)   | 0.350 +/- 0.001 |
> | | **CLEF-xLSTM (FFN = 1)**   | **0.348 +/- 0.001** |
> | WOT | MOMENT                  | 0.386 +/- 0.002 |
> | | LowR-MOMENT (Rank = 4)  | 0.370 +/- 0.003 |
> | | LowR-MOMENT (Rank = 8)  | 0.372 +/- 0.001 |
> | | LowR-MOMENT (Rank = 16) | 0.381 +/- 0.001 |
> | | **CLEF-MOMENT (FFN = 0)**   | **0.356 +/- 0.001** |
> | | **CLEF-MOMENT (FFN = 1)**   | **0.356 +/- 0.001** |
> | M5 | Transformer                  | 1.203 +/- 0.014 |
> | | LowR-Transformer (Rank = 4)  | 1.137 +/- 0.036 |
> | | LowR-Transformer (Rank = 8)  | 1.136 +/- 0.015 |
> | | LowR-Transformer (Rank = 16) | 1.151 +/- 0.011 |
> | | **CLEF-Transformer (FFN = 0)**   | **1.089 +/- 0.001** |
> | | **CLEF-Transformer (FFN = 1)**   | **1.086 +/- 0.000** |
> | M5 | xLSTM                  | 1.306 +/- 0.032 |
> | | LowR-xLSTM (Rank = 4)  | 1.029 +/- 0.011 |
> | | LowR-xLSTM (Rank = 8)  | 1.039 +/- 0.007 |
> | | LowR-xLSTM (Rank = 16) | 1.077 +/- 0.028 |
> | | CLEF-xLSTM (FFN = 0)   | 1.025 +/- 0.008 |
> | | **CLEF-xLSTM (FFN = 1)**   | **0.845 +/- 0.006** |
> | M5 | MOMENT                  | 1.156 +/- 0.043 |
> | | LowR-MOMENT (Rank = 4)  | 0.835 +/- 0.019 |
> | | LowR-MOMENT (Rank = 8)  | 0.849 +/- 0.006 |
> | | LowR-MOMENT (Rank = 16) | 0.875 +/- 0.023 |
> | | **CLEF-MOMENT (FFN = 0)**   | **0.786 +/- 0.003** |
> | | **CLEF-MOMENT (FFN = 1)**   | **0.778 +/- 0.015** |
>
> We find that CLEF models yield comparable or lower MAE (i.e., better) than non-CLEF and LowR models across all datasets. These results suggest that CLEF’s concept decoder, $c \odot x$, is **appropriate for capturing the dynamics of the trajectories. To supplement these experiments, we have noted the limitations of this decoder and opportunities for future improvements.**

---

> > ### Author Response · Authors · 2025-11-21
> > **Rebuttal (Part 3 of 4)**
> >
> > # Q1: Clarification on the assumptions of the concept decoder (Part 2/2)
> >
> > (Continued) We evaluate LowR models against CLEF and non-CLEF models on WOT, eICU, MIMIC-IV, and M5 for **delayed sequence editing**.
> >
> >
> > ## Delayed sequence editing
> >
> > | Dataset | Model | MAE |
> > | --- | --- | --- |
> > | MIMIC-IV | Transformer                        | 8.346 +/- 0.129 |
> > | | **LowR-Transformer (Rank = 4)**        | **7.818 +/- 0.045** |
> > | | LowR-Transformer (Rank = 8)        | 7.943 +/- 0.041 |
> > | | LowR-Transformer (Rank = 16)       | 7.928 +/- 0.027 |
> > | | CLEF-Transformer (FFN = 0)         | 7.915 +/- 0.046 |
> > | | **CLEF-Transformer (FFN = 1)**         | **7.863 +/- 0.065** |
> > | MIMIC-IV | xLSTM                      | 7.678 +/- 0.029 |
> > | | **LowR-xLSTM (Rank = 4)**      | **7.742 +/- 0.075** |
> > |  | LowR-xLSTM (Rank = 8)      | 7.853 +/- 0.026 |
> > |  | LowR-xLSTM (Rank = 16)     | 7.823 +/- 0.025 |
> > |  | CLEF-xLSTM (FFN = 0)       | 8.420 +/- 0.692 |
> > |  | **CLEF-xLSTM (FFN = 1)**       | **7.794 +/- 0.042** |
> > | MIMIC-IV | MOMENT                    | 10.807 +/- 0.054 |
> > |  | LowR-MOMENT (Rank = 4)    | 8.928 +/- 0.362 |
> > |  | LowR-MOMENT (Rank = 8)    | 8.926 +/- 0.221 |
> > |  | LowR-MOMENT (Rank = 16)   | 9.001 +/- 0.493 |
> > |  | **CLEF-MOMENT (FFN = 0)**     | **8.688 +/- 0.074** |
> > |  | **CLEF-MOMENT (FFN = 1)**     | **8.563 +/- 0.182** |
> > | eICU | Transformer                      | 8.377 +/- 0.047 |
> > |  | **LowR-Transformer (Rank = 4)**      | **7.465 +/- 0.201** |
> > |  | **LowR-Transformer (Rank = 8)**      | **7.567 +/- 0.070** |
> > |  | LowR-Transformer (Rank = 16)     | 7.788 +/- 0.128 |
> > |  | CLEF-Transformer (FFN = 0)       | 7.643 +/- 0.083 |
> > |  | **CLEF-Transformer (FFN = 1)**       | **7.566 +/- 0.040** |
> > | eICU | xLSTM                     | 7.086 +/- 0.066 |
> > |  | **LowR-xLSTM (Rank = 4)**     | **7.253 +/- 0.023** |
> > |  | LowR-xLSTM (Rank = 8)     | 7.375 +/- 0.094 |
> > |  | LowR-xLSTM (Rank = 16)    | 7.414 +/- 0.112 |
> > |  | CLEF-xLSTM (FFN = 0)      | 7.324 +/- 0.048 |
> > |  | **CLEF-xLSTM (FFN = 1)**      | **7.241 +/- 0.082** |
> > | eICU | MOMENT                    | 10.289 +/- 0.086 |
> > |  | LowR-MOMENT (Rank = 4)    | 8.873 +/- 0.157 |
> > |  | LowR-MOMENT (Rank = 8)    | 9.074 +/- 0.352 |
> > |  | LowR-MOMENT (Rank = 16)   | 9.608 +/- 1.133 |
> > |  | CLEF-MOMENT (FFN = 0)     | 8.773 +/- 0.120 |
> > |  | **CLEF-MOMENT (FFN = 1)**     | **8.472 +/- 0.259** |
> > | WOT | Transformer                      | 1.273 +/- 0.022 |
> > |  | LowR-Transformer (Rank = 4)      | 0.552 +/- 0.049 |
> > |  | LowR-Transformer (Rank = 8)      | 0.547 +/- 0.003 |
> > |  | LowR-Transformer (Rank = 16)     | 0.525 +/- 0.057 |
> > |  | **CLEF-Transformer (FFN = 0)**       | **0.494 +/- 0.045** |
> > |  | CLEF-Transformer (FFN = 1)       | 0.748 +/- 0.035 |
> > | WOT | xLSTM                      | 1.592 +/- 0.022 |
> > |  | LowR-xLSTM (Rank = 4)      | 0.857 +/- 0.086 |
> > |  | LowR-xLSTM (Rank = 8)      | 0.854 +/- 0.047 |
> > |  | LowR-xLSTM (Rank = 16)     | 0.882 +/- 0.048 |
> > |  | **CLEF-xLSTM (FFN = 0)**       | **0.724 +/- 0.016** |
> > |  | CLEF-xLSTM (FFN = 1)       | 1.028 +/- 0.030 |
> > | WOT | MOMENT                      | 0.836 +/- 0.018 |
> > |  | LowR-MOMENT (Rank = 4)      | 0.534 +/- 0.010 |
> > |  | LowR-MOMENT (Rank = 8)      | 0.542 +/- 0.005 |
> > |  | LowR-MOMENT (Rank = 16)     | 0.576 +/- 0.003 |
> > |  | **CLEF-MOMENT (FFN = 0)**       | **0.493 +/- 0.002** |
> > |  | CLEF-MOMENT (FFN = 1)       | 0.512 +/- 0.009 |
> > | M5 | Transformer                       | 1.214 +/- 0.016 |
> > |  | LowR-Transformer (Rank = 4)       | 1.133 +/- 0.036 |
> > |  | LowR-Transformer (Rank = 8)       | 1.137 +/- 0.010 |
> > |  | LowR-Transformer (Rank = 16)      | 1.149 +/- 0.012 |
> > |  | **CLEF-Transformer (FFN = 0)**        | **1.090 +/- 0.001** |
> > |  | **CLEF-Transformer (FFN = 1)**        | **1.086 +/- 0.000** |
> > | M5 | xLSTM                      | 1.345 +/- 0.034 |
> > |  | LowR-xLSTM (Rank = 4)      | 1.067 +/- 0.014 |
> > |  | LowR-xLSTM (Rank = 8)      | 1.075 +/- 0.013 |
> > |  | LowR-xLSTM (Rank = 16)     | 1.117 +/- 0.030 |
> > |  | CLEF-xLSTM (FFN = 0)       | 1.066 +/- 0.009 |
> > |  | **CLEF-xLSTM (FFN = 1)**       | **0.867 +/- 0.008** |
> > | M5 | MOMENT                    | 1.162 +/- 0.044 |
> > |  | LowR-MOMENT (Rank = 4)    | 0.858 +/- 0.015 |
> > |  | LowR-MOMENT (Rank = 8)    | 0.876 +/- 0.006 |
> > |  | LowR-MOMENT (Rank = 16)   | 0.907 +/- 0.026 |
> > |  | **CLEF-MOMENT (FFN = 0)**     | **0.819 +/- 0.004** |
> > |  | **CLEF-MOMENT (FFN = 1)**     | **0.807 +/- 0.017** |
> >
> >
> > We find that CLEF models yield comparable or lower MAE (i.e., better) than non-CLEF and LowR models across all datasets. These results suggest that CLEF’s concept decoder, $c \odot x$, is **appropriate for capturing the dynamics of the trajectories. To supplement these experiments, we have noted the limitations of this decoder and opportunities for future improvements.**

---

> > > ### Author Response · Authors · 2025-11-21
> > > **Rebuttal (Part 4 of 4)**
> > >
> > > # Q2: Clarification on limitations of the concept decoder
> > >
> > > Thank you for your valuable comment regarding scenarios in which the last observed state contains 0’s or missing data. Indeed, it can be a limitation to use the last time step $x_{t_i}$ for decoding the next time step if there are missing or zero values. Two straightforward solutions are to (1) take the mean of the historical context or (2) impute the missing values in the historical data before forecasting $t_j$. **We have added this note to our limitations.**

---

> > > > ### Author Response · Authors · 2025-11-25
> > > > **Reminder: Following up on our rebuttal**
> > > >
> > > > Dear Reviewer QAjY,
> > > >
> > > > We sincerely appreciate your insightful comments and the time you have taken to review our work. In response, we have carefully addressed each of your concerns and made the necessary revisions. We kindly ask if these revisions meet your expectations and if there are any further concerns or feedback you would like to discuss. We are more than happy to continue the conversation and address any additional points you may have.
> > > >
> > > > Thank you once again for your time and thoughtful consideration.
> > > >
> > > > Best regards,
> > > >
> > > > Authors of Submission #13156

---

### Official Review · Reviewer_goGg · 2025-11-02

**Soundness:** 3
**Presentation:** 3
**Contribution:** 3
**Rating:** 6
**Confidence:** 2

**Summary:**

This paper proposes CLEF, a novel framework for controllable sequence editing, designed to address the lack of granular control over timing and scope in the conditional generation of longitudinal data. The core idea is to learn "temporal concepts," which represent the rate of change for each variable, and apply them via a simple multiplicative decoder to generate future states in a single, non-autoregressive step. This approach is designed to handle both immediate and delayed edits—modifying a sequence at a distant future time point without generating intermediate steps. The authors extend this framework to counterfactual outcome estimation, arguing that the architecture provides implicit balancing that outperforms state-of-the-art methods, even without explicit balancing losses. The method's effectiveness is demonstrated through an extensive empirical evaluation on 8 datasets spanning biology, clinical medicine, and finance, where CLEF-augmented models show significant improvements over numerous baselines, particularly in the novel delayed editing task and in zero-shot counterfactual generation. A case study on generating "healthier" trajectories for diabetic patients further highlights the model's practical utility and interpretability.

**Strengths:**

*   **Clear, useful problem framing for controllable sequence editing.** In my opinion, the split between immediate vs. delayed editing—and the choice to define delayed editing as a single-step, non-autoregressive jump—cleanly targets a real pain point (compounding error) in scientific/clinical forecasting. This makes the task definition itself a contribution.

*   **Broad, careful empirical evaluation with new benchmarks.** I think the scope (8 datasets, 4 contributed benchmarks, 9 baselines, plus generalization and zero-shot counterfactual tests) is a major strength. It shows the idea isn’t brittle and gives the community reusable testbeds. The writing is also notably clear about how this fits vs. related work.

*   **Simple, modular architecture that travels across encoders.** In my view, the “temporal concept” (multiplicative edit) is easy to implement, computationally light, and integrates with varied sequence encoders. That portability lowers the barrier to adoption and facilitates ablations/reuse.

*   **Actionable interpretability via direct concept edits.** I like that domain experts can intervene on the learned concept vector and see resulting counterfactuals; the Type 1 Diabetes case study makes this tangible for in-silico hypothesis exploration. It’s not interpretability in the semantic-concept sense, but it is a practical, controllable handle on predictions.

*   **Good engineering trade-offs for stability vs. flexibility.** In my opinion, the “one-step to $t_j$” design is a pragmatic choice: it sacrifices full trajectory modeling to reduce error accumulation and make edits predictable. Given many operational needs are “endpoint-centric,” that’s an appealing trade-off.

**Weaknesses:**

*   **“Implicit balancing” feels asserted more than demonstrated.** In my opinion, the paper leans on architectural intuition and empirical accuracy to suggest that the learned representation is balanced, but it doesn’t directly test balance. It might read more cautiously if the claim were framed as “consistent with improved balance,” and, if feasible, paired with light diagnostics (e.g., predicting treatment from the learned representation and reporting an IPM-style distance such as MMD/HSIC between treated vs. control in the rep space).

*   **Decoder family may be too restrictive for coupled or constrained systems.** I read the diagonal, multiplicative generator $\hat{x} = c \odot x$ as elegant but narrow. In my view, this per-variable scaling risks missing cross-variable couplings, conservation/compositional constraints, or saturation effects. A short limitations note—and, if bandwidth allows, a small ablation with a lightly coupled decoder $(I+W)(c \odot x)$ where $W$ is sparse or low-rank—could clarify where the current choice shines and where it struggles.

*   **One-step “jump” to $t_j$ trades off path information for stability.** Personally, I see the single-step design as a smart way to avoid compounding autoregressive error, but it does mean path-dependent phenomena (transients, intermediate interventions, accumulation) aren’t modeled explicitly. A brief scope statement that CLEF targets endpoint editing—and a short-horizon chaining stress test (3–5 steps) to show how rollouts behave—would, in my opinion, set expectations more cleanly.

*   **Evaluation blends predictive strength with causal validity.** My sense is that strong MAE/RMSE/AUC is being read as support for the causal story, but predictive accuracy alone doesn’t confirm deconfounding. It may help to report a couple of lightweight causal diagnostics alongside accuracy—(i) treatment predictability from the learned reps, (ii) a balance/divergence metric, and (iii) a small sensitivity analysis—to separate “good predictions” from “good balancing.”

**Questions:**

* On the "Implicit Balancing" Claim: The paper's causal claims rely on the idea of "implicit balancing," which is currently supported by downstream predictive accuracy. To substantiate this claim more directly, could you provide a more targeted diagnostic for balance? For instance, could you report on either (a) the predictability of treatment assignment from the learned representations (where a lower AUC would indicate better balance) or (b) an IPM-style distance like MMD between the representation distributions for treated versus control groups?

* On the Limitations of the Multiplicative Decoder: The diagonal multiplicative decoder, x^=c⊙x, is elegant but seems to impose a strong linearity assumption. Could you please add a discussion on its potential limitations in systems with known non-linear couplings, compositional constraints (e.g., variables that must sum to a constant), or saturation effects? Clarifying the boundaries of this design choice would help readers understand its ideal application scope.

* On the Scope of Endpoint vs. Trajectory Generation: The one-step forecast is a key design choice to avoid compounding error. Could you clarify the model's intended scope (i.e., is it primarily for endpoint editing rather than full trajectory simulation)? To help illustrate this, could you include a short-horizon chaining test (e.g., 3–5 steps) to show how error accumulates when the model is applied autoregressively compared to the one-step approach?

---

> ### Author Response · Authors · 2025-11-21
> **Rebuttal (Part 1 of 4)**
>
> Thank you for your insightful comments and valuable feedback! We appreciate your acknowledgement that our paper provides a “clear, useful problem framing for controllable sequence editing” and a “broad, careful empirical evaluation with new benchmarks.” We are happy that you found CLEF to have “simple, modular architecture that travels across encoders,” “actionable interpretability via direct concept edits,” and “good engineering trade-offs for stability vs. flexibility.”
>
> We have carefully addressed your questions regarding the (Q1) the balanced representations learned by counterfactual models, (Q2) the concept decoder, and (Q3) single-step generation.
>
> We hope that our responses answer all of your questions, and we kindly ask you to consider raising your score. Please let us know if you still have concerns after reviewing our responses. Thank you again!
>
> ---
>
> # Q1: Balanced representations learned by counterfactual models
>
> We address your concern that “the paper leans on architectural intuition and empirical accuracy to suggest that the learned representation is balanced, but it doesn’t directly test balance” in two parts.
>
> **Firstly:** As per your suggestion, we have rephrased the following sentence in Section 5.4 regarding CLEF’s ability to learn balanced representations (edits in **bold**; L404-407):
>
> - “Notably, CLEF-CT and CLEF-CRN without any balancing loss (i.e., neither GR nor CDC; violet-red) are the best performing CT/CRN models. While studies have shown a trade-off between prediction accuracy and balanced representations (Huang et al., 2024; Wang et al., 2025), this finding **is consistent with improved balanced representations and** empirically **supports** Assumption 3.4.”
>
> **Secondly:** Beyond predictive accuracy metrics, we additionally evaluate the treatment predictability of CLEF’s (and non-CLEF’s) learned representations. We compute the **binary cross entropy (BCE) loss** for predicting the treatment from the learned representations of CLEF and non-CLEF models. Because “balanced” representations should be treatment-invariant, **higher BCE loss is better**. We report the BCE loss of models trained on 5 seeds.
>
> **Table 1: Tumor growth (single-sliding treatment)**
>
> | Model | BCE Loss|
> | --- | --- |
> | CT w/CDC | 1.499 +/- 0.099 |
> | **CLEF-CT w/CDC** | 1.506 +/- 0.096 |
> | CRN w/GR | 1.193 +/- 0.169 |
> | **CLEF-CRN w/GR** | 1.196 +/- 0.167 |
> | CRN w/CDC | 1.695 +/- 0.652 |
> | **CLEF-CRN w/CDC** | 1.597 +/- 0.187 |
>
> **Table 2: Tumor growth (random trajectories)**
>
> | Model | BCE Loss|
> | --- | --- |
> | CT w/CDC | 1.497 +/- 0.089 |
> | **CLEF-CT w/CDC** | 1.509 +/- 0.098 |
> | CRN w/GR | 1.194 +/- 0.168 |
> | **CLEF-CRN w/GR** | 1.196 +/- 0.167 |
> | CRN w/CDC | 1.583 +/- 0.178 |
> | **CLEF-CRN w/CDC** | 1.598 +/- 0.188 |
>
> **Table 3: Semi-synthetic patient trajectories**
>
> | Model | BCE Loss|
> | --- | --- |
> | CT w/CDC | 1.656 +/- 0.631 |
> | **CLEF-CT w/CDC** | 1.300 +/- 0.375 |
> | CRN w/GR | 0.330 +/- 0.149 |
> | **CLEF-CRN w/GR** | 0.364 +/- 0.151 |
> | CRN w/CDC | 1.706 +/- 0.662 |
> | **CLEF-CRN w/CDC** | 1.650 +/- 0.640 |
>
> **For all three datasets**, we find that CLEF models have comparable or higher (i.e., better) binary cross entropy (BCE) loss in predicting the treatment compared to non-CLEF models. These results suggest that **CLEF models learn representations that are treatment invariant.**

---

> > ### Author Response · Authors · 2025-11-21
> > **Rebuttal (Part 2 of 4)**
> >
> > # Q2: Ablation on the concept decoder (Part 1/2)
> >
> > Thank you for your valuable suggestion to perform “a small ablation with a lightly coupled decoder”! We have implemented a decoder $(I + W)(c \odot x)$ where $W$ is low-rank, with rank $= 4, 8, 16$. We refer to models with this decoder using the prefix “LowR.” We evaluate LowR models against CLEF and non-CLEF models on WOT, eICU, MIMIC-IV, and M5 for **immediate sequence editing**.
> >
> > ## Immediate sequence editing
> >
> > | Dataset | Model | MAE |
> > | --- | --- | --- |
> > | MIMIC-IV | Transformer                  |  7.284 +/- 0.005 |
> > | | LowR-Transformer (Rank = 4)  |  6.673 +/- 0.074 |
> > | | LowR-Transformer (Rank = 8)  |  6.748 +/- 0.067 |
> > | | LowR-Transformer (Rank = 16) |  6.724 +/- 0.155 |
> > | | **CLEF-Transformer (FFN = 0)**   |  **6.577 +/- 0.038** |
> > | | **CLEF-Transformer (FFN = 1)**   |  **6.562 +/- 0.046** |
> > | MIMIC-IV | xLSTM                  |  7.561 +/- 0.354 |
> > | | LowR-xLSTM (Rank = 4)  |  7.018 +/- 0.162 |
> > | | LowR-xLSTM (Rank = 8)  |  7.139 +/- 0.402 |
> > | | LowR-xLSTM (Rank = 16) |  7.031 +/- 0.058 |
> > | | CLEF-xLSTM (FFN = 0)   |  7.493 +/- 0.722 |
> > | | **CLEF-xLSTM (FFN = 1)**   |  **6.942 +/- 0.052** |
> > | MIMIC-IV | MOMENT                  |  14.804 +/- 0.036 |
> > | | LowR-MOMENT (Rank = 4)  |  10.671 +/- 0.290 |
> > | | LowR-MOMENT (Rank = 8)  |  10.603 +/- 0.093 |
> > | | LowR-MOMENT (Rank = 16) |  10.685 +/- 0.204 |
> > | | **CLEF-MOMENT (FFN = 0)**   |  **9.779 +/- 0.148** |
> > | | **CLEF-MOMENT (FFN = 1)**   |  **9.579 +/- 0.199** |
> > | eICU | Transformer                   | 9.439 +/- 0.082 |
> > | | **LowR-Transformer (Rank = 4)**   | **8.135 +/- 0.102** |
> > | | **LowR-Transformer (Rank = 8)**   | **8.265 +/- 0.109** |
> > | | LowR-Transformer (Rank = 16)  | 8.426 +/- 0.162 |
> > | | **CLEF-Transformer (FFN = 0)**    | **8.319 +/- 0.038** |
> > | | **CLEF-Transformer (FFN = 1)**    | **8.338 +/- 0.064** |
> > | eICU | xLSTM                  | 8.041 +/- 0.142 |
> > | | **LowR-xLSTM (Rank = 4)**  | **7.731 +/- 0.054** |
> > | | LowR-xLSTM (Rank = 8)  | 8.018 +/- 0.062 |
> > | | LowR-xLSTM (Rank = 16) | 8.001 +/- 0.111 |
> > | | **CLEF-xLSTM (FFN = 0)**   | **7.815 +/- 0.139** |
> > | | **CLEF-xLSTM (FFN = 1)**   | **7.751 +/- 0.050** |
> > | eICU | MOMENT                  | 13.376 +/- 0.089 |
> > | | LowR-MOMENT (Rank = 4)  | 10.405 +/- 0.099 |
> > | | LowR-MOMENT (Rank = 8)  | 10.256 +/- 0.252 |
> > | | LowR-MOMENT (Rank = 16) | 11.729 +/- 1.460 |
> > | | **CLEF-MOMENT (FFN = 0)**   | **9.955 +/- 0.164** |
> > | | CLEF-MOMENT (FFN = 1)   | 10.111 +/- 0.329 |
> > | WOT | Transformer                   | 0.360 +/- 0.003 |
> > | | LowR-Transformer (Rank = 4)   | 0.365 +/- 0.001 |
> > | | LowR-Transformer (Rank = 8)   | 0.367 +/- 0.001 |
> > | | LowR-Transformer (Rank = 16)  | 0.366 +/- 0.002 |
> > | | CLEF-Transformer (FFN = 0)    | 0.360 +/- 0.001 |
> > | | **CLEF-Transformer (FFN = 1)**    | **0.344 +/- 0.001** |
> > | WOT | xLSTM                  | 0.373 +/- 0.004 |
> > | | LowR-xLSTM (Rank = 4)  | 0.362 +/- 0.002 |
> > | | LowR-xLSTM (Rank = 8)  | 0.363 +/- 0.001 |
> > | | LowR-xLSTM (Rank = 16) | 0.365 +/- 0.001 |
> > | | CLEF-xLSTM (FFN = 0)   | 0.350 +/- 0.001 |
> > | | **CLEF-xLSTM (FFN = 1)**   | **0.348 +/- 0.001** |
> > | WOT | MOMENT                  | 0.386 +/- 0.002 |
> > | | LowR-MOMENT (Rank = 4)  | 0.370 +/- 0.003 |
> > | | LowR-MOMENT (Rank = 8)  | 0.372 +/- 0.001 |
> > | | LowR-MOMENT (Rank = 16) | 0.381 +/- 0.001 |
> > | | **CLEF-MOMENT (FFN = 0)**   | **0.356 +/- 0.001** |
> > | | **CLEF-MOMENT (FFN = 1)**   | **0.356 +/- 0.001** |
> > | M5 | Transformer                  | 1.203 +/- 0.014 |
> > | | LowR-Transformer (Rank = 4)  | 1.137 +/- 0.036 |
> > | | LowR-Transformer (Rank = 8)  | 1.136 +/- 0.015 |
> > | | LowR-Transformer (Rank = 16) | 1.151 +/- 0.011 |
> > | | **CLEF-Transformer (FFN = 0)**   | **1.089 +/- 0.001** |
> > | | **CLEF-Transformer (FFN = 1)**   | **1.086 +/- 0.000** |
> > | M5 | xLSTM                  | 1.306 +/- 0.032 |
> > | | LowR-xLSTM (Rank = 4)  | 1.029 +/- 0.011 |
> > | | LowR-xLSTM (Rank = 8)  | 1.039 +/- 0.007 |
> > | | LowR-xLSTM (Rank = 16) | 1.077 +/- 0.028 |
> > | | CLEF-xLSTM (FFN = 0)   | 1.025 +/- 0.008 |
> > | | **CLEF-xLSTM (FFN = 1)**   | **0.845 +/- 0.006** |
> > | M5 | MOMENT                  | 1.156 +/- 0.043 |
> > | | LowR-MOMENT (Rank = 4)  | 0.835 +/- 0.019 |
> > | | LowR-MOMENT (Rank = 8)  | 0.849 +/- 0.006 |
> > | | LowR-MOMENT (Rank = 16) | 0.875 +/- 0.023 |
> > | | **CLEF-MOMENT (FFN = 0)**   | **0.786 +/- 0.003** |
> > | | **CLEF-MOMENT (FFN = 1)**   | **0.778 +/- 0.015** |
> >
> > We find that CLEF models yield comparable or lower MAE (i.e., better) than non-CLEF and LowR models across all datasets. These results indicate that CLEF’s concept decoder, $c \odot x$, is **appropriate for capturing the dynamics of the trajectories. To supplement these experiments, we have noted the limitations of this decoder and opportunities for future improvements.**

---

> > > ### Author Response · Authors · 2025-11-21
> > > **Rebuttal (Part 3 of 4)**
> > >
> > > # Q2: Ablation on the concept decoder (Part 2/2)
> > >
> > > (Continued) We evaluate LowR models against CLEF and non-CLEF models on WOT, eICU, MIMIC-IV, and M5 for **delayed sequence editing**.
> > >
> > > ## Delayed sequence editing
> > >
> > > | Dataset | Model | MAE |
> > > | --- | --- | --- |
> > > | MIMIC-IV | Transformer                        | 8.346 +/- 0.129 |
> > > | | **LowR-Transformer (Rank = 4)**        | **7.818 +/- 0.045** |
> > > | | LowR-Transformer (Rank = 8)        | 7.943 +/- 0.041 |
> > > | | LowR-Transformer (Rank = 16)       | 7.928 +/- 0.027 |
> > > | | CLEF-Transformer (FFN = 0)         | 7.915 +/- 0.046 |
> > > | | **CLEF-Transformer (FFN = 1)**         | **7.863 +/- 0.065** |
> > > | MIMIC-IV | xLSTM                      | 7.678 +/- 0.029 |
> > > | | **LowR-xLSTM (Rank = 4)**      | **7.742 +/- 0.075** |
> > > |  | LowR-xLSTM (Rank = 8)      | 7.853 +/- 0.026 |
> > > |  | LowR-xLSTM (Rank = 16)     | 7.823 +/- 0.025 |
> > > |  | CLEF-xLSTM (FFN = 0)       | 8.420 +/- 0.692 |
> > > |  | **CLEF-xLSTM (FFN = 1)**       | **7.794 +/- 0.042** |
> > > | MIMIC-IV | MOMENT                    | 10.807 +/- 0.054 |
> > > |  | LowR-MOMENT (Rank = 4)    | 8.928 +/- 0.362 |
> > > |  | LowR-MOMENT (Rank = 8)    | 8.926 +/- 0.221 |
> > > |  | LowR-MOMENT (Rank = 16)   | 9.001 +/- 0.493 |
> > > |  | **CLEF-MOMENT (FFN = 0)**     | **8.688 +/- 0.074** |
> > > |  | **CLEF-MOMENT (FFN = 1)**     | **8.563 +/- 0.182** |
> > > | eICU | Transformer                      | 8.377 +/- 0.047 |
> > > |  | **LowR-Transformer (Rank = 4)**      | **7.465 +/- 0.201** |
> > > |  | **LowR-Transformer (Rank = 8)**      | **7.567 +/- 0.070** |
> > > |  | LowR-Transformer (Rank = 16)     | 7.788 +/- 0.128 |
> > > |  | CLEF-Transformer (FFN = 0)       | 7.643 +/- 0.083 |
> > > |  | **CLEF-Transformer (FFN = 1)**       | **7.566 +/- 0.040** |
> > > | eICU | xLSTM                     | 7.086 +/- 0.066 |
> > > |  | **LowR-xLSTM (Rank = 4)**     | **7.253 +/- 0.023** |
> > > |  | LowR-xLSTM (Rank = 8)     | 7.375 +/- 0.094 |
> > > |  | LowR-xLSTM (Rank = 16)    | 7.414 +/- 0.112 |
> > > |  | CLEF-xLSTM (FFN = 0)      | 7.324 +/- 0.048 |
> > > |  | **CLEF-xLSTM (FFN = 1)**      | **7.241 +/- 0.082** |
> > > | eICU | MOMENT                    | 10.289 +/- 0.086 |
> > > |  | LowR-MOMENT (Rank = 4)    | 8.873 +/- 0.157 |
> > > |  | LowR-MOMENT (Rank = 8)    | 9.074 +/- 0.352 |
> > > |  | LowR-MOMENT (Rank = 16)   | 9.608 +/- 1.133 |
> > > |  | CLEF-MOMENT (FFN = 0)     | 8.773 +/- 0.120 |
> > > |  | **CLEF-MOMENT (FFN = 1)**     | **8.472 +/- 0.259** |
> > > | WOT | Transformer                      | 1.273 +/- 0.022 |
> > > |  | LowR-Transformer (Rank = 4)      | 0.552 +/- 0.049 |
> > > |  | LowR-Transformer (Rank = 8)      | 0.547 +/- 0.003 |
> > > |  | LowR-Transformer (Rank = 16)     | 0.525 +/- 0.057 |
> > > |  | **CLEF-Transformer (FFN = 0)**       | **0.494 +/- 0.045** |
> > > |  | CLEF-Transformer (FFN = 1)       | 0.748 +/- 0.035 |
> > > | WOT | xLSTM                      | 1.592 +/- 0.022 |
> > > |  | LowR-xLSTM (Rank = 4)      | 0.857 +/- 0.086 |
> > > |  | LowR-xLSTM (Rank = 8)      | 0.854 +/- 0.047 |
> > > |  | LowR-xLSTM (Rank = 16)     | 0.882 +/- 0.048 |
> > > |  | **CLEF-xLSTM (FFN = 0)**       | **0.724 +/- 0.016** |
> > > |  | CLEF-xLSTM (FFN = 1)       | 1.028 +/- 0.030 |
> > > | WOT | MOMENT                      | 0.836 +/- 0.018 |
> > > |  | LowR-MOMENT (Rank = 4)      | 0.534 +/- 0.010 |
> > > |  | LowR-MOMENT (Rank = 8)      | 0.542 +/- 0.005 |
> > > |  | LowR-MOMENT (Rank = 16)     | 0.576 +/- 0.003 |
> > > |  | **CLEF-MOMENT (FFN = 0)**       | **0.493 +/- 0.002** |
> > > |  | CLEF-MOMENT (FFN = 1)       | 0.512 +/- 0.009 |
> > > | M5 | Transformer                       | 1.214 +/- 0.016 |
> > > |  | LowR-Transformer (Rank = 4)       | 1.133 +/- 0.036 |
> > > |  | LowR-Transformer (Rank = 8)       | 1.137 +/- 0.010 |
> > > |  | LowR-Transformer (Rank = 16)      | 1.149 +/- 0.012 |
> > > |  | **CLEF-Transformer (FFN = 0)**        | **1.090 +/- 0.001** |
> > > |  | **CLEF-Transformer (FFN = 1)**        | **1.086 +/- 0.000** |
> > > | M5 | xLSTM                      | 1.345 +/- 0.034 |
> > > |  | LowR-xLSTM (Rank = 4)      | 1.067 +/- 0.014 |
> > > |  | LowR-xLSTM (Rank = 8)      | 1.075 +/- 0.013 |
> > > |  | LowR-xLSTM (Rank = 16)     | 1.117 +/- 0.030 |
> > > |  | CLEF-xLSTM (FFN = 0)       | 1.066 +/- 0.009 |
> > > |  | **CLEF-xLSTM (FFN = 1)**       | **0.867 +/- 0.008** |
> > > | M5 | MOMENT                    | 1.162 +/- 0.044 |
> > > |  | LowR-MOMENT (Rank = 4)    | 0.858 +/- 0.015 |
> > > |  | LowR-MOMENT (Rank = 8)    | 0.876 +/- 0.006 |
> > > |  | LowR-MOMENT (Rank = 16)   | 0.907 +/- 0.026 |
> > > |  | **CLEF-MOMENT (FFN = 0)**     | **0.819 +/- 0.004** |
> > > |  | **CLEF-MOMENT (FFN = 1)**     | **0.807 +/- 0.017** |
> > >
> > >
> > > We find that CLEF models yield comparable or lower MAE (i.e., better) than non-CLEF and LowR models across all datasets. These results indicate that CLEF’s concept decoder, $c \odot x$, is **appropriate for capturing the dynamics of the trajectories. To supplement these experiments, we have noted the limitations of this decoder and opportunities for future improvements.**

---

> > > > ### Author Response · Authors · 2025-11-21
> > > > **Rebuttal (Part 4 of 4)**
> > > >
> > > > # Q3: Clarification on single-step generation
> > > >
> > > > Thank you for recognizing that “single-step design as a smart way to avoid compounding autoregressive error” and raising the concern that “path-dependent phenomena (transients, intermediate interventions, accumulation) aren’t modeled explicitly.”
> > > >
> > > > To clarify, we evaluate CLEF on datasets of longitudinal trajectories of cells, patients, and sales. While the experiments in the paper focus on predicting the observed $x_{t_j}$ at time $t_j$ based on the observed $x_{t_0:t_i}$ where $t_j > t_i$, **it is possible to use CLEF for predicting intermediate (unobserved or observed) steps** $t_i < t_q < t_j$. That said, the benchmarking results in the paper show CLEF’s performance on observed data only.
> > > >
> > > > As per your request, we perform an experiment in which we arbitrarily add three intermediate steps between $t_i$ and $t_j$ before finally predicting the observed $x_{t_j}$. Because we only have the ground truth for $x_{t_j}$, we evaluate only on the predicted $x_{t_j}$.
> > > >
> > > > | Dataset | Model | MAE |
> > > > | --- | --- | --- |
> > > > | MIMIC-IV | Transformer                        | 8.346 +/- 0.129 |
> > > > |  | Transformer(intermediate)          | 8.627 +/- 0.365 |
> > > > |  | **CLEF-Transformer**         | **7.915 +/- 0.046** |
> > > > |  | CLEF-Transformer(intermediate)     | 8.279 +/- 0.135 |
> > > > | MIMIC-IV | **xLSTM**                      | **7.678 +/- 0.029** |
> > > > |  | xLSTM(intermediate)        | 8.281 +/- 0.414 |
> > > > |  | **CLEF-xLSTM**       | **8.420 +/- 0.692** |
> > > > |  | CLEF-xLSTM(intermediate)   | 8.684 +/- 0.767 |
> > > > | MIMIC-IV | MOMENT                    | 10.807 +/- 0.054 |
> > > > |  | MOMENT(intermediate)      | 10.673 +/- 0.059 |
> > > > |  | **CLEF-MOMENT**     | **8.688 +/- 0.074** |
> > > > |  | CLEF-MOMENT(intermediate) | 9.096 +/- 0.137 |
> > > > | eICU | Transformer                      | 8.377 +/- 0.047 |
> > > > |  | Transformer(intermediate)        | 9.707 +/- 1.646 |
> > > > |  | **CLEF-Transformer**       | **7.643 +/- 0.083** |
> > > > |  | CLEF-Transformer(intermediate)   | 8.144 +/- 0.247 |
> > > > | eICU | **xLSTM**                     | **7.086 +/- 0.066** |
> > > > |  | xLSTM(intermediate)       | 8.015 +/- 0.566 |
> > > > |  | **CLEF-xLSTM**      | **7.324 +/- 0.048** |
> > > > |  | CLEF-xLSTM(intermediate)  | 8.121 +/- 0.371 |
> > > > | eICU | MOMENT                    | 10.289 +/- 0.086 |
> > > > |  | MOMENT(intermediate)      | 10.233 +/- 0.069 |
> > > > |  | **CLEF-MOMENT**     | **8.773 +/- 0.120** |
> > > > |  | CLEF-MOMENT(intermediate) | 9.591 +/- 0.198 |
> > > > | WOT | Transformer                      | 1.273 +/- 0.022 |
> > > > |  | Transformer(intermediate)        | 1.068 +/- 0.053 |
> > > > |  | CLEF-Transformer       | 0.494 +/- 0.045 |
> > > > |  | **CLEF-Transformer(intermediate)**   | **0.411 +/- 0.026** |
> > > > | WOT | xLSTM                      | 1.592 +/- 0.022 |
> > > > |  | xLSTM(intermediate)        | 1.444 +/- 0.043 |
> > > > |  | CLEF-xLSTM       | 0.724 +/- 0.016 |
> > > > |  | **CLEF-xLSTM(intermediate)**   | **0.395 +/- 0.010** |
> > > > | WOT | MOMENT                      | 0.836 +/- 0.018 |
> > > > |  | MOMENT(intermediate)        | 0.764 +/- 0.015 |
> > > > |  | CLEF-MOMENT       | 0.493 +/- 0.002 |
> > > > |  | **CLEF-MOMENT(intermediate)**   | **0.463 +/- 0.003** |
> > > > | M5 | Transformer                       | 1.214 +/- 0.016 |
> > > > |  | Transformer(intermediate)         | 1.205 +/- 0.016 |
> > > > |  | CLEF-Transformer        | 1.090 +/- 0.001 |
> > > > |  | **CLEF-Transformer(intermediate)**    | **1.075 +/- 0.001** |
> > > > | M5 | xLSTM                      | 1.345 +/- 0.034 |
> > > > |  | xLSTM(intermediate)        | 1.271 +/- 0.030 |
> > > > |  | **CLEF-xLSTM**       | **1.066 +/- 0.009** |
> > > > |  | CLEF-xLSTM(intermediate)   | 1.121 +/- 0.011 |
> > > > | M5 | MOMENT                    | 1.162 +/- 0.044 |
> > > > |  | MOMENT(intermediate)      | 1.137 +/- 0.043 |
> > > > |  | CLEF-MOMENT     | 0.819 +/- 0.004 |
> > > > |  | **CLEF-MOMENT(intermediate)** | **0.799 +/- 0.005** |
> > > >
> > > > On the patient datasets, MIMIC-IV and eICU, we find that **adding intermediate steps leads to compounding autoregressive error**. However, on the cellular reprogramming (WOT) and Walmart sales (M5) datasets, **adding intermediate steps can improve the performance**. There are a few possible reasons for these findings:
> > > >
> > > > - The patient datasets have irregular and large time intervals (e.g., hours, days, weeks, months, years), whereas the cellular reprogramming and Walmart sales datasets have regular and small time intervals (e.g., every 12 hours, daily).
> > > >
> > > > - Because MIMIC-IV and eICU capture patients in the intensive care unit, these patients’ lab test measurements (e.g., white blood cell count, glucose) can change rapidly.
> > > >
> > > > Arbitrarily adding intermediate steps may introduce noise that compounds and degrades predictions at $t_j$. On the other hand, the progression of cellular and sales trajectories may fluctuate less during the relatively small and regular time interval, so generating intermediate steps may improve the forecasting of measurements (e.g., gene expression, sales) at $t_j$.

---

> > > > > ### Author Response · Authors · 2025-11-25
> > > > > **Reminder: Following up on our rebuttal**
> > > > >
> > > > > Dear Reviewer goGg,
> > > > >
> > > > > We sincerely appreciate your insightful comments and the time you have taken to review our work. In response, we have carefully addressed each of your concerns and made the necessary revisions. We kindly ask if these revisions meet your expectations and if there are any further concerns or feedback you would like to discuss. We are more than happy to continue the conversation and address any additional points you may have.
> > > > >
> > > > > Thank you once again for your time and thoughtful consideration.
> > > > >
> > > > > Best regards,
> > > > >
> > > > > Authors of Submission #13156

---

### Author Response · Authors · 2025-12-03
**Summary of Reviews and Rebuttals**

We summarized our reviews & rebuttals. Many thanks to the reviewers, ACs, & PCs for your feedback, time, and thoughtful consideration.

---

# Strengths

The 4 reviewers highlight 5 major strengths:

- **Novel & meaningful problem framing** (Quotes: “clear, useful problem framing for controllable sequence editing”; “addresses an interesting problem with significant practical applications”; “cleanly targets a real pain point (compounding error) in scientific/clinical forecasting”; “the work addresses [...] an important problem in computational biology and healthcare”)

- **Elegant & practical methodology** (Quotes: “easy to implement, computationally light, and integrates with varied sequence encoders”; the “portability lowers the barrier to adoption and facilitates ablations/reuse”; “simple, modular architecture that travels across encoders”; “model design is simple and clean”; “good engineering trade-offs for stability vs. flexibility”)

- **Novel task & capabilities** (Quotes: “the formalization of ‘delayed sequence editing’ as a one-step generation task is [an] interesting contribution” and it “distinguishes the problem from standard auto-regressive forecasting”; “the task definition itself [is] a contribution”; “actionable interpretability via direct concept edits”; “tangible for in-silico hypothesis exploration”)

- **Comprehensive empirical evaluation & new benchmarks** (Quotes: “broad, careful empirical evaluation with new benchmarks”; “the scope (8 datasets, 4 contributed benchmarks, 9 baselines, plus generalization and zero-shot counterfactual tests) is a major strength”; “experimental section is thorough and convincing”; “the method is evaluated on a substantial number of datasets from three domains”; “good empirical results CLEF-based models demonstrate consistent and often large improvements over their non-CLEF counterparts”)

- **Well-written paper & clearly motivated method** (Quotes: “is well-written and the proposed method is clearly motivated”; “the writing is also notably clear about how this fits vs. related work”; “introduction motivates the problem well”; “Fig. 1 is informative”; “writing is generally clear”; “figures are descriptive and polished”)


---


# Revisions

We expanded the performance metrics, ablations, baselines, literature review, limitations, and future work. These revisions carefully & thoroughly address all reviewers’ comments.

- **New performance metric on balanced representations learned by counterfactual models.** We added empirical results to demonstrate that CLEF learns balanced representations (i.e., treatment invariant). We computed BCE loss for predicting the treatment using the model’s learned representations. On all **3 benchmark datasets** for counterfactual outcomes estimation, we reported BCE loss of **6 models** trained on **5 seeds**. We find that CLEF models have comparable or better BCE loss in predicting the treatment compared to non-CLEF models, suggesting that CLEF models learn representations that are treatment invariant.

- **New ablations on concept decoder.** To evaluate the strengths and limitations of CLEF’s concept decoder, we implemented **3 ablations**. On all **4 benchmark datasets**, we reported MAE of these ablations trained on **3 seeds** and evaluated on **2 tasks**. We find that CLEF models yield comparable or better MAE than non-CLEF and ablated models across all datasets, indicating that CLEF’s concept decoder is appropriate for capturing the dynamics of the trajectories.

- **New ablations to evaluate single-step generation.** To empirically demonstrate that single-step generation can help avoid compounding autoregressive error, we performed an experiment in which we arbitrarily added 3 intermediate steps before finally predicting the observed readouts. On all **4 benchmark datasets**, we reported MAE of **6 ablations** trained on **3 seeds**. We find that CLEF models yield comparable or better MAE than non-CLEF and the new ablated models in real-world patient datasets, indicating that adding intermediate steps can lead to compounding autoregressive error.

- **New baseline to evaluate intervention mechanics.** We added a baseline to generate intervened trajectories and compared them against CLEF’s. On the **2 datasets** of patient trajectories, we compared their trajectories generated via **2 types of interventions**. Unlike CLEF, the baseline cannot generate trajectories that resemble the trajectories of healthier or sicker patients, depending on the intervention.

- **New qualitative trajectories.** We provided trajectory examples for **2 patients**. CLEF seems to generate lab test values that are closer to the observed lab test values than non-CLEF models.

- **Extended literature review, limitations, & future work.** We clarified the distinction between CLEF and biological sequence design and reinforcement learning methods. We included more details about the limitations of the concept decoder and opportunities to extend CLEF.

---

### Meta-Review · Area_Chair_a7k8 · 2026-01-07

**Summary:**

The paper proposes CLEF, a framework for controllable sequence editing designed to modify longitudinal sequences based on specific conditions. Reviewers questioned whether the linear assumption relating future and current states was sufficient for complex dynamics; the authors addressed this via a "LowR" experiment, showing that increasing decoder complexity did not improve performance, which I find acceptable. Regarding the concern that a SimpleLinear baseline outperformed deep learning methods in most settings, the authors clarified that their approach enables specific capabilities like zero-shot generation and concept intervention, which linear models cannot perform. The rebuttal addressed the primary concerns with experimental support. I encourage the authors to further investigate the comparison with the SimpleLinear baseline to determine whether this result stems from dataset bias or deep model's overfitting to define the proposed method's boundaries. Overall, I recommend acceptance.

**Reviewer Concerns:**

The authors addressed the shared concern regarding the linear assumption via the "LowR" experiment, which showed that introducing additional relations actually degraded performance. Regarding Reviewer goGg's concern that high downstream accuracy is insufficient to prove Implicit Balancing, the authors rephrased their claims and provided further experimental support. They also addressed Reviewer goGg's worry about path dependency in the single-step design through additional experiments. For Reviewer QAjY's point that zero values in the current state cause prediction failure, the authors acknowledged this limitation and proposed using a longer historical window as a mitigation. Additionally, the authors clarified questions regarding related work and implementation details. Regarding the concern that the SimpleLinear baseline outperformed many deep learning methods, the authors highlighted their method's unique capabilities, such as zero-shot generation, which linear models cannot perform.  Further analysis is needed to explain the strong performance of the linear baseline and define the proposed model's effective boundaries.

**Reviewer Scores:**

I anticipate that, with the exception of Reviewer hatS, the other reviewers will finally give relatively positive evaluations.

goGg 6->6
QAjY 4->6
v6m5 6->6
hatS 4->4

But I consider Reviewer hatS's concern regarding the SimpleLinear baseline to be important. I view this not as a critique of the method proposed in this paper, but rather as a reflection of the bias inherent in current base models for this specific task, although this does weaken the value of the paper.

---

### Decision · Program_Chairs · 2026-01-26

Accept (Poster)